# Integration of rate and phase codes by hippocampal cell-assemblies supports flexible encoding of spatiotemporal context

Eleonora Russo [1,2,3] ✉, Nadine Becker[4,6], Aleks P. F. Domanski[4], Timothy Howe[4], Kipp Freud[5], Daniel Durstewitz [2,7] & Matthew W. Jones [4,7] ✉

Spatial information is encoded by location-dependent hippocampal place cell firing rates and sub-second, rhythmic entrainment of spike times. These rate and temporal codes have primarily been characterized in low-dimensional environments under limited cognitive demands; but how is coding configured in complex environments when individual place cells signal several locations, individual locations contribute to multiple routes and functional demands vary? Quantifying CA1 population dynamics of male rats during a decision-making task, here we show that the phase of individual place cells' spikes relative to the local theta rhythm shifts to differentiate activity in different place fields. Theta phase coding also disambiguates repeated visits to the same location during different routes, particularly preceding spatial decisions. Using unsupervised detection of cell assemblies alongside theoretical simulation, we show that integrating rate and phase coding mechanisms dynamically recruits units to different assemblies, generating spiking sequences that disambiguate episodes of experience and multiplexing spatial information with cognitive context.

Hippocampal coding multiplexes over broad temporal scales incorporating prior, current, and future contextual information[1,2]. Among pyramidal cells of hippocampal CA1, transient firing rate increases lasting from hundreds to thousands of milliseconds encode the position of an animal within the environment (place cells[3,4]), routes through paths with overlapping segments (splitter cells[5–7]), signal goal-locations[8], mark time intervals[9], respond to specific odors[10], sounds[11], objects[12] and, in humans, to other people's identities[13]. The information required to form these multimodal representations[14], converges on the hippocampus from cortical and subcortical regions[15], building context-specific cognitive rate-maps[16,17].

In conjunction with rate coding, hippocampal units also coordinate at much faster timescales, entrained to the dominant state-dependent oscillations of the local field potential (LFP): 5–10 Hz theta rhythms during active exploration and REM sleep, and sharp wave-ripples (SWR) during immobility and non-REM sleep. Theta phase precession[18], theta sequences[19], and SWR-associated replay[20] produce single- or multi-unit activity patterns with temporal precision on the order of tens of milliseconds. During phase precession, the spike times of a place cell with respect to the ongoing theta oscillation shift to earlier phases in the cycle as the animal moves through that unit's spatial receptive field (place field). At the population level, the tem-

[1]The BioRobotics Institute, Department of Excellence in Robotics and AI, Scuola Superiore Sant'Anna, 56025 Pisa, Italy. [2]Dept. of Theoretical Neuroscience, Central Institute of Mental Health, Medical Faculty Mannheim, Heidelberg University, 68159 Mannheim, Germany. [3]Department of Psychiatry and Psychotherapy, University Medical Center, Johannes Gutenberg University, 55131 Mainz, Germany. [4]School of Physiology, Pharmacology & Neuroscience, Faculty of Health and Life Sciences, University of Bristol, University Walk, Bristol BS8 1TD, UK. [5]School of Computer Science, Merchant Venturers Building, University of Bristol, Woodland Road, Bristol BS8 1UB, UK. [6]Present address: Nanion Technologies GmbH, Ganghoferstr. 70A, D-80339 Munich, Germany. [7]These authors contributed equally: Daniel Durstewitz, Matthew W. Jones. ✉e-mail: eleonora.russo@santannapisa.it; matt.jones@bristol.ac.uk

porally ordered, sequential activation of place cells within a theta cycle gives rise to characteristic theta sequences. Both processes provide a temporal code for spatial information: during phase precession, the position of the animal within a cell's place field correlates with the theta phase of that cell's spikes[18,21], while theta sequences reflect past and imminent trajectories[22,23]. In addition to spatial information, recent studies have uncovered theta sequences reflecting sequences of events[24], current goals[25,26], and hypothetical future experiences[27], suggesting contributions of hippocampal temporal coding to planning and speculation. There is also growing evidence that spikes fired during different relative phases of local theta cycles may encode different aspects of past and future experiences[28,29].

Despite their different timescales, the information content and processes governing hippocampal rate and temporal coding are not independent[30]. Firstly, the order in which units activate during theta cycles and replays typically reflects the sequences in which place fields are crossed by the animal during exploration (but see also ref. 31), or sequences in which sensory cues are encountered[24]. Mechanistically, the interplay between fast somatic inhibition and slow dendritic depolarization as the animal crosses the respective neuron's place field has been proposed as a possible mechanism linking firing rate with phase precession[32–35] that may be tuned by local inhibitory interneurons[36]. Whatever their mechanisms, despite prevailing evidence for cross-temporal dependencies, the functional implications of interactions between rate and phase coding for hippocampal information processing remain equivocal.

Here, we investigate and quantify the extent to which flexible and transient activation of place cell assemblies affects both firing rate and phase/temporal coding modalities of individual place cells, enabling the discrimination of different visits to the same locations under varied cognitive demands. We, therefore, used a method for unsupervised detection of functional assemblies that is able to identify the composition of detected assemblies alongside their characteristic coordination timescales. In particular, we aimed to quantify how the information encoded by rate-assemblies at 100 ms – 1 sec timescales can affect the <100 ms temporal coding of their constituent units in the CA1 region of rats performing a spatial memory and decision-making task on a complex maze. Under these conditions, we found that both the theta phase and the firing rate of place cells shift when the cell activates within different assemblies recruited according to task trial demands. Rate and temporal codes, therefore, interact, allowing CA1 populations to parse repeated visits to the same locations into different cognitive contexts.

## Results

Six adult male Long Evans rats were trained to perform a spatial working memory decision-making task on an end-to-end T-maze[37,38]. During each trial, rats learned to run from one side of the maze to the opposite to collect 0.1 ml of sucrose solution at reward locations. Trials were subdivided into free choice and guided runs. During choice runs, rats started from one of the two reward locations marked with G in Fig. 1a and were directed by a moveable door to turn right (from G1) or left (from G2) into the central arm of the maze. Having traversed the central arm, rats had to choose whether to turn right or left at the open T-junction to continue towards the reward locations in C. A correct choice required rats to leave the central arm by turning in the same direction as they entered it (i.e., correct runs were from G1-C1 or G2-C2). The reward was delivered only upon correct trials. Choice trials were followed by guided trials that led the rats back to the G side of the maze via a pair of predetermined turns guided by motorized moveable doors. All guided runs ended with a reward. Data presented here are from rats that had learned task rules over between 16 and 23 days of habituation and training, and were performing 40 trials per recording session at between 71–90% correct. A total of 322 units was recorded from the dorsal CA1 (Supplementary Fig. 1) during 24 recording

sessions from six rats. Among these, we isolated putative place cells by selecting units with a mean, on-maze firing rate between 0.2 and 4 Hz and with spatial information above 0.5 bit/s[39]. The following analyses focus exclusively on the 218 units identified as putative place cells (with a median of 7, a minimum of 2, and a maximum of 20 putative place cells per session).

Cell assemblies were identified with an unsupervised machine-learning algorithm for cell assembly detection (CAD)[40]. CAD detects and tests arbitrary multi-unit activity patterns that re-occur more frequently than chance in parallel single-unit recordings. The algorithm automatically corrects for non-stationarity in the units' activities and scans spike count time series at multiple temporal resolutions, returning the characteristic timescales at which individual assembly patterns coordinate (Supplementary Fig. 2). Thanks to a flexible agglomeration algorithm, CAD can detect assemblies with any activity pattern, avoiding a priori suppositions about the characteristics of the detected motifs (see Methods). We use the term cell assemblies without making any assumptions about the anatomical connectivity between assembly-units, which are identified solely based on their co-activation. Here, we thus refer to a functional cell assembly as any group of units whose activation coordinates with temporal precision between 5 ms and 5 s, and arbitrary time lags between the unit activations, in a consistently reoccurring pattern.

### The two predominant timescales of hippocampal assemblies

As expected based on extensive previous analyses of place cell physiology, the temporal precision of hippocampal assemblies active during the task ranged from milliseconds to seconds and is bimodally distributed (Hartigan's dip test for unimodality, n. bootstrap samples = $10^5$, dip = 0.03, $p = 0$) into two major groups. We found: (1) 137 sharp spike patterns involving on average about 17% units per session per pattern (with a maximum of a 3-unit assembly in a 20-unit set) and temporal precision in the range of 0.006 – 0.06 sec centered around 0.028 sec (spike-assemblies) and (2) 204 broader firing rate patterns with on average 28% units per session per pattern (with a maximum of an 11-unit assembly in a 19-unit set) and temporal precision between 0.07–5 sec (rate-assemblies) (Fig. 1b). This segregation into different timescales did not, however, correspond to different hippocampal cell populations; rather, spike- and rate-assemblies were composed of largely overlapping populations. About 83% of all assembly-units participated in assemblies at both timescales. Moreover, two units taking part in the same spike-assembly were more likely to join the same rate-assembly than expected by chance (average probability of 0.9 against a chance level of 0.6, $p < 10^{-5}$ computed by bootstrap, see Methods). Consistent with previous place cell analyses, this indicates that the same sets of hippocampal units coordinated at temporal precisions of both tens and hundreds of milliseconds[41,42].

In order to understand the origin of these two characteristic timescales, we examined assembly activations in space and time. Assemblies are considered active whenever all units composing the assembly fire spikes matching the assembly activation pattern identified by the algorithm. The assembly is considered to have an activation of $n$, when all units composing the assembly fire at least $n$ spikes in the bins matching the assembly-pattern. Figure 1 shows representative examples of activity patterns (Fig. 1d, e) and activation maps (Fig. 1f, g) for both assembly groups. Rate-assemblies reflected the simultaneous or sequential activation of the place fields of their constituent units in specific maze locations and/or along task-relevant trajectories, respectively (Supplementary Fig. 3). Their characteristic temporal scale, ranging from hundreds of milliseconds to seconds, was indeed compatible with the time needed by the animal to traverse the place field of a unit. Spike-assemblies, whose timescale is compatible with replay events or theta sequences[19,20], had a more localized activation (with average spatial information of 2.67 ± 0.09 bit/s in contrast to

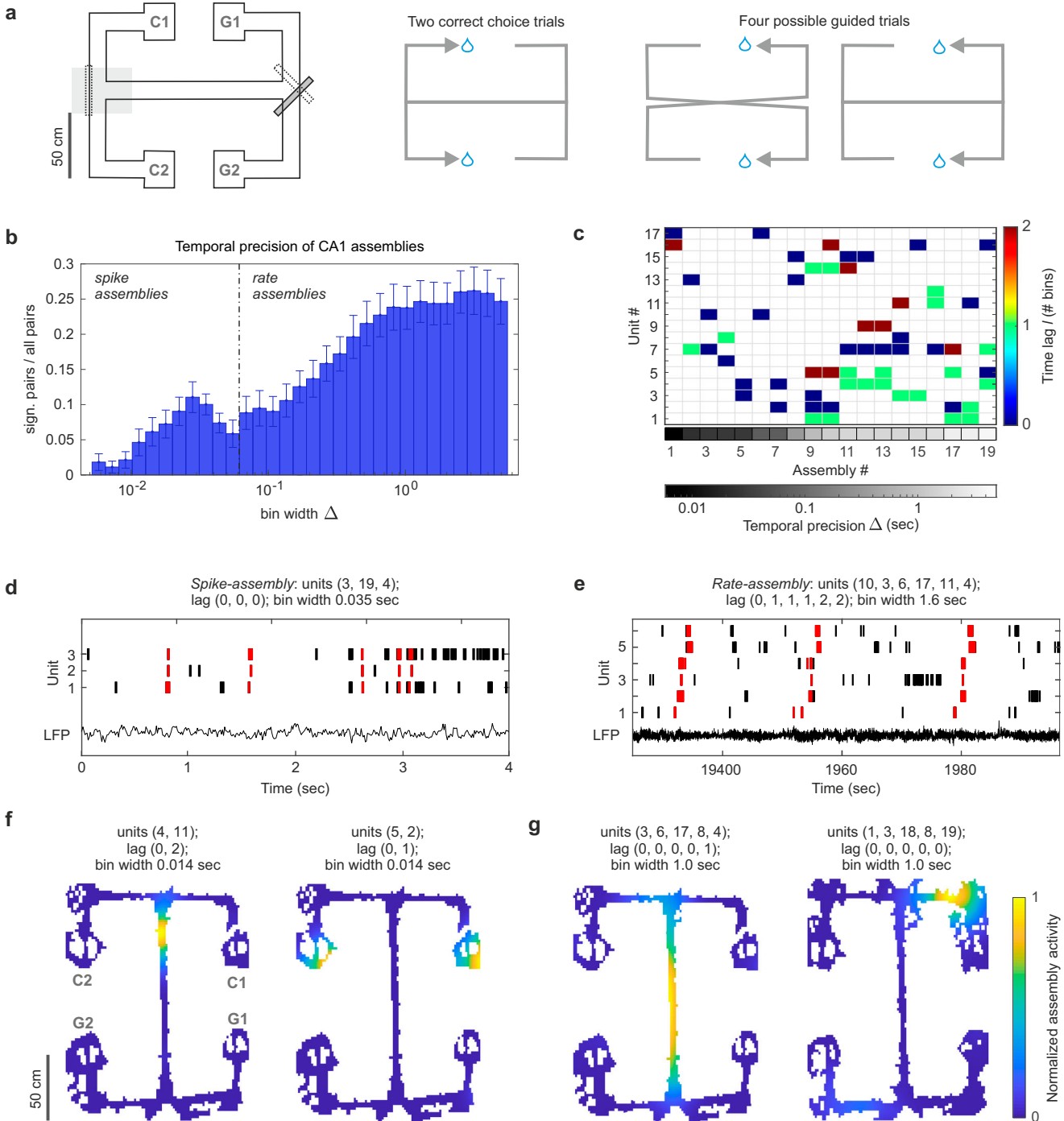

**Fig. 1 | The two timescales of hippocampal assemblies. a** Maze and task schematic. Task trials constitute choice and guided runs. In choice runs, the animal runs in direction G → C, deciding between left or right turns at the T-junction marked in gray. The correct choice is contingent upon the starting location. In guided trials, the animal runs in direction C → G, following a predetermined path guided by motorized moveable doors; **b** The distribution of the temporal precision of the assemblies detected during the spatial working memory task shows the presence of two predominant timescales: one peaked around 28 ms and one on the second scale. Bars show weighted mean and SE computed across six animals and four sessions (sessions without assemblies were excluded, *n. sessions* = 22, *n. assembly pairs* = 4914). The mean is weighted by the number of place cells recorded in each session. See Supplementary Fig. 4 for the same analysis on assemblies detected excluding spikes fired during SWR; **c** Assembly-assignment matrix for one exemplary dataset. The grayscale indicates the temporal resolution at which assemblies are detected; the color scale shows the lag between the activation of each assembly unit with respect to the unit first active in the assembly. Units marked in dark blue (lag of 0) are the first to activate within the assembly, and units marked in dark red are the last (two bins after the activation of the first assembly unit). Hippocampal place cell units were typically found taking part in multiple assemblies; **d, e** Examples of spike- (**d**) and rate- (**e**) assembly activity patterns (red) and raw LFP. Temporal resolution, composing units, and lag of activation of each unit with respect to the activation of the first assembly unit are indicated in the figure. Spike-assembly activations appear to be locked to the theta rhythm of the LFP; **f, g** Example of spike- (**f**) and rate- (**g**) assembly activation maps (activity normalized to 1, unit number indicated with respect to the relative session). See also Supplementary Fig. 3 for place fields of the relative assembly composing units. Source data are provided as a Source Data file.

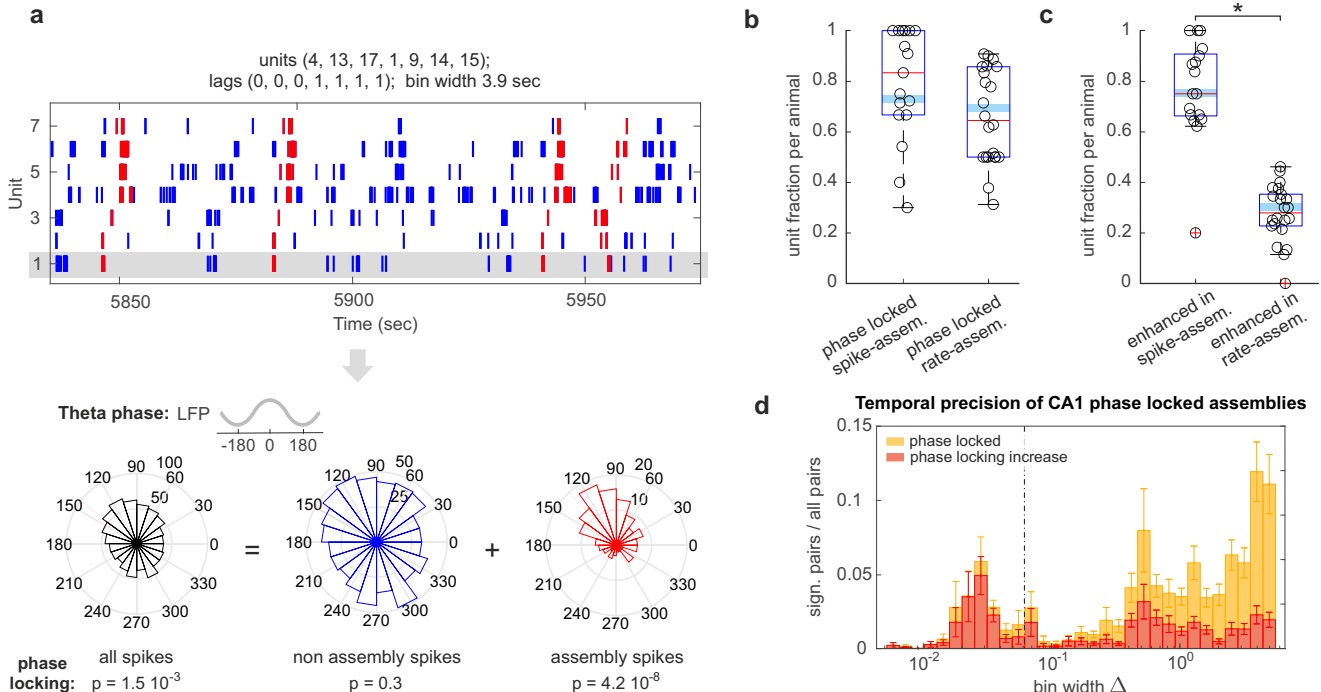

**Fig. 2 | Spike-assemblies fire phase-locked spikes. a** Raster plot of one typical assembly and its composing units (spikes fired during assembly activations marked in red). Below is a phase histogram of the spikes of one exemplary unit (highlighted in gray) showing how assembly-spikes (red) have an enhanced phase modulation with respect to either all (black) or non-assembly (blue) spikes. In the figure, *p* values for the Hodges–Ajne two-sided nonparametric test for non-uniformity, no multiple-comparison adjustments (see Methods); **b** Fraction of units with phase-modulated assembly-spikes and **c** units with assembly-spikes with enhanced phase-modulation with respect to the totality of the unit spikes (i.e., red vs. black in (**a**)) in at least one of the assemblies they take part in. Boxplots mark the median (red), the mean weighted by the number of tested assemblies per session (cyan), min and max point (whiskers), and the 25th and 75th percentiles (bottom and top edges of the box) computed across animals after Benjamini–Hochberg correction for multiple comparisons ($\alpha = 0.05$), data points correspond to individual recording sessions (four sessions of six rats, sessions where no units met the inclusion criteria

were excluded, $n = 17$ and 22 for the first and second bar, respectively, for both (**b**, **c**). Generalized linear mixed-effects model with logit link function: in **b** $F_{(1,1805)} = 0.15$, $p = 0.70$ and in **c** $F_{(1,1310)} = 134.15$, $p = 1.3 \times 10^{-29}$). * marks significance. **d** Temporal precision of spike- and rate- assemblies with phase-modulated spikes (yellow) and assemblies with spikes with enhanced phase-modulation with respect to their composing units (red). Assembly-pairs were detected with CAD*opti* separately in the 5–60 ms (spike-assembly) and 0.07–5.0 sec (rate-assembly) resolution window. CAD*opti* prunes redundant assemblies and selects those with the lowest *p* value in each resolution window (see Methods). Bars show weighted mean and SE pooled from all sessions (mean weighted by the number of units per session). While the activity of both spike- and rate-assemblies is theta modulated, spike-assemblies, in particular, recruit spikes that have a stronger phase-locking than the totality of spikes fired by the unit. Source data are provided as a Source Data file.

$1.91 \pm 0.07$ bit/s for rate-assemblies, general linear mixed-effects model of the spatial information of assemblies according to the assembly type, spike- vs rate-assembly, $F_{(1, 419)} = 70.55$, $p = 7.0 \times 10^{-16}$) which often appeared to be coordinated with the theta rhythm of the local field potential (Fig. 1d). To understand whether the observed coordination at short timescales was related specifically to replay events, we repeated the assembly detection but excluding epochs in correspondence of SWRs. As shown in Supplementary Fig. 4, the temporal resolution of the detected assemblies was conserved, showing that SWRs were not the prevailing source of fast coordination under these conditions.

## Spike-assemblies fire spikes phase-locked to the local theta rhythm

To quantify the relationships between assembly activations and the theta rhythm, we isolated the spikes fired by a unit while participating in assembly activations (assembly-spikes, in red in Fig. 2a) and compared their theta phase preference with the overall firing phase preference of that unit (in black in Fig. 2a). We found that both spike- and rate-assemblies were similarly phase modulated (Fig. 2b, generalized linear mixed-effects model of the probability of a unit phase-locking when firing within an assembly according to the assembly type, spike-vs rate-assembly; with binary dependent variable for significant phase

locking and logit link function: $F_{(1,1805)} = 0.15$, $p = 0.70$. The model accounts for rat identity and recording sessions as covariates. See Supplementary Fig. 5a for the same analysis excluding spikes fired during SWR), with about 70% of assembly-units phase-locked when active within an assembly configuration (all fractions presented here are computed after Benjamini–Hochberg correction for multiple comparisons, $\alpha = 0.05$, see Methods for Hodges–Ajne test on phase locking). Yet, of these, the fraction of units with a stronger phase modulation during assembly activity when compared to their overall activity was significantly higher for spike-assemblies (75%) than for rate-assemblies (30%) (Fig. 2c, d, generalized linear mixed-effects model of the probability of a unit increasing phase modulation when firing within an assembly according to the assembly type, spike- vs rate-assembly; with binary dependent variable for significant increases in phase modulation and logit link function: $F_{(1,1310)} = 134.15$, $p = 1.3 \times 10^{-29}$. The model accounts for rat identity and recording sessions as covariates. See Supplementary Fig. 5b for the same analysis, excluding spikes fired during SWR). Thus, while theta-modulated units contribute to both spike- and rate-assemblies, rate-assembly activations did not specifically coincide with temporal windows of high theta modulation. On the other hand, the higher temporal precision of spike-assemblies was associated with enhanced phase-locking of their contributing members when active within the assembly configuration.

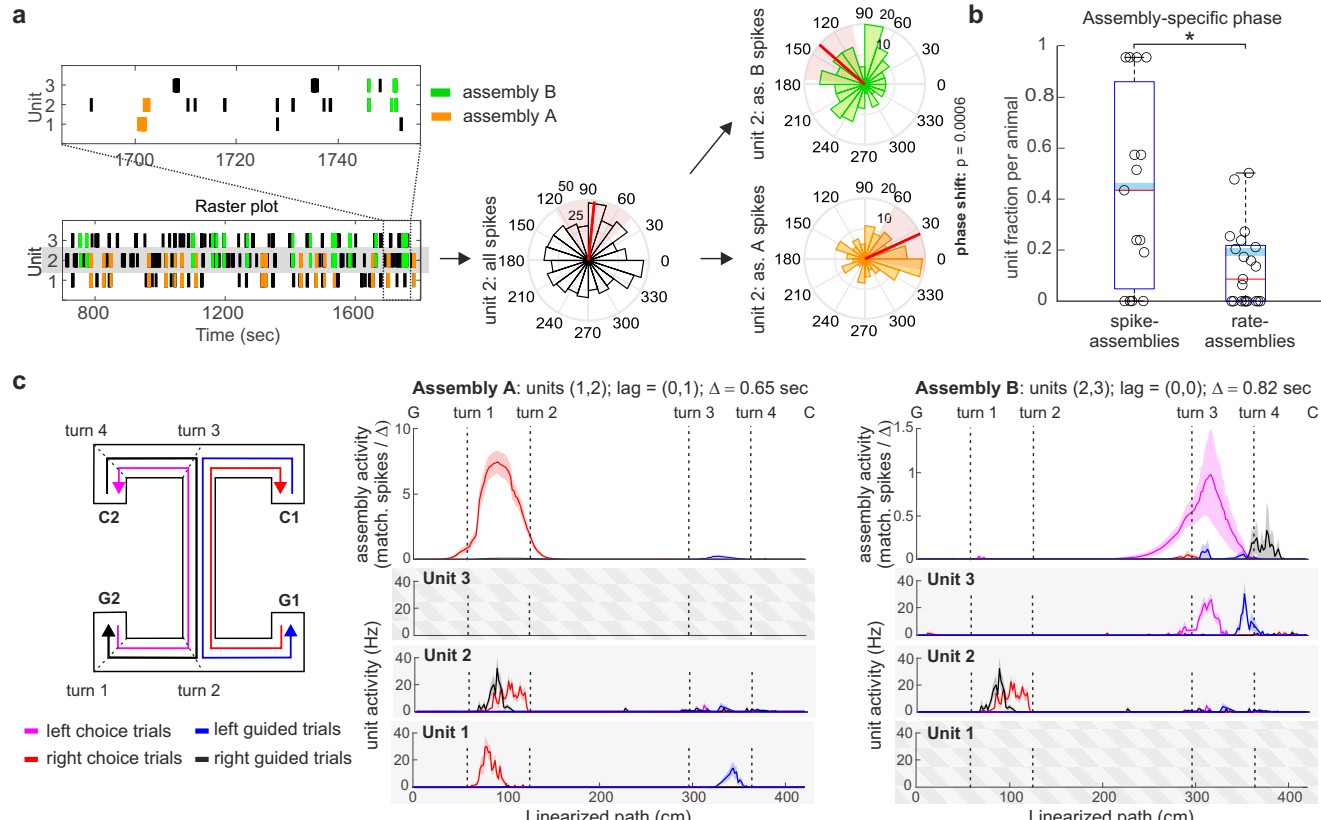

**Fig. 3 | Single units change phase preference when active in different assemblies. a** Raster plot and phase preference of two example assemblies, assembly A and assembly B, with a shared unit (unit 2). Unit 2 changes phase preference when active in the two different assembly configurations, Kruskal–Wallis two-sided nonparametric test for angular data, *p* value reported in figure with no multiple-comparison adjustments (see Methods); **b** Fraction of units that change their phase preference when active in different assemblies. Boxplots mark the median (red), the mean weighted by the number of tested assemblies per session (cyan), min and max point (whiskers), and the 25th and 75th percentiles (bottom and top edges of the box) computed across animals after Benjamini–Hochberg multiple-comparison correction (α = 0.05), data points correspond to distinct recording sessions (four sessions of six rats, sessions where no units met the inclusion criteria were excluded, *n* = 15 and 21 for first and second bar, respectively); generalized linear mixed-effects model with logit link function: *F*(1,231) = 12.54, *p* = 4.8 × 10⁻⁴. * marks significance; **c** Mean and SE (shaded area) activity along the maze of the two assemblies displayed in (a) (top) and their composing units during different trial types (bottom). The trial typologies displayed are: left (pink) and right (red) choice trials, and left (blue) and right (black) guided trials. The assembly and unit activity are plotted along the linearized path. Vertical dashed lines mark different task segments, along the path from C to G, as indicated in the maze scheme for the left choice trial. Assembly temporal resolution, composing units, and lag of activation of each unit with respect to the activation of the first assembly-unit are indicated in the figure title. Unit 2 takes part in both assemblies A and B, joining unit 1 and 3, respectively (the checkerboard covers the activity of units not part of the assembly). When firing in the two assemblies, the unit fires preferentially at two different firing phases (a) to encode different task-related information (left vs. right choice trials). See also Supplementary Fig. 6 for other examples of assembly-dependent phase modulation of CA1 units. Source data are provided as a Source Data file.

## Individual units change phase preference when active in different assemblies

As single units were often contributors to multiple assemblies (cf. Fig. 1c) and assemblies activated with a characteristic phase preference (cf. Fig. 2b), we wondered whether the phase preference of hippocampal units is an assembly-specific property rather than a unit-specific one. In other words, do hippocampal units change their phase preference when active in different assemblies? We found that among all units taking part in at least two phase-locked assemblies (*n* = 64 in spike-assemblies and *n* = 169 in rate-assemblies) an average of 27% (fraction computed after Benjamini–Hochberg correction for multiple comparisons, α = 0.05) changed their phase preference when active in different assemblies of the same temporal resolution (Fig. 3a, b, see Methods for nonparametric test on equality of median phase). This relative *phase-shift* was found both in spike- and rate-assemblies, with a higher proportion of units with significant phase-shift in spike-assemblies (Fig. 3b, generalized linear mixed-effects model of the probability of a unit to phase-shift when firing in different assemblies according to the assembly type, spike- vs rate-assembly; with a binary dependent variable for significant phase shift and logit link function:

*F*(1,231) = 12.54, *p* = 4.8 × 10⁻⁴. The model accounts for rat identity and recording sessions as covariates. See Supplementary Fig. 5c for the same analysis, excluding spikes fired during SWR).

### Phase coding of hippocampal assemblies

To investigate whether such shifts in phase encoded spatial or contextual information, we focused the analysis on those units changing phase when active in different contexts, comparing their activation during task epochs corresponding to different locations and cognitive demands on the end-to-end T-maze. Fig. 3c and Supplementary Fig. 6 show the activation along the maze of some typical units and the assemblies they joined. While single units fired in multiple locations, assembly activations were more selective (resulting in average spatial information of 2.25 ± 0.06 bit/s when compared with 1.52 ± 0.05 bit/s for single putative place cells), typically signaling one of the place fields of their constituent units and/or only a particular run type or direction.

This enhanced selectivity suggests that theta phase coding in hippocampal units extends beyond phase precession: while phase precession relative to the ongoing theta oscillation correlates with the

distance covered by the animal within a unit's place field, here we observed a phase preference disambiguating the activation of different assemblies in disjoined locations. We therefore hypothesized that theta phase coding of hippocampal units is not limited to the animal's position within a place field, but is also associated with different locations or contexts in the environment.

## The theta-firing phase of place cells can discriminate between distinct place fields of the same unit

A possible confound for the presence of contextual phase-shift coding in different assemblies comes from the fact that the spikes fired within an assembly might not uniformly sample the place field of a unit. Thus, if two assemblies systematically sampled the initial and final part of a phase-precessing unit's place field, respectively, this could result in an assembly-specific change in phase preference. To rule out this possibility, we analyzed the phase preference of single units, this time separating their spikes according to their own place fields instead of by assembly membership. In single units with multiple place fields, different rate-assemblies often activated in correspondence to different place fields of the unit (e.g., cf. Fig. 3c and Supplementary Fig. 6). As single units changed their phase preference when active in different rate-assemblies (cf. Fig. 3b), separating unit spikes by place field could reveal similar shifts in phase to those observed when separating them by rate-assembly.

Accordingly, we selected all units with multiple place fields and clustered their spikes according to their firing location (Fig. 4a, see Methods). We included in the analysis all recorded putative place cells, and not only those detected as participating in assemblies, as we assume that all CA1 place cells do participate in some assembly, but that sparse electrophysiological sampling of the CA1 population precludes the detection of all potential assemblies. In our dataset, putative place cells had more than one place field with a median of four fields per unit. Such a high number of place fields per unit was due to the relatively large size of the maze[43] and, critically, to its compartmentalization in clearly identifiable segments. Comparing the phase preference with respect to place field location, we found that 43% of the tested units changed firing phase when active in different place fields (value obtained as mean across sessions weighted by the number of units tested per session, 208 units among 24 sessions. Fraction computed after Benjamini–Hochberg correction for multiple comparisons, $\alpha = 0.05$, Figs. 4b, 5b). Note that for this test, we included all spikes fired within a place field and compared them with all spikes fired in another place field of the same unit. Thus, the observed phase shift cannot be explained by a biased sampling of different subregions of the place fields.

## The firing phase of place cells can encode distinct task-related information within the same place field

Beyond encoding spatial information, the hippocampus has been shown to carry information about episodic memories[44,45], sequences[46,47], and abstract relations[48,49]. Thus, hippocampal assemblies may encode task-relevant information beyond purely spatial parameters. We can, therefore, expect the recruitment of different assemblies when the animal has to remember, for example, a left turn rather than a right turn, or has to perform a guided turn rather than a choice turn. At the single-unit level, this should be reflected by a change in a unit's phase preference for different trial types, even within the same location.

Separating unit spikes according to place field as in the previous analysis, we further divided the spikes according to the type of trial in which they occurred. Trials were divided into left and right choice runs, when the animal had to choose between left and right turn; and four different types of guided runs, in which the sequence of turns was predetermined by the experimenter (Fig. 5a). We found that 36% of units changed their phase preference when active in different trial

types above chance level, despite overlapping place-field locations (value obtained as mean across sessions weighted by the number of units tested per session, 160 units among 23 sessions. Fraction computed after Benjamini–Hochberg correction for multiple comparisons, $\alpha = 0.05$, Fig. 5b, see also Supplementary Fig. 8 for same analysis with place fields identified with different methods and Supplementary Fig. 9 for same analysis with spikes fired only in epochs of high theta-power and no SWR). In fact, separating a unit's spikes not just by place field but according to the task epoch during which they occurred resulted in narrower and more coherent phase distributions of a unit's spikes (Fig. 5c). This observation was corroborated by training a support vector machine (SVM) classifier on the phase of spikes fired in an individual place field to distinguish between trial types. We found that for 32% of units, at least two trial types could be distinguished above chance level within at least one of the unit place fields (see Methods for details). This result could not be explained by covariates such as the animal speed and the ongoing theta power. In fact, we found that adding phase information to a classifier built on the animal speed and on theta power increased its accuracy (Classifier 1: SVM on speed and theta power; Classifier 2: SVM on speed, theta power, and theta phase. General linear mixed-effects model of Classifier 1 and Classifier 2 accuracy, contrast tests between the two classifier types: $F(1,530) = 8.9$, $p = 0.003$. Test computed on the subset of place fields which could distinguish above chance trial identity by Classifier 2, more powerful and thus granting a larger sample size). The observed relation between the phase of firing and trial type could also not be explained by the sorting cluster quality of the units (generalized linear mixed-effects model of a unit to phase-shift, either in different place fields or within the same place field but in different trials, based on its L-ratio, recording session and animal; with binary dependent variable for significant phase change and logit link function: $F(1,60) = 0.12$, $p = 0.74$). Moreover, while it is known that splitter cells can differentiate between trial types by rate modulation, we found that adding information relative to the spike phase to the instantaneous firing rate further improved the decoding performance of the SVM (generalized linear mixed-effects model of the accuracy of an SVM classifier trained on the instantaneous firing rate of each spike or on instantaneous firing rate and phase of each spike. Contrast tests between the two classifier types: $F(1,684) = 5.7$, $p = 0.017$. Test computed on the subset of place fields which could distinguish above chance trial identity in the latter, more powerful, classifier).

Fig. 5c shows an example unit with phase-shift coding at the choice junction of the maze, distinguishing left and right turns in choice trials and right turns in different guided trial types. Interestingly, in this example, the degree of differentiation of the unit phase is maximal when the animal has to actively remember its previous path to inform its next turn, and is absent in the guided trials when no active choice has to be made, and the path covered from the trial onset is identical (with a difference of 2.8 rad between the average spiking phase of the left and right choice trials, and of 0.2 rad between the two guided trials types). When examining instances of phase-shift coding among trial types, we found that the majority of phase-shifts distinguished between left vs. right choice trials and choice vs. guided trials (Fig. 5d, generalized linear mixed-effects model of the probability of a unit to change firing phase within the same place field according to the trial type, with a binary dependent variable for significant phase change and logit link function: $F(4,641) = 10.2$, $p = 4.9 \cdot 10^{-8}$; contrast tests between specific condition/bars (I vs II) $F(1,641) = 0.2$, $p = 0.6$; (I vs III) $F(1,641) = 5.6$, $p = 1.8 \times 10^{-2}$; (I vs IV) $F(1,641) = 10.7$, $p = 1.2 \times 10^{-3}$; (II vs III) $F(1,641) = 9.8$, $p = 1.8 \times 10^{-3}$; (II vs IV) $F(1,641) = 16.4$, $p = 5.8 \times 10^{-5}$; (III vs IV) $F(1,641) = 1.9$, $p = 0.2$. The model accounts for rat identity and recording sessions as covariates). These results withstood using different methods to identify place fields (Supplementary Fig. 10) and removing spikes fired during epochs of low theta-power or during SWR (Supplementary Fig. 9). This further supports the link between phase-

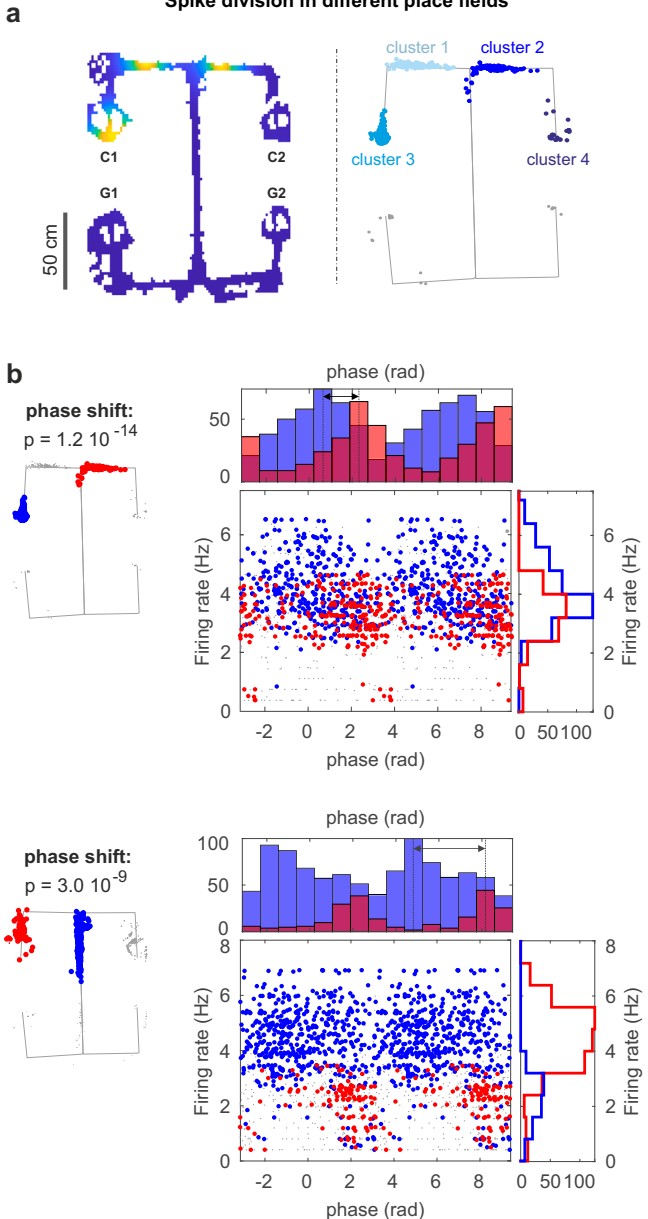

**Fig. 4 | The theta-firing phase of place cells can discriminate between distinct place fields of the same unit. a** Example of unsupervised spike clustering based on the spike position for a unit with multiple place fields. The number of place fields present in the spike set was established with DBSCAN, a density-based spatial clustering algorithm. The cluster memberships resulting from DBSCAN were then fed to a Gaussian mixture model for the final step of the classification (see Supplementary Fig. 7 for a comparison between this and other methods for place field detection). **b** Rate-phase and respective marginal distributions of unit spikes color-coded according to their field of firing. The exemplary units changed their firing phase when active in different place fields (two-tailed Kruskal–Wallis nonparametric test for angular data, see Methods, *p* value reported in figure with no multiple-comparison adjustments). Gray dots are spikes fired out of the two tested place fields.

shift and information encoding. The type of trial was, in fact, the most relevant information for performing the task correctly.

## Phase shift leads to context-dependent fine temporal coordination among units

The firing phase of hippocampal CA1 principal cells can, therefore, encode task-related information to differentiate distinct maze

locations or distinct mnemonic information within the same location. Such phase coding goes beyond what would be expected by phase precession. Nevertheless, phase-shifting and phase precession could share common underlying mechanisms. We observed that the population of units with phase-shift coding in at least one of their place fields correlated with the population of units with phase precession in at least one of their place fields (chi-square test for independence: $\chi^2(1, N = 213) = 7.3$, $p = 6.8\,10^{-3}$, $p$ value threshold for significance on the test for phase precession and phase-shift of 0.05, see Methods). Phase-shifting could occur between place fields with or without phase precession (Fig. 6a–c). When place-field-dependent phase-shift and phase precession co-occurred, the spike phases covered during the precession spanned different phase ranges in the two different fields (Fig. 6c).

Both experimental and theoretical studies have shown how a change in excitation received by a hippocampal unit can modify its phase of discharge[32–36,50]. This suggests a possible interpretation of the phase-shift phenomena: spatial exploration or cognitive tasks recruit specific and diverse cell (rate-)assemblies; each assembly is characterized by a set of synaptic connections that provides the assembly-units with a characteristic level of excitation whenever the assembly is activated upon a specific task event; this assembly-specific degree of depolarization, combined with an oscillatory somatic inhibition, would thereby generate a phase preference, or phase-range preference, typical and specific for the activity of the unit within that assembly (Fig. 6d). In line with this hypothesis, we found that differences in average phase preference between two sets of spikes also co-occurred with differences in the average instantaneous firing rate (see Methods for methodological details). This was true when comparing spikes from different place-field locations, within the same location but from different trial types, or spikes occurring as part of different spike- or rate-assemblies (chi-square test of independence on pairs of spike-sets with $p$ value > or <0.05 when testing phase and instantaneous firing rate differences - for different place fields: $\chi^2(1, N = 2165) = 28.9$, $p = 7.5\,10^{-8}$; same location different trial types: $\chi^2(1, N = 1004) = 20.5$, $p = 5.9\,10^{-6}$; spike-assemblies: $\chi^2(1, N = 318) = 4.9$, $p = 0.027$; rate-assemblies: $\chi^2(1, N = 5078) = 63.2$, $p = 2.0\,10^{-15}$).

Finally, as for phase precession, the fine coordination of neuron firing with the theta oscillation will generate, at the network level, a stereotypical sequence of unit activations[19]. In the case of phase-shift coding however, the sequence of active units during a theta cycle will be determined not only by the spatial proximity of their place fields, but will also be affected by the identity of the assembly recruited at the time. This implies that the lag between the activation of two units within the theta cycle, and consequently the sequence order and composition, could vary according to the cognitive demand. We investigated this hypothesis by testing if the lag of maximal cross-correlation between two units within a theta cycle changed in different trial types (Fig. 6e). To make sure that such a change was not due to occasional outliers but was consistent across trials, we selected the lag of maximal cross-correlation for each trial and compared the set of lags so obtained by trial type. We performed the analysis for each place field of each unit and found that changes in cross-correlation lags occurred with higher probability in place fields where units displayed phase-shift coding across trials (chi-square test of independence performed on place fields according to DBSCAN + GMM, $\chi^2(1, N = 11305) = 95.8$, $p = 0$; DBSCAN, $\chi^2(1, N = 11305) = 63.6$, $p = 1.6\,10^{-15}$; RM + GMM, $\chi^2(1, N = 8844) = 172.7$, $p = 0$; RM, $\chi^2(1, N = 9747) = 113.1$, $p = 0$; also confirmed when excluding spikes fired during SWR and low theta-power epochs: DBSCAN + GMM, $\chi^2(1, N = 11305) = 62.9$, $p = 2.2\,10^{-15}$; DBSCAN, $\chi^2(1, N = 11305) = 69.2$, $p = 1.1\,10^{-16}$; RM + GMM, $\chi^2(1, N = 8844) = 84.0$, $p = 0$; RM, $\chi^2(1, N = 9747) = 25.3$, $p = 4.8\,10^{-7}$).

To explore how the activation of different rate-assemblies can produce different activation sequences along the theta cycle even

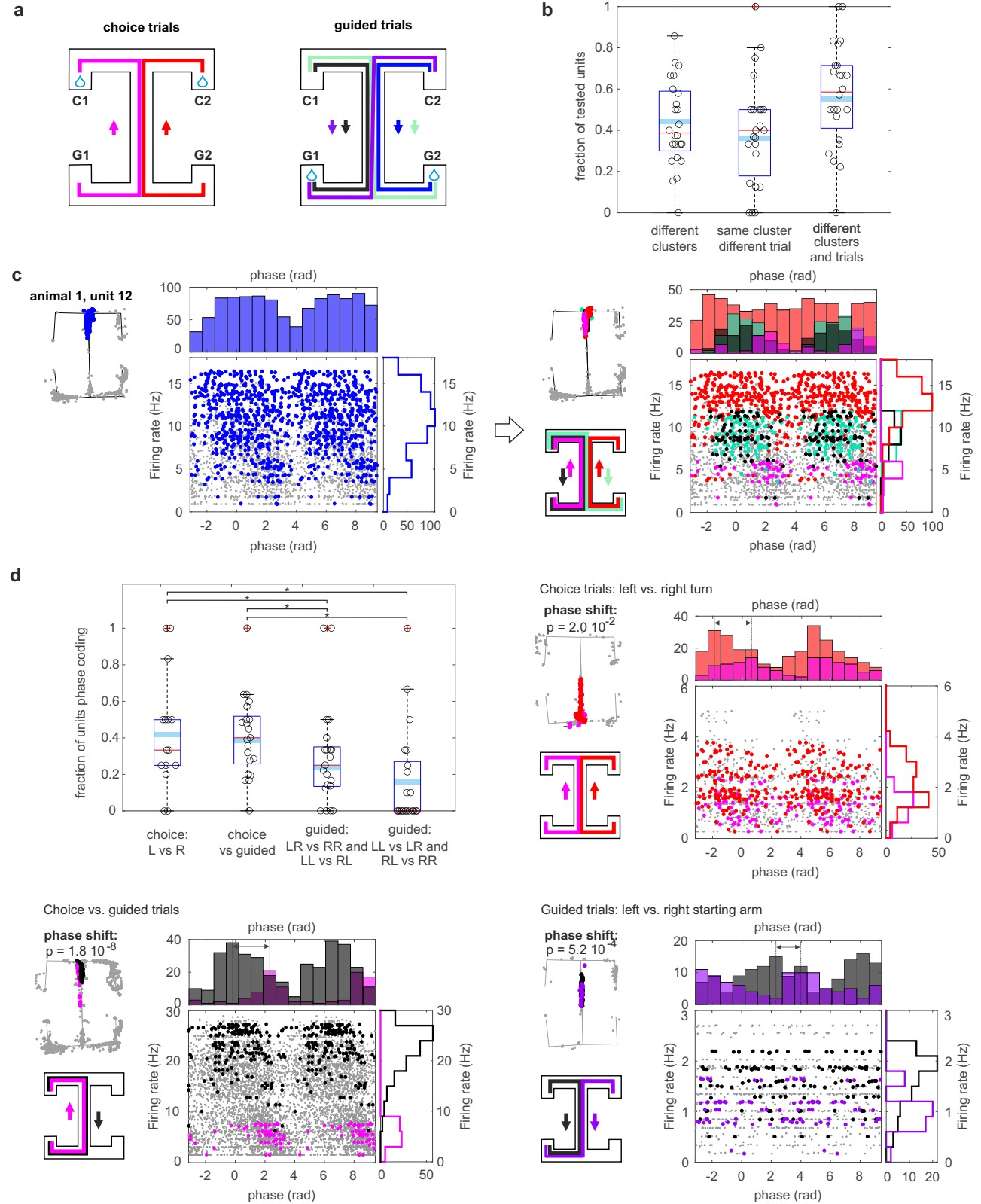

within the same set of units, we simulated the phenomenon with an adaptive exponential integrate-and-fire model (ref. 51; see Methods for formal description of the model). As observed in the experimental data, single units could participate in multiple assemblies (cf. Fig. 3) and respond at different, but context-consistent, average firing rates (cf. Fig. 5). We simulated three units taking part in two rate-assemblies. During the assembly activation, each unit received an assembly-

specific degree of depolarization. Assembly retrieval was modeled by the synchronous and transient depolarization of its constituent units, while inhibition was, on average, constant over time, and identical for all units and both assemblies. Similarly to soma-dendritic interference models[32,50], we captured the interplay between the oscillatory inputs to units by sinusoidally modulating both excitatory and inhibitory conductances with a relative offset of π rad (Fig. 6f top). As expected,

**Fig. 5 | The firing phase of place cells can encode distinct task-related information within the same place field. a** Trial categories: choice left (magenta), choice right (red), forced left (blue), forced right (black), forced switch right-left (light-yellow) forced switch left-right (dark-yellow); **b** fraction of units changing phase for different locations irrespective of the trial type, different trial types but same location, different trial type and/or location. See also Supplementary Figs. 8, 9; **c** joint rate-phase distribution and marginal distributions for spikes fired by a unit in one of its place fields (left) and the same spikes divided by trial type (right, color-coding as (a)). Gray dots are spikes fired out of the two tested place fields; **d** fraction of units changing phase in left vs. right choice trials; choice vs. forced trials; forced trials with different origin arms; forced trials with same origin arms but different forced turn. Generalized linear mixed-effects model with logit link function: $F_{(4,641)} = 10.2$, $p = 4.9 \times 10^{-8}$; contrasts (I vs II) $F_{(1,641)} = 0.2$, $p = 0.6$; (I vs III) $F_{(1,641)} = 5.6$, $p = 1.8 \times 10^{-2}$; (I vs IV) $F_{(1,641)} = 10.7$, $p = 1.2 \times 10^{-3}$; (II vs III)

$F_{(1,641)} = 9.8$, $p = 1.8 \times 10^{-3}$; (II vs IV) $F_{(1,641)} = 16.4$, $p = 5.8 \times 10^{-5}$; (III vs IV) $F_{(1,641)} = 1.9$, $p = 0.2$. * marks significance. See also Supplementary Figs. 10, 11. Displayed also example units changing phase preference for the same place field during different trial types (two-tailed Kruskal−Wallis nonparametric test for angular data, $p$ values reported with no multiple-comparison adjustments). In (**b**, **d**), boxplots mark the median (red), the mean weighted by the number of tested units per session (cyan), min and max point (whiskers), outliers (red crosses), and the 25th and 75th percentiles (box edges) computed across animals after Benjamini−Hochberg correction for multiple comparisons ($\alpha = 0.05$). Data points correspond to distinct recording sessions (four sessions of six rats, sessions where no units met the inclusion criteria were excluded. Sample $n = 24, 23, 24$ for the first to the third bar of (**b**); $n = 15, 21, 21, 21$ for the first to the fourth bar of (**d**)). Source data are provided as Source Data file.

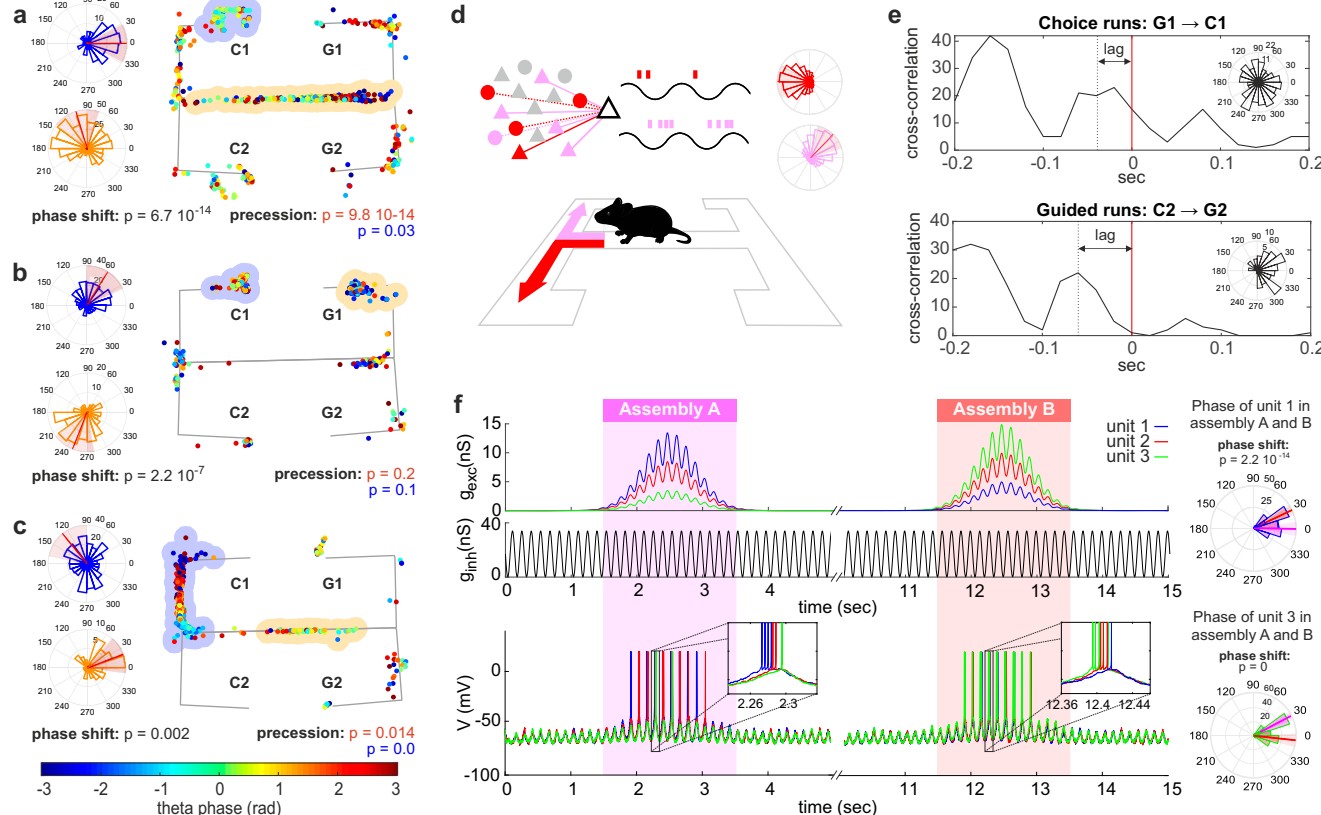

**Fig. 6 | Phase-shift leads to context-dependent fine temporal coordination among units. a**–**c** Spike-phase and spike-location (right) for three example units, along with phase histograms of the spikes fired within the place fields highlighted in blue and yellow (left). All three units show a change in the firing phase when active in different locations. Phase shift can occur between place fields with (**a**, **c**) and without (**b**) phase precession. Phase-shift test: two-tailed Kruskal−Wallis nonparametric test for angular data; phase precession test: two-sided test for circular-linear correlation[97] (see Methods), $p$ values reported without multiple-comparison adjustments. **d** Schematic of assembly-specific phase coding: Task performance requires the recall of past information and future goals. Different contextual conditions trigger the retrieval of distinct assemblies, even within the same location. The retrieval of each assembly provides different degrees of depolarization to their member units, thereby setting assembly-specific firing phases and phase-coding contextual information; **e** Cross-correlation between the spikes of two units, $w$ and $m$, during choice- and guided-right trials. Unit $w$ changes firing phase between choice and guided trials (two-tailed Kruskal−Wallis nonparametric test for angular

data, $p = 0.03$, phase histograms of $w$'s spikes in the two trial types shown as inset), thereby changing its relative lag of activation with unit $m$ within the theta cycles (two-tailed Wilcoxon rank-sum test, $p = 0.01$). Phase-shift coding can, therefore, produce context-dependent theta sequences. **f** Simulation of the recruitment of three neurons by two rate-assemblies, A and B. (Top) Excitatory ($g_{exc}$) and inhibitory ($g_{inh}$) conductances of three neurons (blue, red, and green) modeled with an adaptive exponential integrate-and-fire model. Rate-assembly activation is modeled with a transient assembly- and unit-specific increase in excitatory conductances. (Bottom) Membrane potential $V$ of the simulated units during assembly retrieval. (Right) Phase histogram of spikes of units 1 (blue) and 3 (green) when active in assemblies A (magenta) and B (red). Spike's phases are computed with respect to the oscillatory modulation of $g_{exc}$. The assembly-specific depolarization alters the preferred firing phase of the simulated units (right, two-sided Kruskal−Wallis nonparametric test for angular data, $p$ values reported with no multiple-comparison adjustments), resulting in a population-level shift in the order of unit activation along the theta cycle (inset).

assembly activations produced a broad increase in unit firing rate punctuated by faster temporal coordination with the theta oscillation (Fig. 6f bottom). This coordination was assembly-dependent and the simulated units changed their phase preference of firing, here computed with respect to the timecourse of the excitatory depolarization, when active in the two assemblies (Fig. 6f right). To formally detect multi-unit activity patterns generated by the assembly activations, we simulated multiple retrievals of the two assemblies and analyzed the obtained spike trains with the CAD*opti* assembly detection algorithm, as previously done with our experimental dataset (see Methods and Supplementary Table 1 for simulation details and CAD*opti* parameters). The activation of both assemblies produced, at a fine temporal scale, sequential activity patterns with a mean lag of $0.06 \pm 0.005$ sec and $0.07 \pm 0.005$ sec of the second and the third unit from the first active. Despite the similarity in pattern structure, the two assemblies triggered a different activation order of their constituent units. In 98% of the simulations ($n = 100$ simulations performed with different noise realizations), the detected activation sequence reflected the assembly-specific degree of depolarization provided to each unit (Fig. 6f inset). More generally, units that received the highest depolarization activated first within the theta cycle, while, importantly, units receiving just a light depolarization terminated the activation sequence.

Overall, these results highlight how systematic shifts in theta phase preference not only broaden the range of information encoded by the hippocampal temporal coding but also contribute to the generation of context-specific theta sequences.

## Discussion

This study extends analyses of phase coding by CA1 place cell assemblies to a spatial working memory and decision-making task on a complex maze. We detected place cell assemblies using an unsupervised algorithm able to extract coordinated activity from the data without pre-defining timescales of interest. This corroborated extensive evidence that hippocampal coding is characterized by two predominant timescales: a rate scale (rate-assemblies) reflecting place field firing rate modulation, and a sub-second temporal scale (spike-assemblies) compatible with the entrainment of spikes by theta rhythms and during SWR-associated replay events. The relatively broad timescale characteristic of rate-assemblies suggests that their coordination is not solely imposed by the shared modulation of their composing units by the local theta rhythm, in agreement with their robustness to degradation of cholinergic signaling[52]. Nevertheless, spikes fired within either spike- or rate-assemblies coordinated with the theta rhythm, in line with previous literature reporting theta phase-locking of CA1 neurons when active in an assembly configuration[53–55]. We show that such theta locking is most pronounced for spike-assemblies, but present in rate-assemblies as well. A possible explanation for this enhanced phase locking is that isolating spikes fired within an assembly configuration effectively separates a specific mode from the otherwise multimodal phase distribution of the unit firing. In fact, the enhanced phase modulation of hippocampal units during assembly activations also revealed that units taking part in multiple assemblies changed their preferred spiking phase according to the assembly active at the time. This shift often coincided with a change in the location of the place field of activation of the unit when active in one or the other assembly. The observed higher specificity of information coding by assemblies with respect to that of the participating units, therefore, agreed with the higher specificity in phase preference exhibited by single units when active as part of the assembly. This was true both for rate- and, more frequently, spike-assemblies, and was not induced by SWR-associated replay events.

The enhanced phase modulation of hippocampal units during assembly activations also revealed that units taking part in multiple assemblies changed their preferred spiking phase according to the assembly activity at the time. This modulation, not attributable to replay events, was validated at the single-unit level by grouping spikes by the place fields in which they occurred, rather than by assemblies. In the dorsal CA1's deep sublayers, place cells can exhibit dual theta-phase firing preferences as the animal crosses the cell's place field[28,56,57]. Here, we show that changes in the preferred firing phase can distinguish between distinct place fields or between different visits to overlapping locations on alternative routes, particularly under conditions that require active use of spatial memory and/or decision-making. These changes were fast and reversible, in line with the hypothesis that they were generated by the transient activation of different cell assemblies.

Our findings may reflect mechanisms related to those driving theta phase precession in CA1 place cells. Although unanimous agreement about those mechanisms is yet to be reached, models fall into three broad categories: interference between oscillations of the somatic membrane potential and of dendritic potentials at a slightly higher frequency[18,58]; progressive dendritic depolarization coupled with somatic oscillatory inhibition discharge[32–35,50]; and patterns of synaptic transmission delays[59]. To test these various hypotheses, numerous experimental efforts have been made to elucidate the relationship between cell depolarization and the spiking phase. Spatially uniform inhibitory conductance has been shown to enhance the range of phase precession[36]. It has been observed that while the animal moves toward the center of a cell's place field, the rate and phase of spiking strongly correlate[32,35]. This correlation is, however, lost as the rate peak is passed and the animal leaves the cell's place field[60,61]. In vivo whole-cell recordings have shown that, during phase precession, the baseline membrane potential of CA1 pyramidal neurons undergoes a ramp-like depolarization[62]. In vitro, whole-cell patch-clamp recordings from dendrites and somata showed that an increase in dendritic excitation, coupled with phasic somatic inhibition, causes an increase in the neuron's firing and the advancement of the spiking phase with respect to the somatic modulation[34,50]. Similar results were observed when progressively depolarizing the membrane potential of hippocampal cells in anesthetized animals[33].

In line with this evidence that changes in depolarization lead to changes in the discharge phase, we found in our data that the changes in phase preference of individual units also co-occurred with changes in the instantaneous firing rate. While a recent study has shown that anatomically distinct place cell subpopulations in superficial and deep sublayers of dorsal CA1 pyramidal cell layer are biased towards rate and phase coding of spatial information, respectively, as the richness of sensory cues in the local environment is experimentally manipulated[63], our results suggest an integration of rate and phase coding within a population to support coding of complex information. This was also replicated through a model of adaptive exponential integrate-and-fire units similar in spirit to the soma-dendritic interference models (refs. 32,50; c.f. Fig. 1e). The model confirmed that the activation of an assembly imposed an assembly-specific rate and phase preference on each assembly unit, thus producing, at the population level, assembly-specific theta sequences. Thus, while correlation between rate and phase changes has been observed during rate remapping[64], our findings demonstrate that phase-shift coding extends beyond rate remapping and occurs also between distinct place fields or assemblies, frequently coinciding with changes in instantaneous firing rate, similar to those observed for splitter cells[5–7]. The model predictions are also in line with recent work showing that individual place cells quickly switch between the encoding of alternative future locations or heading directions on alternate theta cycles, with lower firing rates for non-preferred directions occurring in later theta phases[27]. The mechanism highlighted by our analysis and model is not the sole modulator of the firing of CA1 cells, as the timing of specific inputs from Schaffer collaterals and the perforant path also plays a role in their phase of discharge[56,65]. Nevertheless, it captures the

role played by various depolarization settings in inducing a modulation and discretization of phase preference of hippocampal CA1 place cells. We, therefore, propose that phase-shift coding may be a consequence of the different levels of depolarization generated by the specific constellations of synaptic input resulting from the activation of different assemblies as the animal encounters different cognitive or environmental contexts (cf. Fig. 6d).

The fine temporal coordination of unit activities imposed by phase precession and theta sequences is commensurate with the induction of plasticity mechanisms for binding episodic information that, otherwise, would be separated by seconds[66]. For this reason, the processes have been proposed as a network mechanism for episodic sequence learning[1,19,35,67–70]. In support of this hypothesis, degradation of the temporal coordination of hippocampal units with the theta rhythm, e.g., by the administration of the cannabinoid receptor agonist[71] or by muscimol injection into the medial septum[72], led to reduced performance in memory tasks despite leaving place-field representations intact. Degraded phase precession caused by the passive transportation of rats during spatial exploration also drastically reduced replay during subsequent sleep[73], potentially reflecting impaired memory consolidation[74–76]. The changes in assembly-specific phase preference of hippocampal units reported here allow rapid reconfiguration of different theta sequences within the same neural population, differentiating distinct maze locations or distinct mnemonic information. This could contribute to the formation of context-dependent theta sequences[77], supporting the formation of episodic memories and planning[27].

Goal-dependent theta sequences have been observed during decision-making tasks, where the theta sequences terminated with the activation of cells encoding distant goal locations[26]. The process by which goal-related theta sequences form is still unclear, as is the causal relation between phase precession and theta sequences (see ref. 78 for review). One hypothesis is that during the early stages of learning, inter-regional assemblies (e.g. prefrontal-hippocampal assemblies[77,79–83] or medial-septum-hippocampal assemblies[84]), recruited at each theta cycle[85], modulate the depolarization of hippocampal units thereby dynamically producing goal-dependent cycling activity. In particular, as suggested by our theoretical model (cf. Fig. 6f), a low-level generalized depolarization of the cells encoding the current goal location could explain their spiking at the end of theta sequences, even when the animal is far from that location[26]. Similarly, the cognitive segmentation of a task induced by the presence of landmarks and corners within a maze, could induce a shared enhanced depolarization to all the cells involved in the same cognitive segment, also in those with place fields far from the animal but within the traveled maze segment. This could give rise to the observed space chucking of hippocampal theta sequences[22,26]. Finally, similar mechanisms as observed here for assembly-dependent phase preference of CA1 units may also play roles in organizing phase preferences of neurons across other brain regions, consistent with evidence for phase precession in the dentate gyrus[86], CA3[86,87], entorhinal cortex[86,88], subiculum[89], ventral striatum[90] and in the medial prefrontal cortex[38]. Such distributed processing would, therefore, support the integration of spatial and temporal information into cognitive contexts at a timescale commensurate with rapid adaptive behaviors, dynamically aligning different hippocampal assemblies with different subsets of neocortical and subcortical neurons.

## Methods
### Animals and husbandry
All procedures were conducted in accordance with the UK Animals (Scientific Procedures) Act 1986 and approved by the University of Bristol Animal Welfare and Ethical Review Board. Six adult male Long Evans rats (9–16 weeks, 300–500 g, Harlan, UK) were used in this study. Prior to surgery, rats were group-housed on a 12/12 h light/dark cycle (lights on from 07:00–19:00) with free access to food and water. At least 1 week was allowed for animals to habituate to the new holding facility before surgery was performed. Post-surgery, animals were singly housed with additional bedding and cardboard tubes in high-roofed cages that allowed unconstrained head movement with cranial implants.

### Implantation of recording array
Custom-built adjustable tetrode (twisted 12.7-µm nichrome wire, Kanthal, gold-plated to 250–300 kΩ at 1 kHz) microdrives were implanted under isoflurane anesthesia using an aseptic technique and perioperative analgesia (Buprenorphine, 0.02 mg/kg s.c.). Craniotomies of diameter 1–1.5 mm were made over dorsal CA1 (AP -4.2 mm, ML 3.0 mm from Bregma). Implants were fixed to the skull using stainless steel screws (M1.4 × 2 mm, Newstar Fastenings) and Gentamicin bone cement (DePuy). Tetrode positions were adjusted over the course of 2–3 weeks after surgery.

Tetrode signals were amplified by headstages (HS-36, Neuralynx, MT, USA) and relayed via fine-wire tethers to a Digital Lynx system (Neuralynx), which sampled thresholded extracellular action potentials at 32 kHz (filtered at 600–6000 Hz) and continuous local field potentials (LFP) at 2 kHz (filtered at 0.1–475 Hz) using the Cheetah software package (Neuralynx) running on a desktop PC.

### Training
Once post-surgery body weight had stabilized, rats were placed on a regulated feeding regimen to maintain body weight at 85–90% of free-feeding levels. The rats were trained to perform a spatial memory-based decision-making task on an end-to-end T-maze, as described in ref. 37 and illustrated in Fig. 1a. Maze dimensions were 170 × 130 cm. Training occurred during the light phase at a similar time each day. Habituation: Rats were placed in the maze for 20–30 min without any boundaries in place. Rewards were provided at every visit to a reward zone. After 2 days, the rats advanced to the next stage. Guided trials: Rats ran a series of guided trials. Each trial consisted of a run from a reward point at one side of the maze, via the long central arm to a reward point at the opposite side. At the starting end of the maze, the opposite arm was blocked off with a barrier, guiding the rat onto the central arm. At the distal end of the central arm, a barrier blocking one of the arms (pseudorandomly selected) guided the rat to the end of the unobstructed arm, where a reward (0.1 ml of 20% sucrose solution in water) was delivered remotely through tubing connecting reward wells to syringe pumps located in the adjacent room, where the experimenter sat. Only one running trajectory was possible in each guided trial. In one training session, a rat was allowed to perform up to 40 trials. After a minimum of 2 days of at least 20 trials, rats advanced to the next training stage. Full Task: Rats performed a series of guided trials, interleaved with choice trials. Choice trials differed from guided trials in that there was no barrier in place at the far end of the central arm, requiring the rat to choose a turn direction. The correct turn direction was the same direction that the rat had initially turned when entering the central arm. If the rat chose correctly, a reward was delivered at the end of the arm. If the rat chose incorrectly, it was placed back at the start and allowed to undertake the trial again, until the correct choice was made. All guided trials began at the C end of the maze and ended at the G end, while the interleaved choice trials ran in the opposite direction. Rats were allowed to perform up to 40/40 guided/choice trials. Learning of the task rule was assessed by the percentage of correct choices made (>70% correct trials over at least 3 consecutive days). In this manuscript, we analyzed recordings from four days (for a total of 24 sessions) after the performance criterion was reached.

### Histology
At the end of each experiment, the rat was deeply anesthetized with intraperitoneal sodium pentobarbital, and a small electrolytic lesion

was made at the tip of each tetrode (positive current of 0.3 mA for 10 s). After lesions had been made on each tetrode, the rat was perfused transcardially with 0.9% saline and then 4% paraformaldehyde/0.9% saline solution. The brain was post-fixed, transferred to a cold 30% sucrose solution for cryoprotection, and cut into 50 µm coronal sections on a freezing microtome. Lesion locations were compared against the corresponding sections in the Rat Brain Atlas[91] in order to determine the tetrode recording sites.

## Spike sorting and cell selection

Spikes were sorted semi-automatically on the basis of waveform characteristics (waveform energy and first principal component) using KlustaKwik (K.D. Harris, http://klustakwik.sourceforge.net/), followed by manual refinement of cluster boundaries with the MClust package for Matlab (A.D. Redish, http://redishlab.neuroscience.umn.edu). After clustering, only units with a mean spike peak amplitude of >50 µV, isolation distances of ≥15[92], and <1% of interspike intervals (ISIs) below 2 ms were retained for further analysis. Our analysis focused on putative place cells. To select putative place cells we restrict to units with firing rate between 0.2 and 4 Hz and with spatial information above 0.5 bit/s on the maze[39]. Because of the dependence introduced in the data by pooling units from multiple sessions of the same animals, when appropriate we perform statistical tests by generalized linear mixed-effects models, where we explicitly account for session and rat identity.

## Cell assembly detection

Cell assemblies were identified with the unsupervised machine-learning algorithm for cell assembly detection (CAD)[40] (algorithm available at https://github.com/DurstewitzLab/Cell-Assembly-Detection). CAD detects recurrent activity patterns of arbitrary structure and temporal precision in multivariate time series. The algorithm is based on a recursive agglomeration scheme, at each step of which it detects and tests assemblies of progressively larger size. As the first step of the algorithm, CAD scans the activity of all possible pairs of units in the recorded set to select the most common shared (potentially lagged) activation pattern, and test if this reoccurs more frequently than expected by chance. If the test is significant, a new time series is created to reflect the activation of the detected assembly-pattern. In the second step of the algorithm, the new assembly activation time series are tested in turn including the remaining recorded units. If the test is significant, the unit becomes part of the assembly, and a new assembly activation time series is generated. The algorithm stops when no more units can be added to the assemblies detected in the previous agglomeration step (see ref. 40 for a more detailed description of the algorithm). It follows that the hypothetical inclusion of supplementary units beyond those actually recorded would affect the number and size of the detected assemblies, but it would not alter the coordination patterns already detected in the existing set. In this manuscript, the detection of paired assemblies was performed, stopping the agglomeration at the initial pairwise step, while full-size assemblies were detected letting the algorithm agglomerate until completion.

To uncover the temporal scales most represented in the hippocampal spike trains, we ran CAD on a broad spectrum of temporal resolutions sampled with a logarithmic scale in the interval [0.005–5.0] sec. CAD tests multiple temporal resolutions, and if the same sets of units coordinate at multiple timescales, the algorithm will return all of them. This analysis revealed the presence of two distinct timescales: one between 0.005 and 0.06 sec (spike-assemblies) and a second between 0.07 and 5.0 sec (rate-assemblies). To compare the extent of phase-locking and phase shift-coding within the two assembly groups, we repeated the assembly detection separately for the two time windows using CAD*opti*[93,94] (algorithm available at https://github.com/DurstewitzLab/CADopti). After testing multiple temporal

resolutions, CAD*opti* selects and returns the timescale at which each assembly has been detected with the lowest *p* value. Thus, each assembly was unique within each window but could be detected in both time windows. This pruning procedure allowed a fair comparison between the two timescales, without the distortion given by considering as independent assemblies the same set of units detected at neighboring temporal resolutions. Finally, we want to note that the detected assemblies cannot result from the detection of spike sorting mistakes. In such a case, in fact, assemblies would have been detected at the highest temporal precision (binning of 0.0058 sec), which is not the case, as shown in Fig. 1b.

For all the analysis of this manuscript the reference lag was set at 2. Tested bin sizes were [0.0058, 0.007, 0.009, 0.011, 0.014, 0.018, 0.022, 0.028, 0.035, 0.044, 0.055, 0.07, 0.09, 0.11, 0.14, 0.17, 0.21, 0.27, 0.33, 0.42, 0.52, 0.65, 0.82, 1.0, 1.3, 1.6, 2.0, 2.5, 3.2, 4.0, 5.0] sec and respective maximal lag: [4, 5, 5, 5, 5, 4, 4, 4, 3, 3, 3, 2, 2, 2, 2, 2, 1, 1, 1, 1, 1, 1, 1, 1, 1, 1, 1, 1, 1, 1, 1]. Values for tested bin sizes were selected based on previous experience in assembly analysis[40,93,94]. The lower limit was chosen because in CA1 we found practically no assemblies at coordination precision higher than 5 ms[40]. The upper limit was chosen to cover in about one bin the time needed by the rat to cross a place field.

## Computation of chance level for the probability of spike-assemblies being detected as rate-assemblies as well

Spike-assembly-pairs had a probability of $p = 0.9$ to be also detected as rate-assembly-pairs. We tested if the obtained value is above chance by bootstrap. To this aim, we considered all possible pairs of units present in the recorded sessions and randomly selected an amount equal to the number of the detected rate-assembly. We then computed the probability of the detected spike-assembly-pairs to be part of this subset $p_{boot}^i$. We repeated the random sampling $10^5$ times and averaged the result to obtain the chance level $\langle p_{boot} \rangle = 0.6$. The *p* value corresponds to the fraction of bootstrap sampling with $p < p_{boot}^i$, which, in the specific bootstrap set, never occurred.

## Instantaneous firing rate

Instantaneous firing rates were computed by convolving unit spikes with a Gaussian kernel with a kernel size of half of the unit's mean interspike-interval.

## Phase extraction and theta power

As the first step, we made sure that the spectrogram of all recorded LFPs peaked in the theta frequency band. Then, to obtain the phase of spikes, we bandpass filtered the LFP between 4 and 10 Hz (LFP$_\theta$) and computed the angle of the Hilbert transformation of LFP$_\theta$ at the time of each spike.

Since recent studies have shown that phase coding can also occur in the presence of a lower power or irregular amplitude of the theta oscillation[95,96], the analyses reported in the main figures of the manuscript are performed with maximal sample size, including all spikes without restriction on theta power. However, to make sure that the reported results were not affected by this choice, we reproduced the most important including only spikes fired in epochs of high theta power (Supplementary Figs. 9, 11). High theta power epochs were defined as periods in which the envelope of the LFP$_\theta$ amplitude surpassed one $\sigma$(LFP$_\theta$). High amplitude, saturating movement artifacts were removed from LFP by excluding periods with LFP$_\theta > 2*\sigma$(LFP$_\theta$). This was done before computing the threshold of $\sigma$(LFP$_\theta$) used to define high power periods.

## Detection of SWR

SWR detection was based on the Sleepwalker Matlab toolbox (https://gitlab.com/ubartsch/sleepwalker). In brief, LFPs were down-sampled to 1000 and 50 Hz notch filtered prior to bandpass filtering (using least squares filters) between 120–220 Hz. Candidate ripple events were

identified based on threshold crossings in the z-scored 120–220 Hz power, using a threshold of 3.5 x SD of the signal. The start and finish times of the ripple were calculated based on 2 x SD of 120–220 Hz power. Representative samples of individual and averaged events were then visually inspected. Ripples were rejected if they were shorter than 50 ms or longer than 500 ms; if gaps of less than 50 ms occurred between events, they were treated as a single ripple. Ripple start/end timestamps were used to exclude SWR-associated spiking for some analyses (Supplementary Figs. 4, 5, 9, 11).

### Phase-locking test
Phase modulation of spike sets was tested with the Hodges–Ajne test for non-uniformity[97,98] that, unlike the more common Rayleigh test, does not assume unimodality in deviation from uniformity. The test was performed on sets bigger than 50 samples, limit imposed by the approximations performed in the test. Significance was established with an alpha value of 0.05 and Benjamini–Hochberg correction for multiple comparisons on all tests performed.

### Phase-locking of assembly-spikes vs. unit-spikes
This test compares the strength of phase-locking of two sets of phase values. In particular, we tested via bootstrap if assembly activations elicited spikes with higher phase-locking than those overall fired by the unit. For each unit, we collected the phase of the $n$ spikes fired in correspondence with all activations of one assembly and produced 1000 replica sets composed of the phase of $n$ spikes randomly selected among all spikes of the same unit. For each set, we computed the length $R$ of the mean resultant vector. $P$ values were established by counting the fraction of replica sets with $R_{rep.} > R_{orig.}$. Significance was established with an alpha value of 0.05 and Benjamini–Hochberg correction for multiple comparisons on all tests performed.

### Change in phase preference for spikes fired in different assemblies
This test assesses if two sets of phase values have the same median phase. For each unit taking part in multiple assemblies, we divided into separate sets the spikes fired in different assembly-pairs. Sets not phase-locked to the ongoing theta oscillation or with less than 50 spikes (a limitation imposed to perform the phase-locking test) were discarded. To test the null hypothesis that for the same unit, each phase-set had an equal median, we performed a multi-sample non-parametric test, circular analog to the Kruskal–Wallis test[97,99]. Significance was established with an alpha value of 0.05 and Benjamini–Hochberg correction for multiple comparisons on all tests performed. Finally, we tested whether the number of units recorded in each session correlated with the fraction of units per session changing phase when firing in different assemblies. We found no significant correlation (Spearman's correlation $r_s(34) = 0.25, p-\text{value} = 0.14$).

### Trial categories
We divided trials into 6 categories according to the specific task required by each trial. Trials were first divided into choice trials, when the animal had to choose if to turn left or right on the basis of its position at the beginning of the trial, and guided trials, when turns were forced by the set-up. Both categories were then further divided according to the two turns performed entering and leaving the central arm of the maze: left-left/left-right/right-left/right-right.

### Isolation of place fields
Place fields were established only for units with spatial information above 0.5 bit/s (place cells) and with phase-modulated spikes. Spikes of each unit were divided into different clusters (place fields) on the basis of their place of firing. Place fields are often identified as a region of connected bins in a unit's rate map that surpasses a fixed threshold in firing rate. For example, for a maze comparable to the one used in

this manuscript, ref. 100 uses a threshold of a standard deviation over the mean firing rate of the unit. In units with multiple place fields, such a high threshold leads to a selection among the fields, at the expense of those with lower rates which remain undetected (c.f. Supplementary Fig. 7). While this is typically not an issue, this study aims to investigate the changes in phase and rate a unit exhibits across different place fields and trial types. Therefore, it is here important to capture a larger variety of fields. We thus explored different techniques to detect place fields and tested the robustness of the analyses in Fig. 5b, d.

The compared techniques were:

**Rate map Wirtshafter and Wilson (RM WW).** In ref. 100 place fields are detected by: (1) computing a rate map of firing per occupancy binned with a 2 cm grid and smoothed with a 10 cm standard deviation Gaussian kernel. Only epochs in which the rat moved faster than 12 cm/s were included; (2) thresholding the rate map at $\theta^1_{RMWW} = \mu(fr) + \sigma(fr)$, with $\mu(fr)$ and $\sigma(fr)$ rate mean and standard deviation, respectively; (3) only fields with at least one bin with rate above $\theta^2_{RMWW} = \mu(fr) + 2 \cdot \sigma(fr)$ were selected; (4) fields of length less than 15 cm were discarded.

**Rate map (RM).** To detect also place fields with lower firing rates we repeated the procedure described in RM WW but using a $\theta^1_{RM} = 0.7 \cdot \mu(fr)$ and omitting point 3.

We also included methods that aimed to group spikes according to the density of their spatial clustering:

**Rate map + GMM (RM + GMM).** We modeled the place fields of a unit with a Gaussian Mixture model. In the first step, we proceeded as described in RM but with a harsher threshold of $\theta^1_{RMGMM} = \mu(fr)$. This allowed us to establish the overall number of place fields and obtain a first estimation of cluster memberships. In the second step, this first clustering was then used as an initial condition for the estimation of a Gaussian mixture model (function fitgmdist, Matlab). To train the model, we only used the spikes identified as place field members in the first step. Once obtained the model, we used it to cluster (function cluster, Matlab) all spikes fired by the unit. Spikes with a low probability of being part of any of the identified clusters (i.e., with a logarithm of the estimated probability density function smaller than logpdf < − 20) were discarded as not assigned to any of the modeled Gaussians (place fields).

**DBSCAN.** Spikes were clustered with a density-based spatial clustering algorithm (DBSCAN[101], Matlab function *dbscan* with parameters $\varepsilon = 0.05$ and $MinPts = 15$). Spikes identified as outliers by the algorithm were discarded.

**DBSCAN + GMM.** We proceeded as in RM + GMM but used the output of DBSCAN as the initial place field estimate.

Common to all tested place field detection methods, only spikes fired when the rat moved faster than 12 cm/s were included.

Supplementary Fig. 7 shows a comparison between the outputs of the different place field detection methods. Key analyses were repeated on unit's activity parsed by place field computed with different detection methods (Supplementary Figs. 8, 9, 10, 11, and text).

### Classifier
For each place field, we trained a support vector machine (SVM) classifier with linear kernel (slack variables minimized with L1 norm and box constraint = 1) to divide trials according to their trial type (the trial types categories here considered were: correct choices left, correct choice right, forced left, forced right, forced switch right-left, forced switch left-right). For every two types of trials, we build three SVM models: one based on the spike's phase, one on the spike's instantaneous firing rate, and one on both phase and instantaneous firing rate. Spike phases are a circular quantity and cannot be used directly to train

the SVM. Thus, phase information was passed to the classifier as $[\cos(\theta_t), \sin(\theta_t)]$, where $\theta_t$ is the phase relative to the theta band of the LFP of the spike fired at time $t$. The classifier accuracy was computed with a 50-fold cross-validation, to avoid overfitting. Significance was established via bootstrap. Bootstrapped samples were created by shuffling the trial labels. Since this step removes not only the spike-label association but also any autocorrelation in the label time series (which might affect the accuracy when performing block cross-validation), for a fair comparison, we jointly shuffled the order of the spike-label elements when training the SVM classifier on the original set. The bootstrap procedure was repeated 500 times, and $p$ values were assigned by counting the fraction of bootstrap sets with an accuracy higher or equal to the original set.

### Phase precession units
We tested phase precession for each place field of each unit. Precession was assessed separately for each trial type by computing the circular-linear correlation[97,98] between the unit phase and the position of the animal along the linearized trials-specific path when the spike was fired.

### Changes in the lag of activation between units along theta cycles
To assess if pairs of units significantly changed their relative lag of activation during different trial types, we first computed the maximal cross-correlation lag of the two units in each trial. Cross-correlation was computed with a 0.02 sec binning and, to remain within a theta cycle, within the [−3, 3] bin window (in Fig. 6e, we chose a larger window exclusively for visualization purposes). Once obtained the maximal correlation lag per trial, we divided the trials according to their trial type and tested for a change in lag with a two-sided Wilcoxon signed-rank test. Since single units had different phase preferences in different place fields, testing was performed separately for each unit place field.

### AdEx model and assembly recruitment
Neuronal activity was simulated by an Adaptive Exponential Integrate-and-Fire model[51]. In AdEx models, the evolution of the neuron membrane potential $V$ and adaptation current $w$ is defined by the equations:

$$C\frac{dV}{dt} = -g_L(V - E_L) + g_L\Delta_T \exp\left(\frac{V - V_T}{\Delta_T}\right) \\ -g_e(V - E_e) - g_i(V - E_i) - w + \varepsilon \tag{1}$$

$$\tau_w\frac{dw}{dt} = a(V - E_L) - w \tag{2}$$

With membrane capacitance $C$, leak conductance $g_L$, threshold slope factor $\Delta_T$, resting potential $E_L$, threshold potential $V_T$, adaptation time constant $\tau_w$, and subthreshold adaptation $a$. Noise in the evolution of the membrane potential was introduced through the parameter $\varepsilon \sim N(0, 1.6\,10^{-19})$. Synaptic currents were modulated through the time-dependent excitatory and inhibitory conductances $g_e(t)$ and $g_i(t)$, and the excitatory and inhibitory current reversals $E_e$ and $E_i$, respectively. At every time $\bar{t}$ the membrane potential reached 0 mV, an action potential was fired and $V(\bar{t})$ was set to $V_s$. Afterward, both the membrane potential and the adaptation current were reset to $V(\bar{t}+1) = V_r$ and $w(\bar{t}+1) = w(\bar{t}) + b$, respectively.

To reflect the presence of theta oscillations, we modulated both excitatory and inhibitory conductances at $\theta = 7$ Hz with a relative offset of $\pi$ rad. The excitatory conductance was then further modulated by a Gaussian-shaped depolarization to mimic transient assembly activation

$$g_i(t) = k_i\left(\sin(2\pi\theta t - \pi) + 1\right) \tag{3}$$

$$g_e(t) = k_e\left(\sin(2\pi\theta t) + 4\right)\exp\left(\frac{-(t - c)^2}{2\,\sigma^2}\right). \tag{4}$$

with $c = 2.5\,s$ and $\sigma = 0.4\,s$ center and standard deviation of the Gaussian, respectively. The activation of an assembly provided to its composing unit $n$ an assembly-specific degree of depolarization $k_e^n$. We simulated 2 assemblies composed of 3 units. In the first assembly $k_e^1 = 2.7\,nS$, $k_e^2 = 1.7\,nS$ and $k_e^1 = 0.7\,nS$. In the second assembly $k_e^1 = 1.0\,nS$, $k_e^2 = 2.0\,nS$ and $k_e^1 = 3.0\,nS$. Average inhibitory conductance was set at $k_i = 17\,nS$ for all units and all assemblies.

To formally evaluate the fine temporal coordination of network units induced by the recruitment of different context-specific cell assemblies, we concatenated 400 retrievals of each of the two modeled assemblies and ran CADopti on the spike time series so obtained. We performed and analyzed 100 simulations generated with different noise realizations. CADopti parameters: reference lag = 2; bin sizes: [0.0058, 0.007, 0.009, 0.011, 0.014, 0.018, 0.022, 0.028, 0.035, 0.044, 0.055] sec; maximal lag: [18, 21, 22, 22, 22, 20, 18, 17, 15, 13, 11].

### Reporting summary

Further information on research design is available in the Nature Portfolio Reporting Summary linked to this article.

## Data availability

The hippocampal data generated in this study have been deposited in the OSF database under the identifier DOI 10.17605/OSF.IO/PESKV accessible at https://osf.io/peskv/[102]. Source data are provided with this paper.

## Code availability

Analyses were performed with Matlab2018a. Code for cell assembly detection at multiple timescales CAD and CADopti are available at https://github.com/DurstewitzLab/Cell-Assembly-Detection and https://github.com/DurstewitzLab/CADopti, respectively.

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

## Acknowledgements

We would like to thank Richard J. Gardner for spike sorting, Andreas Draguhn and Martin Both for their support and stimulating discussions, and Thomas McHugh and David Foster for commenting on the manuscript. The drawing of the rat in Fig. 6 has been modified from https://scidraw.io/. E.R. has been supported by #NEXTGENERATIONEU (NGEU) and funded by the Ministry of University and Research (MUR), National Recovery and Resilience Plan (NRRP), project MNESYS (PE0000006) – A Multiscale integrated approach to the study of the nervous system in health and disease (DN. 1553 11.10.2022), by the Boehringer Ingelheim Foundation grant "Complex Systems", and by the Ch. and H. Schaller Foundation. D.D. was supported by the Deutsche Forschungsgemeinschaft (DFG) within CRC-1134 (subproject D01) and through individual grant Du 354/10-1. Data acquisition was supported by BBSRC grant BB/G006687/1 awarded to M.W.J. and a Newton International Fellowship awarded to N.B., with further analyses supported by a Wellcome Senior Research Fellowship to M.W.J. (202810/Z/16/Z).

## Author contributions

Experimental design and electrophysiology, N.B., T.H., and M.W.J.; Analysis, visualization and interpretation, Writing—original draft, ER; Writing—review and editing, E.R., M.W.J., A.P.F.D., K.F., and D.D.; Resources, M.W.J., E.R., and D.D.; Funding acquisition, E.R., N.B., M.W.J., and D.D.

## Competing interests

The authors declare no competing interests.
