## [Peer Review File · Nature Communications]

Integration of rate and phase codes by hippocampal cell-assemblies supports flexible encoding of spatiotemporal contextREVIEWER COMMENTS

Reviewer #1 (Remarks to the Author):

In this experiment, CA1 neurons were recorded as rats ran a forced-choice double T-maze experiment in which the start location was experimentally controlled ("guided") and the choice depended on the start. It is thus a working memory spatial task that switches between guided and free-choice phases. The authors deployed a recently developed analytic method that looks for repeated patterns of co-firing among ensembles, or "assemblies," of neurons: the period of time over which the firing was analysed varied from 5ms to 5s. It was found that within this span, patterns of co-firing (activations of an assembly) clustered around either short times (~30ms) or long times (~3s). Most units belonged to assemblies active at both timescales, and there was a very high likelihood that neurons that co-fired at the short timescale would also co-fire on the long timescale. When the authors looked at the firing of units in relation to theta phase they found that phase preference changed between different place fields of the same unit, and also between different task states (guided vs. choice, left-turn runs vs. right-turn runs etc) and that cells that showed theta phase precession tended also to be the ones that showed this phase shift between fields. The authors suggest that the variation in phase relations could result in a plastic sorting process, driven by the level of input activation, in which units could be temporally shuffled within sequential assembly activations, as a function of task state: something that could be used to encode episodic components of a memory.

This is a well-designed and well-illustrated experiment and there are some very clever ideas in this paper, but I am concerned about the sample size and the possibility that the phenomena uncovered here could arise from sampling biases. The issues are as follows.

First, the sample size is rather small, being only 131 units across five animals, which makes these population-coding analyses slightly insecure. I didn't get a sense of how large the assemblies were nor how many there were, as data were pooled across animals and separated by session, such that two sessions from each animal contributed data. These were not analysed (as they should have been) with reference to the interdependence of data (that is, with a multi-level analysis).

Second, the unbiased assembly analysis is presented as uncovering two timescales as if this is a new finding, but we have known for decades that there are two timescales in CA1 activation, which occur in three contexts. These are: (i) short-timescale synchronous bursting activity that is often/usually accompanied by sharp-wave ripples in the LFP and is usually called "replay", (ii) theta sequences which occur in a single theta cycle when the animal is in a theta state (e.g., when running) and reflect the spike timing relationships in the active place cell population, and (iii) place field activation which also occurs during theta and reflects the overall time-averaged firing rate pattern of place cells as the rat passes through each field in succession. The bulk of the analyses in the paper did not separate these three contexts so it seems likely that the short-timescale assemblies are a mixture of replay events and theta sequences, and the long-timescale activations are place field traversals. This does not invalidate the phase findings but the paper does present, as a new result, something that is not really new, and it requires some effort and prior knowledge to uncover what is going on here.

The first really novel and potentially very interesting observation is that neurons changed their phase preference when active in different assemblies of the same type (short-timescale vs long-timescale). However I am troubled by a potential confound here which arises from the fact that the allocation of a spike to a given assembly was determined by the timing of the other neurons in that assembly. It is possible therefore that one assembly tends to associate with all the spikes at the beginning of that neuron's place field and another at the end, due to the relative locations of the place fields of that assembly's other neurons, and thus that one assembly tends to link to the beginning of the phase precession cycle and the other at the end. One can actually see this in the example in Fig. 3A where the top neuron tends to fire slightly ahead of the bottom neuron and thus the middle neuron's assembly activation with the top neuron precedes, in time and thus phase, its co-activation with the bottom neuron. Not only would this selection bias explain the apparent phase shift when the spikes are categorized by assembly, it would also explain why the phase shift phenomenon occurred predominantly in phase-precessing neurons.

Putting all this together, I'm not sure what the assembly-based analysis is contributing, beyond the more traditional spike ensemble and sequence analyses, and it might be introducing a confound.

The single-neuron studies were more convincing to my mind. Here there are some other interesting phase-related observations: that phase preference varied according to spatial location, task type (choice vs. guided) and route (left-turns vs right-turns). However I am bothered by the correlation with firing rate, exposed by the analyses and evidenced in the examples in Figs. 4 and 5. This seems like another potential confound to me, perhaps because rate and phase tend to correlate within individual place fields (the later part of the field being lower in rate and earlier in phase). That rate and phase correlate does not rule out that the phase part of this effect is relevant, but it does seem slightly hard to reconcile with the story that phase is a cognitive code, because the coupling seems to reduce the degrees of freedom in the code (unless phase is the *only* important variable and rate is just a means to tweak the phase). A counterargument is that decoding was better with phase added than with rate alone, but the evidence to support this is weak as the decoding only made binary decisions between trial types. What would be more convincing is if the classifier is given only spike identity, spike time (or firing rate) and phase, and is able to decode both location and task context to a higher degree of accuracy than without the phase info. Trying the decoder with phase alone might also be interesting.

To summarise the second part, I think these are interesting and potentially valuable observations but with the small sample and theoretical uncertainties, I'm not fully sure what to take away from this other than "needs more data".

That said, I found the findings thought-provoking, and thus valuable from that perspective.

Other comments

Title: I am not a fan of "temporal" in relation to theta coding because everything about neurons is in a sense temporal, including firing rate – I'd suggest "phase" which is more precise

Abstract: "Sub-second, rhythmic modulation of spike times" is not very informative – how can you rhythmically modulate a spike time? I suppose what is meant is "in relation to the LFP oscillatory phase" but for an abstract this is too technical, or at least under-explained.

P3 What does "coding regimens" mean? What are "functional assemblies"? No demonstration of the functional nature of these temporally associated phenomena has yet been made. "Rate and temporal codes therefore coalesce" is hard to understand here.

P4 "Flexible agglomeration algorithm" needs explaining.

The cell assembly detection method needs to be explained – I ended up having to go to the eLife paper to understand even superficially how the analysis worked. The figure from that paper was very helpful and I suggest to reproduce it here, in the methods.

How many assemblies were recorded and how many units per assembly?

P5 "chance level of 0.2" – how was that determined? For the theta phase preference analysis, it wasn't clear here what aspect was being compared – the phase itself or the strength of the preference?

P6 "among all units taking part in at least two phase-locked assemblies" – how many was this? Fig. 3B looks like more than a slight trend to me – are the authors sure this analysis is correct? (It's hard to tell without knowing which data points are paired though and this could perhaps be shown).

P7 "This is not what we observed..." – this whole section could do with re-working as it was cognitively very demanding. First, the reader is asked to construct an imaginary scenario but then told to discard it, and the replacement scenario is many sentences in the making. I had to read this part several times before I understood it. In fact, in general the text is sometimes slow to get

to the point, which makes things a little hard-going sometimes.

P8 "53% of the tested units" – how many units is this?

No mention is made of splitter activity, but presumably the rate/phase correlates of task context (as in Fig. 5C) are an example of this? This might be worth discussing, in light of the possibility that the important thing about the "splitting" of the trial-types is actually not the rate but the phase.

Fig. 1D it seems like there is an assembly activation there that didn't get picked up by the algorithm... is it just that there were not quite enough spikes?

Fig. 2 legend what does "pruned" mean?

Fig. 3C Why do units 1 and 3 have the same plot?

Reviewer #2 (Remarks to the Author):

Integration of rate and temporal codes by hippocampal cell-assemblies supports theta phase coding of episodic information

Eleonora Russo, Nadine Becker, Aleks P. F. Domanski, Daniel Durstewitz, Matthew W. Jones

Summary

Russo et al. combine electrophysiological, behavioural and simulation approaches to characterise two different coding types expressed by hippocampal place cells: a rate code in the form of increased firing rate in specific regions of space, and a temporal code found in the phase-locking between individual spikes and local theta rhythm. While these two codes have already been demonstrated and extensively investigated, the authors evidence several potentially new findings: 1) the spikes of a single place cell can be phase-locked to different phases, and this is more likely to happen in different place fields (or different 'assemblies'); 2) locking to different phases can also take place in the same location if the ongoing task or future trajectory is different; 3) the amount of phase coding increases in trials involving memory of past trajectories. These findings will certainly be very relevant to researchers in the hippocampal /memory / plasticity field interested in network and cellular mechanisms. They raise interesting questions such as what are the mechanisms behind phase-locking, what are its roles if any, and whether individual place fields of the same cell are physiologically and functionally as different as place fields from different neurons.

Overall, the paper is concise and well-written, the methods are generally clearly explained and the combination of an experimental and modelling approach has a strong potential. The authors find that the same neuron can have multiple preferred phases, which is very interesting and, I believe, challenges some of the original models of phase precession. It is also interesting that the unsupervised approach used by the authors was able to find assemblies of two timescales that seem to correspond to those generally used in the field (place fields and theta sequences), providing an unbiased validation that these timescales are likely important for hippocampal information processing. However, it is unclear from the current analyses and literature review if the manuscript is not just providing a different way to look at previously-evidenced phenomena. I believe this can be clarified by adding a few more analyses or visualisations of the data as well as improving reports of past findings and their links with the authors' findings.

Major concerns

1. I am wondering to what extent the findings of different phase-locking of the same neuron correspond to phenomena that have already been discovered. In theta sequences, early phase firing tends to encode past/current locations while later phases encode future ones (e.g. Wang et al., 2020 (<https://science.sciencemag.org/content/370/6513/247>); Kay et al., 2020 but see also earlier literature). In addition, theta sequences are known to 'chunk' the environment (Gupta et

al., 2012, <https://www.nature.com/articles/nn.3138>), with a tendency to start and stop at maze turns or landmarks. Taken together, these phenomena will make it more likely that in the start of the track, future positions would be more strongly represented, thus expressing more spikes in the late phases of theta, while upon arrival at a maze turn or end, past positions will be more represented, thus firing more spikes in the early phase of theta. This has the potential to cause some of the 'trajectory/episodic' changes in theta phase locking evidenced in the manuscript. Could the authors add an analysis or discussion point investigating to what extent these known properties of theta sequences explain their findings? A possible way to start answering this could be to look at the spatial distribution of phase preferences on the maze (separated by trajectory type) and see how these are affected by turns and landmarks.

2. On a related note, it is unclear which of the findings are new, for example, Wang et al., 2020 found that "a subset of neurons ("bimodal cells") displayed a bimodal relationship with theta phase", showing that the same neuron can potentially have different preferred phases. The authors could do a more in-depth analysis and report of the existing literature to clarify which of their findings are completely novel and which replicate previously-reported phenomena (see more references in minor comments).

3. Not much is said in the manuscript about sharp-wave ripples and associated 'replay' sequences, however, replay and theta sequences are very different phenomena which should not be studied as one. Did the authors remove sharp-wave ripples to avoid contaminating their findings with replay? Did they speed-filter the data to remove low speeds where replay events are more likely to happen? Even if they only analysed times of high theta power, this could potentially still include 'exploratory' sharp wave activity (O'Neill et al., 2006, <https://www.sciencedirect.com/science/article/pii/S089662730500961X>). Mixing up potential replay with theta sequences in the 'spike-assemblies' would mean that some of the findings (e.g. different phase for different assemblies) could be due to the different nature of such assemblies. I would recommend either removing or (ideally) analysing separately the assemblies corresponding to putative replay events (increased multiunit activity taking place during sharp-wave ripples at low-speed).

4. Another potential concern is the relatively low number of simultaneously-recorded cells: ~130 cells recorded across 10 sessions, so about 13 cells per session on average. The maze used seems quite large with respect to average place field size, so it is unlikely that co-recorded place fields will homogeneously cover the maze, even with multiple fields. Could this sparse coverage interfere with the results? Specifically, could the rate assemblies discovered here be a result of inhomogeneous coverage of the maze by place fields, restricting the apparent co-firing of cells, while if the coverage was total, no or different assemblies would be discovered? Potential ways to alleviate this concern, apart from recording from larger ensembles, would include 1) comparing the results obtained for the session with lowest and highest cell yield (perhaps by highlighting these in the individual data plots) and/or 2) computing correlations between number of cells in a session and some of the parameters analysed and/or 3) show and compare the spatial distribution of the centroids and extent of detected cell assemblies, and that of the cumulated rate maps of all cells, for each session. If cell yield is found to impact the results, the authors should highlight this and discuss how it might impact the interpretation of the results.

Minor concerns

1. There are a few typos or grammatical mistakes that should be corrected (including in the figure legends).

2. The unsupervised assembly detection used by the authors uncovers mechanisms that have already been extensively known and studied (rate coding and time coding in hippocampus, place field scale and theta sequences scale). While the authors mention this (e.g. L124), they could attempt to quantify this link somehow, perhaps by comparing some previously published properties of place field and theta sequences to properties of the assemblies? Related to one of my main comments, they could also discuss and quantify to what extent spike assemblies relate more to replay or theta sequences.

3. The authors use the term 'episodic' in the title and elsewhere but episodic memories generally have 3 components: what, where and when. The authors do not evidence a temporal ('where') coding aspect. I would recommend replacing 'episodic' throughout the manuscript and title by 'task-related' or a more accurate word.

4. The introduction could be a bit less general and more focused. The discussion could also benefit from being more focused on matters directly related to the findings; in the present state, they both dwell quite a lot on theta sequences, which the authors did not directly analyse. I would recommend focusing more on the themes investigated in the paper: cell assemblies, phase locking mechanisms, within-cell variability of phase locking, what is known of the cellular mechanisms of phase locking and phase precession (ideally laying down different models or hypotheses). Some notions quickly mentioned in the results (e.g. L256 "Both experimental and theoretical studies have shown how a change in excitation received by a hippocampal unit can modify its phase of discharge") could be developed in the introduction. The goal of the study could be generally clarified (especially sentence l64) – what were the hypotheses and predictions?

5. The purpose and findings of the integrate-and-fire neuronal simulation could also be explained better. It would help if the authors explicitly stated different working hypotheses and the associated predictions that it is testing, and concluded on which one is supported. It is also unclear how this relates to the rest of the paper, given that the experimental findings focus on the difference of theta phase locking between different assemblies while the simulation attempts to explain how different assemblies can lead to different sequences of activated cells. Perhaps the simulation should focus on explaining phase-locking results instead?

6. General figure comments

- a. Could the authors indicate the start location (e.g. with a 'S') in all relevant maze plots?
- b. Given the generally small sample size in plots ($n \sim 10$) the authors could use boxplots to describe the data instead of bar graphs, which rely on mean and standard deviation, parameters that are likely not accurately summarising samples. This is however not crucial as the authors already show individual data points.
- c. Could the authors avoid using bright yellow in plots as it is hard to see?
- d. It would help in some places if legends were added inside the plots instead of having to read the text legend (e.g. Figure 3c).
- e. Could the authors show the actual position data of the animal on the example spike plots, when applicable (e.g. Figure 5)?
- f. In some of the plots showing individual data points there seems to be less points than the indicated n . This is probably because points are fully overlapping. Could the authors fix this (e.g. Fig 2b)?

7. Assembly-related questions

- a. How are the "assembly activity" and "assembly place fields" computed? From the examples provided (in Fig 3 or fig s3), they don't seem to correspond to a simple sum of individual rate maps. Could the authors explain this early in the results? e.g. mention of 'place field' of assemblies L 121 (same question for sup fig 2). Note for Fig S2, what is shown seem to be rate maps, not place fields (place fields would be a selected subset of the rate map).
- b. What is the unit of the assembly activity plots (e.g. Fig 1f,g "normalised assembly activity")?
- c. In Figure 1 L449 "the color scale shows the lag between the activation of each assembly-unit with respect to the unit first active in the assembly." Could the authors clarify this?
- d. What are the minimum and maximum sizes of assemblies, in terms of number of cells, if any? Could the authors provide a justification for the chosen duration limits (5 msec and 5 sec)? Could the authors clarify what "consistently reoccurring patterns" mean - is it enough for the same group of cells to co-activate within a given time-window to be considered an assembly, or do they have to co-activate in a specific temporal pattern, i.e., order?
- e. Figure 2a, unit 3 seems to be firing continuously like an interneuron. However, the authors mentioned that they only analyse place cells. Can the authors explain this apparent discrepancy? See also my later comment on only analysing putative pyramidal cells.
- f. Bimodality of assembly sizes: L 107 "strongly bimodal distribution": the authors should provide the results of a statistical test of bimodality (e.g. Hartigan's Dip Test?) or otherwise mention instead that the distribution appears / seems bimodal.

g. Could the author show the spatial distribution of both assembly types (ie, where do they occur) and comment on any biases in the spatial distribution?

8. The dimensions of the maze should be indicated in methods and the median number of place cells per session as well as lowest and highest numbers should be indicated in results (I90).

9. One might wonder to what extent spike-sorting errors (in particular, mistakenly combining two clusters that belong to different cells) could lead to the results about different preferred phases in different place fields. The authors could show that cluster quality measures are not correlated with percentage of cells having mixed phases. See for example Maurer et al., 2006.

10. Can the authors clarify in methods if, and if yes, how they checked if separate sessions recorded from the same animal might contain the same cells?

11. Are error trials included in the analysis? If the authors could analyse error trials and show, for example, a decrease of phase coding in these it would support the idea that phase coding is relevant for behaviour (one of the most interesting findings in my opinion).

12. Figure 3:

a. If the goal of figure 3c is to show task-dependent assemblies, it would be more convincing to show an example with cells active in the common, central stem, with a different activity for choice vs guided trials for the same place and movement direction. The currently-shown cells just seem to be directional place cells.

b. More generally, could the authors clarify what is the general message of panel c and what this means: "The change in firing phase of unit 2 when active within assembly A and B (a) is associated with a change in encoded information of the two assemblies (c)"

c. What is the unit of the yaxis (assembly activity) and what is the shaded area around the line?

13. To provide some explanation of the reason for different phases expressed by the same cell in different assemblies, the authors could investigate whether the cells belonging to a given assembly have more similar phases, i.e. is the phase somehow driven by the assembly? In other words, it would be interesting to compare the within-cell and across-cell as well as within assembly and across-assembly variability in phase preference.

14. Figure 4: Could the plots in b be chosen to correspond to the place cell example given in a?

15. Figure 5:

a. Could the authors indicate what the grey color represent in the rate-phase plots (I assume, out of field or other fields' spikes)?

b. "the degree of differentiation of the unit phase is maximal" where is this shown or tested?

16. L177 "Place-cell firing phase can encode distinct place fields" It is unclear what 'encoding a place field' means. Perhaps the authors mean "Distinct place fields can express different firing phases"?

17. To properly demonstrate that phase of spikes 'encodes' trial type and not other, covarying, parameters, the authors should incorporate information about possible confounding factors, such as speed and movement direction in the SVM classifier (if this is possible with this type of analysis). It could also include distance to the closest maze end /turn to address one of my main comments.

18. The authors could cite the recent findings of Tang et al., 2021 (<https://elifesciences.org/articles/66227>) in the discussion when mentioning prefrontal-hippocampal assemblies as well as context-dependent theta sequences (I390).

19. It seems that the Benjamini-Hochberg correction relies on defining a false discovery rate. Have the authors defined this somewhere?

20. The authors compare properties of firing within assemblies and out of assemblies. They could

comment on whether these differences, for rate assemblies, relate to observed differences between in-field and out-of-field firing in past papers (see for example Molter et al., 2012 <https://www.sciencedirect.com/science/article/pii/S0896627312006228>; Grienberger et al., 2017 <https://www.nature.com/articles/nn.4486>).

21. ~L719 please explain how were the rewards provided / delivered, was this automatic or done by the experimenter, could the rat see the experimenter preparing for delivery?

22. Place cell selection: were only putative pyramidal cells selected? If not, this should be added as a criteria (otherwise signals from fibers or low-firing interneurons could be wrongly categorised as hippocampal place cells). In addition, and only as a comment, it is becoming more common to have as an additional criteria a spatial information shuffle to select place cells, i.e. the spatial information of a place cell should be higher than the 95th percentile of the distribution of shuffled data from that same cell (spike train shuffle). This ensures that only cells with some degree of reproducibility of their spatial firing are included. I would recommend that the authors do this at least in future place cell studies.

23. The place field detection method (L831+) used by the authors seems new and relatively complex. If it has been used before, the authors should cite the relevant references. To help the reader get a good intuition of what the method does, the authors could add a supplementary figure explaining the method and showing the detected place field contours for a selection of example cells, with spike plots and rate maps. Ideally this figure would also show examples from a more classical field detection method [e.g., for a similar maze, Wirtshafter & Wilson, 2020; <https://elifesciences.org/articles/55252#s4-4-3>]. I note that Figure 4 shows one example but without a visualisation of the place field extent and without comparison to other place field detection methods. In addition, as the methods seems to rely only on spike numbers but not firing rate, can the authors explain how they deal with spurious regions of increased spike count due to the animal spending more time there?

24. L841/ L865: can the authors remind the reader of what is considered a trial category/type?

25. As mentioned in main comments, the authors could generally link better their findings to the existing literature, e.g. in the discussion. Here are a few papers that seem related to the manuscript and that the authors might want to cite and discuss:

- O'Keefe & Burgess, 2005 <https://onlinelibrary.wiley.com/doi/abs/10.1002/hipo.20115>
- Terada et al., 2017 <https://www.sciencedirect.com/science/article/pii/S0896627317304622>
- Wu & Yamaguchi, 2010 <https://link.springer.com/article/10.1007/s00422-009-0359-9>
- Venditto et al., 2019 <https://onlinelibrary.wiley.com/doi/full/10.1002/hipo.23100>
- Grienberger et al., 2017 <https://www.nature.com/articles/nn.4486>
- Yu & Frank, 2020 (preprint) <https://www.biorxiv.org/content/10.1101/2020.11.23.395012v1.full>
- Schmidt et al., 2009 <https://www.jneurosci.org/content/29/42/13232.full> (Isolated spiking was also more tightly phase locked to theta compared with adjacent spiking)
- Maurer et al., 2006 <https://onlinelibrary.wiley.com/doi/abs/10.1002/hipo.20202> specifically: "the cell fires with spikes clustered at two different phases over the theta cycles in which the fields overlap"
- Feng et al., 2015 (<https://www.jneurosci.org/content/35/12/4890>)
- The authors cite Kay et al., 2020, but they should more explicitly say how these findings are related to the present findings. See from Kay et al.: "we observed equivalent theta phase coding for additional representational firing patterns in the hippocampus: inbound path coding [...] and extrafield firing"
- The authors do not explicitly mention trajectory-dependent cells ('splitter cells') while several paragraphs seem related to these. They should mention at least the original discoveries (Wood et al., 2000 <https://www.sciencedirect.com/science/article/pii/S0896627300000714>; Frank et al., 2000 <https://www.sciencedirect.com/science/article/pii/S0896627300000180>) L205 and possibly discuss in what way the findings presented here could be related to what is already known about splitter cell activity (e.g. has it been found that different splitter fields have different preferred phases?).
- L310 "generally, units that received the highest depolarization activated first within the theta cycle while, importantly, units receiving just a light depolarization terminated the activation

sequence." => this seems related to a recent paper looking at replay sequences, Fernandez-Ruiz et al, 2019 (<https://science.sciencemag.org/content/364/6445/1082.editor-summary>), that the authors could discuss.

26. L529 "phase histogram of A spikes are shown" – what is A? and what is B mentioned later, do they correspond to cells shown in insets a and b?

27. L330 "we showed that such enhanced locking is due to the coordination within spike-assemblies" This seems to imply a causal link which has not been demonstrated. Please reformulate or provide an explanation to support this causal statement.

28. L374 "we found that the changes in phase preference also co-occurred with changes in the instantaneous firing rate of the unit ": where is this shown?

Reviewer #3 (Remarks to the Author):

This manuscript by Russo et al studies how the hippocampal place cell activities depend on task variables other than those directly related to place field properties, including task demand and the future and past of spatial trajectories. The authors examined two types of activities. One is participation in a cell assembly and the other is theta phase. They found that same cells can participate in different cell assemblies, which are expressed in two related time scales, and display different preferred theta phases, when the cells have multiple place fields that reflect different levels of task demand (choice vs guided trials) or different past/future trajectories. Although the topic has been examined in many previous studies, the results obtained here are valuable. However, there are a couple of issues that could be addressed to improve the manuscript.

1. A key question is whether the main result in the manuscript can be explained by the rate remapping phenomenon together with phase precession, without assigning phase preference itself into a functional role. A major part of the manuscript examines different phase preferences when cells participate in different cell assemblies or along different running trajectories. The underlining reason for all these seems due to the rate differences of same place fields along different trajectories, which is the well-known "split cell" phenomenon or trajectory-dependent rate remapping. Across the manuscript, it seems that low rate is associated with phases around 0/360 degree and high rate with 90-270 degree (examples in Figs. 4- 6). Therefore, phases differences in the same field across different conditions maybe just passively reflect the rate remapping, due to less degree of phase precession (e.g. not many spikes reaching the 180-270 phases, which typically associated with high instantaneous rates) when firing rate is low. The authors may consider explicitly to investigate this possibility.

2. It seems that place fields close to the reward sites were included in the analysis. Since theta is reduced and animal speed is different (lower) around the reward sites, the author may consider to remove place fields close the reward sites.

3. This paper by Sanders et al (Hippocampus, 29: 111, 2019) is highly relevant and should be cited.

Dear Reviewers,

Thank you for carefully and constructively reviewing our manuscript and sincere apologies for our delay in responding. Your primary concerns could only be met by adding substantially more data, which naturally proved challenging in the midst of COVID-19 lockdowns. I have moved across two institutions (and countries) in the meantime, regrettably interrupting our progress. Nevertheless, we have now made extensive additions and amendments in light of your feedback, corroborating and strengthening our original results.

In summary: we added data from one new rat and two extra sessions for the previously included 5 rats, giving a total of 14 extra recording sessions (amounting to 159 new units, and an updated total of 322 CA1 units). We re-ran *all* analyses of the manuscript with the new population of units and according to the different controls you suggested, inserting 7 extra supplementary figures and 19 new citations. We also added new analyses to address specific concerns, including re-running all analyses but excluding spikes coinciding with sharpwave-ripples. All updated analyses replicated and/or extended our original findings.

Please note that all central analyses in this manuscript are computed using only putative place cells, identified here by selecting units with a mean firing rate between 0.2 Hz and 4 Hz on the maze and with spatial information above 0.5 bits/s. While re-running all analyses for revision of the manuscript, we realized that we applied this filter on spatial information only after performing the assembly analyses. Because of this, some assemblies in the previous version included a few non place cell units. We apologize for this mistake which we have now corrected, re-running the assembly analysis from scratch. As the erroneously included units were few (32 out of the previous total 163 units; the new unit count is 322 CA1 units, 218 of which were classified as putative place cells) and rarely took part in assemblies, this did not impact the previous results which are confirmed in the new analysis.

In the following reply: blue marks the reviewer comments, italic marks the text extracts from the manuscript, and red marks the new additions to the text.

Reviewer #1 (Remarks to the Author):

In this experiment, CA1 neurons were recorded as rats ran a forced-choice double T-maze experiment in which the start location was experimentally controlled (“guided”) and the choice depended on the start. It is thus a working memory spatial task that switches between guided and free-choice phases. The authors deployed a recently developed analytic method that looks for repeated patterns of co-firing among ensembles, or “assemblies,” of neurons: the period of time over which the firing was analysed varied from 5ms to 5s. It was found that within this span, patterns of co-firing (activations of an assembly) clustered around either short times (~30ms) or long times (~3s). Most units belonged to assemblies active at both timescales, and there was a very high likelihood that neurons that co-fired at the short timescale would also co-fire on the long timescale. When the authors looked at the firing of units in relation to theta phase they found that phase preference changed between different place fields of the same unit, and also between different task states (guided vs. choice, left-turn runs vs. right-turn runs etc) and that cells that showed theta phase precession tended also to be the ones that showed this phase shift between fields. The authors suggest that the variation in phase relations could result in a plastic sorting process, driven by the level of input activation, in which units could be temporally shuffled

within sequential assembly activations, as a function of task state: something that could be used to encode episodic components of a memory.

This is a well-designed and well-illustrated experiment and there are some very clever ideas in this paper, but I am concerned about the sample size and the possibility that the phenomena uncovered here could arise from sampling biases.

We thank the Reviewer for the careful reading and assessment, and for appreciating our work. We hope that the new data, analyses, and additions to the text have improved the manuscript and allayed concerns about sampling size and bias.

The issues are as follows.

R1.1 First, the sample size is rather small, being only 131 units across five animals, which makes these population-coding analyses slightly insecure.

We now added one extra rat to the original five, plus included data from two additional recording days for each animal (for a total of 14 extra recording sessions and 24 sessions in total). The new dataset now includes six animals recorded in four sessions, giving a total of 322 recorded CA1 units (218 of which classified as putative place cells on the maze). All tests were rerun from scratch on this new dataset, confirming and strengthening the previously reported results.

R1.2 I didn't get a sense of how large the assemblies were nor how many there were, as data were pooled across animals and separated by session, such that two sessions from each animal contributed data.

This information is now added in line 131: "*We found: (1) 137 sharp spike patterns involving on average about 17% units per session per pattern (with a maximum of a 3-unit assembly in a 20-unit set) and a temporal precision in the range of 0.006 - 0.06 sec centered around 0.028 sec (spike-assemblies) and (2) 204 broader firing rate patterns with on average 28% units per session per pattern (with a maximum of an 11-unit assembly in a 19-unit set) and temporal precision between 0.07 - 5 sec (rate-assemblies) (Fig. 1b).*"

R1.3 These were not analysed (as they should have been) with reference to the interdependence of data (that is, with a multi-level analysis).

Thank you for raising this oversight, which we have corrected throughout the manuscript. Where appropriate, we now performed general (or generalized) linear mixed-effects models to account for rat identity and recording sessions as well.

At line 1080 of the Methods we now write: "*Because of the dependence introduced in the data by pooling units from multiple sessions of the same animals, when appropriate we perform statistical tests by generalized linear mixed-effects models, where we explicitly account for session and rat identity.*", updated tests whenever necessary throughout the manuscript, and specified details of the mixed-effects model when reporting each performed test.

R1.4 Second, the unbiased assembly analysis is presented as uncovering two timescales as if this is a new finding, but we have known for decades that there are two timescales in CA1 activation, which occur in three contexts. These are: (i) short-timescale synchronous bursting activity that is often/usually accompanied by sharp-wave ripples in the LFP and is usually called "replay", (ii) theta sequences which occur in a single theta cycle when the animal is in a theta state (e.g., when running) and reflect the spike

timing relationships in the active place cell population, and (iii) place field activation which also occurs during theta and reflects the overall time-averaged firing rate pattern of place cells as the rat passes through each field in succession. The bulk of the analyses in the paper did not separate these three contexts so it seems likely that the short-timescale assemblies are a mixture of replay events and theta sequences, and the long-timescale activations are place field traversals. This does not invalidate the phase findings but the paper does present, as a new result, something that is not really new, and it requires some effort and prior knowledge to uncover what is going on here.

The assembly analysis is indeed an unbiased approach to studying the coordination patterns present in a dataset. The assembly detection method applied here is the only one, to the best of our knowledge, to return explicit information about the coordination timescale of the detected assemblies without a need to pre-specify their expected structure. Thus, while we completely agree that the two predominant timescales of CA1 activity are well-established, we suggest that reporting the unsupervised timescale analysis serves as confirmation of the known coding modalities of the hippocampus and is therefore relevant to the field (e.g. Reviewer #2 writes: “It is also interesting that the unsupervised approach used by the authors was able to find assemblies of two timescales that seem to correspond to those generally used in the field (place fields and theta sequences), providing an unbiased validation that these timescales are likely important for hippocampal information processing.”). The manuscript’s opening analysis of the assembly characteristics reassured us (and the reader) that the detected assemblies were in line with what is known from the literature. We did not mean to describe the presence of different timescales in hippocampal coding as a new finding, and we thank the reviewer for pointing out the possible misunderstanding. The presence of three main coordination timescales in the hippocampus (replay events, theta sequences, and place fields) is covered in the Introduction; we have now also reinforced in the Results section that spike-assemblies could, in principle, capture both replay events and theta sequences.

We now included the information at line 128: “*As expected based on extensive previous analyses of place cell physiology, the temporal precision of hippocampal assemblies active during the task ranged from milliseconds to seconds and is bimodally distributed (Hartigan’s dip test for unimodality, n . bootstrap samples = 10^5 , $dip=0.03$, $p = 0$) into two major groups.*”

And at line 151: “*Rate-assemblies reflected the simultaneous or sequential activation of the place fields of their constituent units in specific maze locations and/or along task-relevant trajectories, respectively (Supplementary Fig. 3). Their characteristic temporal scale, ranging from hundreds of milliseconds to seconds, was indeed compatible with the time needed by the animal to traverse the place field of a unit. Spike-assemblies, whose timescale is compatible with replay events or theta sequences (Lee and Wilson, 2002; Skaggs et al., 1996), had a more localized activation (with average spatial information of 2.67 ± 0.09 in contrast to 1.91 ± 0.07 for rate-assemblies, general linear mixed-effects model of the spatial information of assemblies according to the assembly type, spike- vs rate-assembly, $F(1, 419) = 70.55$, $p = 7.0 \cdot 10^{-16}$) which often appeared to be coordinated with the theta rhythm of the local field potential (Fig. 1d).*”

Further, to determine whether the observed phase-shift phenomenon was potentially modulated by the occasional presence of sharp wave ripples and replay events, we repeated all fundamental analyses of the manuscript (both at assembly and single unit level) but excluding spikes fired during SWR. The results of the analyses performed excluding SWR epochs are reported in **Supplementary Fig. 4, 5, 9,**

11. This control analysis confirmed our original results, ruling out the possibility that contextual phase-shift could be due to the presence of SWR-associated replay events.

We now added the following text.

At line 161: *“To understand whether the observed coordination at short timescales was related specifically to replay events, we repeated the assembly detection but excluding epochs in correspondence of SWRs. As shown in **Supplementary Fig. 4**, the temporal resolution of the detected assemblies was conserved, showing that SWRs were not the prevailing source of fast coordination under these conditions.”*

At line 175: *“See **Supplementary Fig. 5a** for the same analysis excluding spikes fired during SWR.”*

At line 185: *“. See **Supplementary Fig. 5b** for the same analysis excluding spikes fired during SWR.”*

At line 582: *“See **Supplementary Fig. 4** for the same analysis on assemblies detected excluding spikes fired during SWR;”*

At line 206: *“See **Supplementary Fig. 5c** for the same analysis excluding spikes fired during SWR”*

At line 313: *“These results withstood using different methods to identify place fields (**Supplementary Fig. 10**) and removing spikes fired during epochs of low theta-power or during SWR (**Supplementary Fig. 9**).”*

In the Method section at line 1158:

“Detection of SWR

*SWR detection was based on the ‘Sleepwalker’ MATLAB toolbox (<https://gitlab.com/ubartsch/sleepwalker>). In brief, LFPs were down-sampled to 1000 Hz and 50 Hz notch filtered prior to band-pass filtering (using least squares filters) between 120-220 Hz. Candidate ripple events were identified based on threshold crossings in the z-scored 120-220 Hz power, using a threshold of 3.5 x SD of the signal. Start and finish times of the ripple were calculated based on 2 x SD of 120-220 Hz power. Representative samples of individual and averaged events were then visually inspected. Ripples were rejected if they were shorter than 50 ms or longer than 500 ms; if gaps of less than 50 ms occurred between events, they were treated as a single ripple. Ripple start/end timestamps were used to exclude SWR-associated spiking for some analyses (**Supplementary Fig. 4, 5, 9, 11**).”*

Finally, to further clarify that the detected assemblies are in line with known phenomena we added in the discussion 406: *“We detected place cell assemblies using an unsupervised algorithm able to extract coordinated activity from the data without pre-defining timescales of interest. This corroborated extensive evidence that hippocampal coding is characterized by two predominant time scales: a rate scale (‘rate-assemblies’) reflecting place field firing rate modulation, and a sub-second temporal scale (‘spike-assemblies’) compatible with the entrainment of spikes by theta rhythms and during SWR-associated replay events. The relatively broad timescale characteristic of rate-assemblies suggests that their coordination is not solely imposed by the shared modulation of their composing units by the local theta rhythm, in agreement with their robustness to degradation of cholinergic signaling (Venditto et al., 2019). Nevertheless, spikes fired within either spike- and rate-assemblies coordinated with the theta rhythm, in line with previous literature reporting theta phase-locking of CA1 neurons when active in an assembly configuration (Buzsáki and Moser, 2013; Harris et al., 2003; Lopes-dos-Santos et al., 2013). We show that such theta locking is most pronounced for spike-assemblies, but present in rate-assemblies as well.”*

And in the Discussion at line 429: *“This was true both for rate- and, more frequently, spike-assemblies, and was not the by-product of occasional replay events.”*

R1.5 The first really novel and potentially very interesting observation is that neurons changed their phase preference when active in different assemblies of the same type (short-timescale vs long-timescale). However I am troubled by a potential confound here which arises from the fact that the allocation of a spike to a given assembly was determined by the timing of the other neurons in that assembly. It is possible therefore that one assembly tends to associate with all the spikes at the beginning of that neuron's place field and another at the end, due to the relative locations of the place fields of that assembly's other neurons, and thus that one assembly tends to link to the beginning of the phase precession cycle and the other at the end. One can actually see this in the example in Fig. 3A where the top neuron tends to fire slightly ahead of the bottom neuron and thus the middle neuron's assembly activation with the top neuron precedes, in time and thus phase, its co-activation with the bottom neuron. Not only would this selection bias explain the apparent phase shift when the spikes are categorized by assembly, it would also explain why the phase shift phenomenon occurred predominantly in phase-precessing neurons.

When we first observed that the same neuron could change phase preference when active in different assemblies we had exactly the same concern: if two assemblies would systematically sample (activate in) the initial and final part of a place field this could result in an assembly-specific change in phase preference due to sheer phase precession. This concern was in fact what motivated us to further analyze the activation phase of single units with respect to their *whole* place field. We discussed the issue and the rationale for the single-unit analysis in the first paragraph of the section "Place-cell firing phase can encode distinct place fields".

We now rephrased it to be more clear, at line 227: "*A possible confound for the presence of contextual phase-shift coding in different assemblies comes from the fact that the spikes fired within an assembly might not uniformly sample the place field of a unit. Thus, if two assemblies systematically sampled the initial and final part of a phase-precessing unit's place field respectively, this could result in an assembly-specific change in phase preference. To rule out this possibility, we analyzed the phase preference of single units, this time separating their spikes according to their own place fields instead of by assembly membership.*"

And at line 250: "*Note that for this test we included all spikes fired within a place field and compared them with all spikes fired in another place field of the same unit. Thus, the observed phase-shift cannot be explained by a biased sampling of different subregions of the place fields.*"

(Just a note in relation to the Reviewer's sentence "due to the relative locations of the place fields of that assembly's other neurons" to clarify: the assemblies identified by the algorithm are not necessarily synchronous and the lag between the activation of different neurons is chosen by the algorithm to maximize the coordination between units. Thus, while it is true that different assemblies could potentially sample different parts of a place field, this would not be due to the position of the other neurons in the assembly but, if anything, to an enhanced coordination that occurs more in one subsection of the place field than another.)

While we don't exclude that some of the phase change detected in association with the activation of different assemblies is due to the phenomenon of phase precession combined with a subsampling of place fields, the single unit analyses confirm that this can not be the only explanation for contextual phase-shifting.

R1.6 Putting all this together, I'm not sure what the assembly-based analysis is contributing, beyond the more traditional spike ensemble and sequence analyses, and it might be introducing a confound. The single-neuron studies were more convincing to my mind. Here there are some other interesting phase-related observations: that phase preference varied according to spatial location, task type (choice vs. guided) and route (left-turns vs right-turns).

Assembly analyses, single-unit analyses, and the computational model are here presented as complementary and mutually reinforcing. The assembly analysis puts the described phase-shift of CA1 units in relation to assembly activations. Single unit analysis, then, aimed to more rigorously exclude that phase-shift could be exclusively the result of phase precession and place field subsampling, highlight the relation between phase and instantaneous firing rate, and study how phase changes can encode task-relevant information. Finally, the computational model binds the assembly and single-unit perspective together, proposing a mechanistic model of assembly-specific phase-shift coding and generation of task-specific theta sequences.

If we had not performed the assembly analysis but only the single-unit analysis and the computational modeling (which proposes a link between assembly activation and phase coding), we would not have been able to address the question of whether there is at all evidence of single-units changing phase preference in different assemblies.

We hope that the new changes throughout the text of the manuscript helped clarify the relative roles of these multiscale analyses and their convergent results.

R1.7 However I am bothered by the correlation with firing rate, exposed by the analyses and evidenced in the examples in Figs. 4 and 5. This seems like another potential confound to me, perhaps because rate and phase tend to correlate within individual place fields (the later part of the field being lower in rate and earlier in phase). That rate and phase correlate does not rule out that the phase part of this effect is relevant, but it does seem slightly hard to reconcile with the story that phase is a cognitive code, because the coupling seems to reduce the degrees of freedom in the code (unless phase is the *only* important variable and rate is just a means to tweak the phase). A counterargument is that decoding was better with phase added than with rate alone, but the evidence to support this is weak as the decoding only made binary decisions between trial types.

Our thesis is that rate and phase are both modulated by the degree of internal depolarization of the place cells. This means that they are correlated and thus, to a certain extent, carry interrelated information. We do not see this as a limitation, but as a strength. Rate coding is more robust than phase coding (as it integrates over many spikes) and can be read reliably by downstream regions in hundreds of ms. The information carried by phase coding, while more dynamic as it relies on only few spikes, is immediately readable in milliseconds.

Moreover, and maybe more importantly, while it was initially proposed that theta sequences are hardwired as resulting of learning mechanisms [Tsodyks, et al. 1996], if unit rate (or better the unit depolarization) and phase correlate, as shown by our results, the spiking order of two neurons along the theta sequence is flexible and can vary according to the specific degree of depolarization of the units in a specific context. This allows the creation of context-specific theta sequences which may distinguish different contexts in the same space. Finally, the temporal coordination between different units of a theta sequence is much higher than that provided by their rate coordination (this is captured explicitly by the

cell assembly analysis which assigns, in an unsupervised manner, a higher temporal resolution to spike-assemblies than rate assemblies). This enhanced coordination facilitates spike-timing dependent plasticity and other forms of LTP likely to shape sequential learning over the course of experience.

We now clarified our logic in the relevant discussion sections. In particular:

We now added extra text and modified Fig. 6f to highlight how the proposed model could provide an explanation of how rate and phase are both modulated by the degree of internal depolarization of the place cells. At line 382 we wrote: *“As expected, assembly activations produced a broad increase in unit firing rate punctuated by faster temporal coordination with the theta oscillation (Fig. 6f bottom). This coordination was assembly-dependent and the simulated units changed their phase preference of firing, here computed with respect to the timecourse of the excitatory depolarization, when active in the two assemblies (Fig. 6f right).”*

In the discussion we also wrote at line 465: *“In line with this evidence that changes in depolarization lead to changes in discharge phase, we found in our data that the changes in phase preference of individual units also co-occurred with changes in the instantaneous firing rate. While a recent study has shown that anatomically distinct place cell subpopulations in superficial and deep sublayers of dorsal CA1 pyramidal cell layer bias towards rate and phase coding of spatial information respectively, as the richness of sensory cues in the local environment is experimentally manipulated (Sharif et al., 2020), our results suggest an integration of rate and phase coding within a population to supports coding of complex information. This was also replicated through a model of adaptive exponential integrate-and-fire units similar in spirit to the soma-dendritic interference models ((Harris et al., 2002; Losonczy et al., 2010); c.f. Fig. 1e). The model confirmed that the activation of an assembly imposed an assembly-specific rate and phase preference on each assembly unit, thus producing, at the population level, assembly-specific theta sequences. Thus, while correlation between rate and phase changes has been observed during rate remapping (Sanders et al., 2019), our findings demonstrate that phase-shift coding extends beyond rate remapping and occurs also between distinct place fields or assemblies, frequently coinciding with changes in instantaneous firing rate, similar to those observed for splitter cells (Duvell et al., 2023; Frank et al., 2000; Wood et al., 2000). The model predictions are also in line with recent work showing that individual place cells quickly switch between the encoding of alternative future locations or heading directions on alternate theta cycles, with lower firing rates for non-preferred directions occurring in later theta phases (Kay et al., 2020). The mechanism highlighted by our analysis and model is not the sole modulator of firing of CA1 cells, as the timing of specific inputs from Schaffer collaterals and the perforant path also plays a role in their phase of discharge (Fernández-Ruiz et al., 2017; Schomburg et al., 2014). Nevertheless, it captures the role played by various depolarization settings in inducing a modulation and discretization of phase preference of hippocampal CA1 place cells. We therefore propose that phase-shift coding may be a consequence of the different levels of depolarization generated by the specific constellations of synaptic input resulting from the activation of different assemblies as the animal encounters different cognitive or environmental contexts (cf. Fig. 6d).”*

To summarise the second part, I think these are interesting and potentially valuable observations but with the small sample and theoretical uncertainties, I'm not fully sure what to take away from this other than “needs more data”. That said, I found the findings thought-provoking, and thus valuable from that perspective.

We thank the Reviewer for appreciating our work and we hope we addressed satisfactorily all raised concerns.

Other comments

R1.8 Title: I am not a fan of “temporal” in relation to theta coding because everything about neurons is in a sense temporal, including firing rate – I'd suggest “phase” which is more precise

We initially chose “temporal code” because it is the more common term used when compared to “rate code” than “phase coding” (phase coding is, in our minds, just one instance of temporal coding). Nevertheless, to specify which type of temporal code we refer to, we embraced the Reviewer's suggestion and changed the title to “*Integration of rate and phase codes by hippocampal cell-assemblies supports flexible encoding of spatiotemporal context*”.

R1.9 Abstract: “Sub-second, rhythmic modulation of spike times” is not very informative – how can you rhythmically modulate a spike time? I suppose what is meant is “in relation to the LFP oscillatory phase” but for an abstract this is too technical, or at least under-explained.

We now changed the abstract sentence as “*Spatial information is encoded by location-dependent hippocampal place cell firing rates and sub-second, rhythmic **entrainment** of spike times.*”

R1.10 P3 What does “coding regimens” mean?

We rephrased the whole sentence as follows, line 81: “*Here, we investigate and quantify the extent to which flexible and transient activation of place cell assemblies affects both firing rate and phase/temporal coding modalities of individual place cells, enabling the discrimination of different visits to the same locations under varied cognitive demands.*”

What are “functional assemblies”? No demonstration of the functional nature of these temporally associated phenomena has yet been made.

The term “functional assemblies” is a term used in the literature on methods for cell assembly detection to refer to the fact that the identified assemblies have been detected on the basis of their coordinated activity rather than being defined anatomically (structurally). In this framework, the adjective “functional” does not refer to the functional role played by the assembly in driving the animal behavior (or encoding task information) but it refers to the modality of assembly detection. Another example of the adjective “functional” used in the same meaning is “functional connectivity”, commonly referred to in the brain imaging literature.

We now specified this information in line 122: “*We use the term cell assemblies without making any assumptions about the anatomical connectivity between assembly units, which are identified solely based on their co-activation. Here we thus refer to a ‘functional cell assembly’ as any group of units whose activation coordinates with temporal precision between 5 msec and 5 sec in an arbitrarily lagged but consistently reoccurring pattern.*”

“Rate and temporal codes therefore coalesce” is hard to understand here.

Now changed to “*Rate and temporal codes therefore **interact**.*”

R1.11 P4 “Flexible agglomeration algorithm” needs explaining. The cell assembly detection method needs to be explained – I ended up having to go to the eLife paper to understand even superficially how the analysis worked. The figure from that paper was very helpful and I suggest to reproduce it here, in the methods.

We thank the reviewer for pointing this out, now added an explanation of the agglomeration algorithm in the method section, lines 1091: “*The algorithm is based on a recursive **agglomeration** scheme, at each step of which it detects and tests assemblies of progressively larger size. As the first step of the algorithm, CAD scans the activity of all possible pairs of units in the recorded set to select the most common shared (potentially lagged) activation pattern, and test if this reoccurs more frequently than expected by chance. If the test is significant, a new time series is created to reflect the activation of the*

detected assembly-pattern. In the second step of the algorithm, the new assembly activation time series are tested in turn including remaining recorded units. If the test is significant the unit becomes part of the assembly and a new assembly activation time series is generated. The algorithm stops when no more units can be added to the assemblies detected in the previous agglomeration step (see (Russo and Durstewitz, 2017) for a more detailed description of the algorithm). It follows that the hypothetical inclusion of supplementary units beyond those actually recorded would affect the number and size of the detected assemblies but it would not alter the coordination patterns already detected in the existing set. In this manuscript, the detection of paired assemblies was performed stopping the agglomeration at the initial pairwise step, while full-size assemblies were detected letting the algorithm agglomerate until completion.”

To which we refer in the main manuscript in line 117: “The algorithm automatically corrects for non-stationarity in the units’ activities and scans spike count time series at multiple temporal resolutions, returning the characteristic timescales at which individual assembly patterns coordinate (**Supplementary Fig. 2**). Thanks to a flexible agglomeration algorithm, CAD can detect assemblies with any activity pattern, avoiding a priori limits on the characteristics of the detected motifs (see *Methods*).”

We also included a description of how the assembly activity is computed, line 146: “Assemblies are considered active whenever all units composing the assembly fire spikes matching the assembly activation pattern identified by the algorithm. The assembly is considered to have an activation of n , when all units composing the assembly fire at least n spikes in the bins matching the assembly pattern.”

and a new supplementary figure:

Supplementary Fig. 2 | Cell assembly detection with CAD. Hippocampal assemblies were detected by applying the method for Cell Assembly Detection (CAD) presented in (Russo and Durstewitz, 2017). CAD is able to detect assemblies in an unsupervised fashion returning for each assembly the identity of its composing units, the activation pattern, and the temporal resolution of the unit coordination. To exemplify the type of information returned by CAD, we here report the algorithm performance when applied to a set of 50 simulated units containing 5 distinct assemblies of 5 units each. To showcase the wide range of CAD sensitivity, the simulated assemblies differed both in activation pattern and temporal resolution. (a) Rastereplots of 25 of the 50 simulated units (in blue), showing the activation of the five simulated assemblies (in red): type I – highly precise lag-0 synchronization; type II – highly precise

sequential pattern; type III – highly precise spike-time pattern without clear sequential structure; type IV – rate pattern with sequential pattern; V – synchronous rate increase. Units from 26 to 50 were not included in any assembly. (b) Assembly-assignment matrix showing the output of CAD. Each detected assembly corresponds to a column of the assembly-assignment matrix. Colored units (rows) have been identified as part of the assembly. The color indicates the lag between the activation of the unit and that of the first unit active within the assembly. Along the abscissa, in grey-scale, the temporal resolution of the detected assembly. (c) Fraction of correctly assigned units (retrieval score) as a function of the temporal resolution (bin size) at which the algorithm scans the spike train. Data averaged across 70 independent runs, error bars = SEM. Thanks to the non-stationarity correction implemented in the algorithm (see (Russo and Durstewitz, 2017)), CAD detects assemblies only at their characteristic temporal resolution. Thus, precise assemblies of type I and II were only detected at very small bin sizes, while rate assemblies of type IV and V only at bigger bin sizes. Interestingly, the type III assembly was detected both at small and larger bin sizes because of the precise timing in the relative activation of assembly units, and the change in firing rate produced by its extended activation pattern. Figure adapted from ref. (Russo and Durstewitz, 2017) under CC-BY 4.0 license.

R1.12 How many assemblies were recorded and how many units per assembly?

This information is now added in line 131: “**We found: (1) 137 sharp spike patterns involving on average about 17% units per session per pattern (with a maximum of a 3-unit assembly in a 20-unit set) and a temporal precision in the range of 0.006 - 0.06 sec centered around 0.028 sec (spike-assemblies) and (2) 204 broader firing rate patterns with on average 28% units per session per pattern (with a maximum of an 11-unit assembly in a 19-unit set) and temporal precision between 0.07 - 5 sec (rate-assemblies) (Fig. 1b).**”

R1.13 P5 “chance level of 0.2” – how was that determined?

We apologize as we noticed we copied the wrong number. We now corrected it and added more details on how we determined such a number. In line 139 we added: “**Moreover, two units taking part in the same spike-assembly were more likely to join the same rate-assembly than expected by chance (average probability of 0.9 against a chance level of 0.6, $p < 10^{-5}$ computed by bootstrap, see Methods).**”

And in the Methods, line 1133: “**Computation of chance level for the probability of spike-assemblies being detected as rate-assemblies as well. Spike-assembly pairs had a probability of $p = 0.9$ to be also detected as rate-assembly pairs. We tested if the obtained value is above chance by bootstrap. To this aim, we considered all possible pairs of units present in the recorded sessions and randomly selected an amount equal to the number of the detected rate-assembly. We then computed the probability of the detected spike-assembly pairs to be part of this subset p_{boot}^i . We repeated the random sampling 10^5 times and averaged the result to obtain the chance level $\langle p_{boot} \rangle = 0.6$. The p -value corresponds to the fraction of bootstrap sampling with $p < p_{boot}^i$, which, in the specific bootstrap set, never occurred.**”

R1.14 For the theta phase preference analysis, it wasn't clear here what aspect was being compared – the phase itself or the strength of the preference?

Both aspects were considered in different analyses. We now added in the method section, line 1175: “**Phase-locking of assembly-spikes vs. unit-spikes. This test compares the strength of phase-locking of two sets of phase values. In particular, we tested via bootstrap if ...**”. And at line 1184: “**Change in phase preference for spikes fired in different assemblies. This test assesses if two sets of phase values have the same median phase. For each unit taking part ...**”

R1.15 P6 “among all units taking part in at least two phase-locked assemblies” – how many was this?

Now added in line 196: “We found that among all units taking part in at least two phase-locked assemblies (*n = 64 in spike-assemblies and n = 169 in rate-assemblies*)...”

R1.16 Fig. 3B looks like more than a slight trend to me – are the authors sure this analysis is correct? (It’s hard to tell without knowing which data points are paired though and this could perhaps be shown).

The analysis was underpowered and with the new data became significant. Moreover, we also now substituted the t-test with a generalized linear mixed-effects model to account for rat identity and recording session as well.

The text in the manuscript has now been changed in line 200: “*This relative phase-shift was found both in spike- and rate-assemblies, with a higher proportion of units with significant phase-shift in spike-assemblies (Fig. 3b, generalized linear mixed-effects model of the probability of a unit to phase-shift when firing in different assemblies according to the assembly type, spike- vs rate-assembly; with binary dependent variable for significant phase shift and logit link function: $F(1,231) = 12.54, p = 4.8 \cdot 10^{-4}$. The model accounts for rat identity and recording session as covariates. See Supplementary Fig. 5c for the same analysis excluding spikes fired during SWR).*”

R1.17 P7 “This is not what we observed...” – this whole section could do with re-working as it was cognitively very demanding. First, the reader is asked to construct an imaginary scenario but then told to discard it, and the replacement scenario is many sentences in the making. I had to read this part several times before I understood it. In fact, in general the text is sometimes slow to get to the point, which makes things a little hard-going sometimes.

We rephrased the paragraph, now hopefully with less convoluted wording, lines 227: “*A possible confound for the presence of contextual phase-shift coding in different assemblies comes from the fact that the spikes fired within an assembly might not uniformly sample the place field of a unit. Thus, if two assemblies systematically sampled the initial and final part of a phase-precessing unit’s place field respectively, this could result in an assembly-specific change in phase preference. To rule out this possibility, we analyzed the phase preference of single units, this time separating their spikes according to their own place fields instead of by assembly membership. In single units with multiple place fields, different rate-assemblies often activated in correspondence to different place fields of the unit (e.g. cf. Fig. 3c and Supplementary Fig. 6). As single units changed their phase preference when active in different rate-assemblies (cf. Fig. 3b), separating unit spikes by place field could reveal similar shifts in phase to those observed when separating them by rate-assembly.*”

R1.18 P8 “53% of the tested units” – how many units is this?

We now specified this information and updated the number including units from the added sessions. In line 246: “*Comparing the phase preference with respect to place field location, we found that 43% of the tested units changed firing phase when active in different place fields (value obtained as mean across sessions weighted by the number of units tested per session, 208 units among 24 sessions. Fraction computed after Benjamini–Hochberg correction for multiple comparisons, $\alpha = 0.05$, Figs. 4b, 5b).*”

R1.19 No mention is made of splitter activity, but presumably the rate/phase correlates of task context (as in Fig. 5C) are an example of this? This might be worth discussing, in light of the possibility that the important thing about the “splitting” of the trial-types is actually not the rate but the phase.

Thanks for pointing this out. Indeed, we mistakenly omitted explicit reference to 'splitter cells'.

We now refer to it in line 38: “Among pyramidal cells of hippocampal CA1, transient firing rate increases lasting from hundreds to thousands of milliseconds encode the position of an animal within the environment (‘place cells’ (Muller and Kubie, 1987; O’Keefe and Dostrovsky, 1971)), routes through paths with overlapping segments (“splitter cells” (Duvelle et al., 2023; Frank et al., 2000; Wood et al., 2000)), signal goal-locations (Hok et al., 2007), mark time intervals (Eichenbaum, 2014), respond to specific odors (Eichenbaum et al., 1987), sounds (Aronov et al., 2017), objects (Fried et al., 1997) and, in humans, to other people’s identities (Rey et al., 2020).”

At line 290: “Moreover, while it is known that splitter cells can differentiate between trial types by rate modulation, we found that adding information relative to the spike phase to the instantaneous firing rate further improved the decoding performance of the SVM...”

And in the discussion at line 477: “Thus, while correlation between rate and phase changes has been observed during rate remapping (Sanders et al., 2019), our findings demonstrate that phase-shift coding extends beyond rate remapping and occurs also between distinct place fields or assemblies, frequently coinciding with changes in instantaneous firing rate, similar to those observed for splitter cells (Duvelle et al., 2023; Frank et al., 2000; Wood et al., 2000).”

R1.20 Fig. 1D it seems like there is an assembly activation there that didn’t get picked up by the algorithm... is it just that there were not quite enough spikes?

All assembly detection algorithms based on binning of time series share the common problem of occasionally missing assembly activations if these occur at the binning edge. In these cases, the assembly pattern is indeed not met because broken into neighboring bins, the activation is thus not accounted for. This is a limited problem, as the number of missed activations is still very small with respect to the detected ones and, if any, it reduces the detection power of the algorithm but doesn’t lead to false positives. Still, occasionally some activation is missed. This is what happened in Fig. 1D. As this happens quite rarely we decided to change the example in the figure to avoid confusion.

R1.21 Fig. 2 legend what does “pruned” mean?

We now explain this at line 614: “(d) Temporal precision of spike- and rate- assemblies with phase-modulated spikes (yellow) and assemblies with spikes with enhanced phase-modulation with respect to their composing units (red). Assembly pairs detected with CADopti separately in the 5 - 60 ms (spike-assembly) and 0.07 – 5.0 sec (rate-assembly) resolution window. CADopti prunes redundant assemblies and selects those with the lowest p-value in each resolution window (see Methods). Bars show weighted mean and SE pooled from all sessions (mean weighted by the number of units per session).”

And in the Method section, line 1117: “Thus, each assembly was unique within each window but could be detected in both time windows. This pruning procedure allowed a fair comparison between the two timescales, without the distortion given by considering as independent assemblies the same set of units detected at neighbouring temporal resolutions.”

R1.22 Fig. 3C Why do units 1 and 3 have the same plot?

In the previous version of Fig. 3C, the title with the name of the unit (e.g. “Unit 1” and “Unit 3”) referred to the graph below the text – so the two units did not have the same plot, but we accept that the layout was confusing. To avoid such confusion we now changed the layout of the figure, hoping for better interpretability.

Assembly B: units (1,2); lag = (0,1); $\Delta = 0.65$ sec

Assembly B: units (2,3); lag = (0,0); $\Delta = 0.82$ sec

Reviewer #2 (Remarks to the Author):

Integration of rate and temporal codes by hippocampal cell-assemblies supports theta phase coding of episodic information

Eleonora Russo, Nadine Becker, Aleks P. F. Domanski, Daniel Durstewitz, Matthew W. Jones

Summary

Russo et al. combine electrophysiological, behavioural and simulation approaches to characterise two different coding types expressed by hippocampal place cells: a rate code in the form of increased firing rate in specific regions of space, and a temporal code found in the phase-locking between individual spikes and local theta rhythm. While these two codes have already been demonstrated and extensively investigated, the authors evidence several potentially new findings: 1) the spikes of a single place cell can be phase-locked to different phases, and this is more likely to happen in different place fields (or different 'assemblies'); 2) locking to different phases can also take place in the same location if the ongoing task or future trajectory is different; 3) the amount of phase coding increases in trials involving memory of past trajectories. These findings will certainly be very relevant to researchers in the hippocampal /memory / plasticity field interested in network and cellular mechanisms. They raise interesting questions such as what are the mechanisms behind phase-locking, what are its roles if any, and whether individual place fields of the same cell are physiologically and functionally as different as place fields from different neurons.

Overall, the paper is concise and well-written, the methods are generally clearly explained and the combination of an experimental and modelling approach has a strong potential. The authors find that the same neuron can have multiple preferred phases, which is very interesting and, I believe, challenges some of the original models of phase precession. It is also interesting that the unsupervised approach used by the authors was able to find assemblies of two timescales that seem to correspond to those generally used in the field (place fields and theta sequences), providing an unbiased validation that these timescales are likely important for hippocampal information processing. However, it is unclear from the current analyses and literature review if the manuscript is not just providing a different way to look at previously-evidenced phenomena. I believe this can be clarified by adding a few more analyses or visualisations of the data as well as improving reports of past findings and their links with the authors' findings.

We thank the Reviewer for appreciating our work. We hope that the new data, analyses, and additions to the text helped improve the manuscript and allay the remaining doubts.

Major concerns

(R2.1) 1. I am wondering to what extent the findings of different phase-locking of the same neuron correspond to phenomena that have already been discovered. In theta sequences, early phase firing tends to encode past/current locations while later phases encode future ones (e.g. Wang et al., 2020 (<https://science.sciencemag.org/content/370/6513/247>); Kay et al., 2020 but see also earlier literature). In addition, theta sequences are known to 'chunk' the environment (Gupta et al., 2012, <https://www.nature.com/articles/nn.3138>), with a tendency to start and stop at maze turns or landmarks. Taken together, these phenomena will make it more likely that in the start of the track, future positions would be more strongly represented, thus expressing more spikes in the late phases of theta, while upon arrival at a maze turn or end, past positions will be more represented, thus firing more spikes in the early phase of theta. This has the potential to cause some of the 'trajectory/episodic' changes in theta phase locking evidenced in the manuscript. Could the authors add an analysis or discussion point

investigating to what extent these known properties of theta sequences explain their findings? A possible way to start answering this could be to look at the spatial distribution of phase preferences on the maze (separated by trajectory type) and see how these are affected by turns and landmarks.

We agree with the Reviewer that the findings of our manuscript are related to, and actually could explain, some of the experimental evidence described in previous literature. We also agree that testing whether space 'chunking' by theta sequences is induced by the activation of discrete depolarization settings is definitely a fascinating and relevant question that builds on our results and could be explored in future experiments. The proposed mechanism of phase modulation by different levels of depolarization, provided by dynamically activated assembly patterns could, in fact, be at the basis of the observations described in Gupta et al., 2012. Gupta and colleagues show how theta sequences tend to 'chunk' the environment and "reflect cognitive segments of the task". Wikenheiser and Redish (Wikenheiser and Redish, 2015) show how place cells encoding for the target location of the animal fire at the end of theta sequences even when the animal is far from such location. Both these observations are in line with the mechanism proposed in this manuscript of joint phase and rate modulation by the activation of (probably extra-hippocampal) cell assemblies. The simulated model of Fig. 6f shows how the activation of different rate assemblies (with rate assemblies we refer to groups of units with sustained temporal coordination lasting from hundreds of milliseconds to seconds) can set different stereotypical phase preferences of a same place cell. According to the model, units receiving a very low depolarization would tend to fire at the end of the theta sequence. As it is possible that units encoding for the goal location of a specific trial might receive a light depolarization during the whole trial, our model would then explain why these units fire even when the animal is far from the goal location and at the end of the theta sequence. Similarly, as spatial chunks of the environment reflect the cognitive structure of the task, it is possible that units involved in the same cognitive 'chunk' may experience a distributed top-down depolarization while the animal navigates a particular maze segment. This continuous depolarization could prompt these units to partake in the same theta sequence, effectively segmenting the maze into distinct cognitive chunks.

To better connect our findings with the previous literature, we have extensively revised the Discussion section and incorporated 19 additional references into the manuscript. For a comprehensive view of the changes made, we direct the Reviewer to the Discussion section. Here, we present only a few selected excerpts that are directly relevant to the points raised above.

At line 435 of the Discussion we added: "*In the dorsal CA1's deep sublayers, place cells can exhibit dual theta-phase firing preferences as the animal crosses the cell's place field (Fernández-Ruiz et al., 2017; Maurer et al., 2006; Wang et al., 2020). Here we show that changes in preferred firing phase can distinguish between distinct place fields or between different visits to overlapping locations on alternative routes, particularly under conditions that require active use of spatial memory and/or decision-making. These changes were fast and reversible, in line with the hypothesis that they were generated by the transient activation of different cell assemblies.*"

At line 747: "*The model confirmed that the activation of an assembly imposed an assembly-specific rate and phase preference on each assembly unit, thus producing, at the population level, assembly-specific theta sequences. Thus, while correlation between rate and phase changes has been observed during rate remapping (Sanders et al., 2019), our findings demonstrate that phase-shift coding extends beyond rate remapping and occurs also between distinct place fields or assemblies, frequently coinciding with*

changes in instantaneous firing rate, similar to those observed for splitter cells (Duvelle et al., 2023; Frank et al., 2000; Wood et al., 2000). The model predictions are also in line with recent work showing that individual place cells quickly switch between the encoding of alternative future locations or heading directions on alternate theta cycles, with lower firing rates for non-preferred directions occurring in later theta phases (Kay et al., 2020)."

And at line 521: *"In particular, as suggested by our theoretical model (cf. Fig. 6f), a low-level generalized depolarization of the cells encoding the current goal location could explain their spiking at the end of theta sequences, even when the animal is far from that location (Wikenheiser and Redish, 2015). Similarly, the cognitive segmentation of a task-induced by the presence of landmarks and corners within a maze, could induce a shared enhanced depolarization to all the cells involved in the same 'cognitive segment', also in those with place field far from the animal but within the traveled maze segment. This could give rise to the observed space chunking of hippocampal theta sequences (Gupta et al., 2012; Wikenheiser and Redish, 2015)."*

R2.2 2. On a related note, it is unclear which of the findings are new, for example, Wang et al., 2020 found that "a subset of neurons ("bimodal cells") displayed a bimodal relationship with theta phase", showing that the same neuron can potentially have different preferred phases. The authors could do a more in-depth analysis and report of the existing literature to clarify which of their findings are completely novel and which replicate previously-reported phenomena (see more references in minor comments).

We thank the reviewer for the suggestion, we now added reference to Wang et al., 2020 as well as to Fernández-Ruiz et al., 2017 and highlighted the connection with the previous literature on "bimodal cells".

At line 435 we added: *"In the dorsal CA1's deep sublayers, place cells can exhibit dual theta-phase firing preferences as the animal crosses the cell's place field (Fernández-Ruiz et al., 2017; Maurer et al., 2006; Wang et al., 2020). Here we show that changes in preferred firing phase can distinguish between distinct place fields or between different visits to overlapping locations on alternative routes, particularly under conditions that require active use of spatial memory and/or decision-making."*

At line 484 we added: *"The mechanism highlighted by our analysis and model here proposed mechanism is not the sole modulator of firing of CA1 cells, as the timing of specific inputs from Schaffer collaterals and the perforant path also plays a role in their phase of discharge (Fernández-Ruiz et al., 2017; Schomburg et al., 2014). Nevertheless, it captures the role played by various depolarization settings in inducing a modulation and discretization of phase preference of hippocampal CA1 place cells."*

We also included a more in-depth discussion of the literature throughout the introduction and discussion (see minor points).

R2.3 3. Not much is said in the manuscript about sharp-wave ripples and associated 'replay' sequences, however, replay and theta sequences are very different phenomena which should not be studied as one. Did the authors remove sharp-wave ripples to avoid contaminating their findings with replay? Did they speed-filter the data to remove low speeds where replay events are more likely to happen? Even if they only analysed times of high theta power, this could potentially still include 'exploratory' sharp

wave activity (O'Neill et al., 2006, <https://www.sciencedirect.com/science/article/pii/S089662730500961X>). Mixing up potential replay with theta sequences in the 'spike-assemblies' would mean that some of the findings (e.g. different phase for different assemblies) could be due to the different nature of such assemblies. I would recommend either removing or (ideally) analysing separately the assemblies corresponding to putative replay events (increased multiunit activity taking place during sharp-wave ripples at low-speed).

We thank the reviewer for the suggestion. We now added a new set of analyses aimed at understanding whether the observed contextual phase-shift arises from mixing replay-assemblies and theta-sequence assemblies together. To this aim, we: identified sharp-wave ripples (SWR); detected new assemblies excluding epochs associated with SWR, repeated all assembly- and single-unit analyses excluding spikes fired during SWR. Moreover, in all reported single-unit analyses, both those in the main text and those in the supplementary material, we also removed all spikes associated with an animal speed lower than 12 cm/sec.

Supplementary Fig 4, 5, 9, and 11 show the results of the new control analyses associated with Fig. 1b, 2c,d, 3b, and 5b,d. The control analyses confirmed our old results, thus excluding that contextual phase-shift could be due to the presence of unaccounted replay events.

We now added the following text.

At line 161: *“To understand whether the observed coordination at short timescales was related specifically to replay events, we repeated the assembly detection but excluding epochs in correspondence of SWRs. As shown in **Supplementary Fig. 4**, the temporal resolution of the detected assemblies was conserved, showing that SWRs were not the prevailing source of fast coordination under these conditions.”*

At line 175: *“See **Supplementary Fig. 5a** for the same analysis excluding spikes fired during SWR.”*

At line 185: *“. See **Supplementary Fig. 5b** for the same analysis excluding spikes fired during SWR.”*

At line 582: *“See **Supplementary Fig. 4** for the same analysis on assemblies detected excluding spikes fired during SWR;”*

At line 206: *“See **Supplementary Fig. 5c** for the same analysis excluding spikes fired during SWR”*

At line 313: *“These results withstood using different methods to identify place fields (**Supplementary Fig. 10**) and removing spikes fired during epochs of low theta-power or during SWR (**Supplementary Fig. 9**).”*

In the Method section at line 1158: **“Detection of SWR**

*SWR detection was based on the ‘Sleepwalker’ MATLAB toolbox (<https://gitlab.com/ubartsch/sleepwalker>). In brief, LFPs were down-sampled to 1000 Hz and 50 Hz notch filtered prior to band-pass filtering (using least squares filters) between 120-220 Hz. Candidate ripple events were identified based on threshold crossings in the z-scored 120-220 Hz power, using a threshold of 3.5 x SD of the signal. Start and finish times of the ripple were calculated based on 2 x SD of 120-220 Hz power. Representative samples of individual and averaged events were then visually inspected. Ripples were rejected if they were shorter than 50 ms or longer than 500 ms; if gaps of less than 50 ms occurred between events, they were treated as a single ripple. Ripple start/end timestamps were used to exclude SWR-associated spiking for some analyses (**Supplementary Fig. 4, 5, 9, 11**).”*

Finally, to further clarify that the detected assemblies are in line with known phenomena we added in the discussion 406: *“We detected place cell assemblies using an unsupervised algorithm able to extract coordinated activity from the data without pre-defining timescales of interest. This corroborated extensive evidence that hippocampal coding is characterized by two predominant time scales: a rate*

scale ('rate-assemblies') *reflecting place field firing rate modulation, and a sub-second temporal scale ('spike-assemblies') compatible with the entrainment of spikes by theta rhythms and during SWR-associated replay events. The relatively broad timescale characteristic of rate-assemblies suggests that their coordination is not solely imposed by the shared modulation of their composing units by the local theta rhythm, in agreement with their robustness to degradation of cholinergic signaling (Venditto et al., 2019). Nevertheless, spikes fired within either spike- and rate-assemblies coordinated with the theta rhythm, in line with previous literature reporting theta phase-locking of CA1 neurons when active in an assembly configuration (Buzsáki and Moser, 2013; Harris et al., 2003; Lopes-dos-Santos et al., 2013). We show that such theta locking is most pronounced for spike-assemblies, but present in rate-assemblies as well.*

And in the Discussion at line 429: *"This was true both for rate- and, more frequently, spike-assemblies, and was not induced by SWR-associated replay events."*

R2.4 4. Another potential concern is the relatively low number of simultaneously-recorded cells: ~130 cells recorded across 10 sessions, so about 13 cells per session on average. The maze used seems quite large with respect to average place field size, so it is unlikely that co-recorded place fields will homogeneously cover the maze, even with multiple fields. Could this sparse coverage interfere with the results? Specifically, could the rate assemblies discovered here be a result of inhomogeneous coverage of the maze by place fields, restricting the apparent co-firing of cells, while if the coverage was total, no or different assemblies would be discovered? Potential ways to alleviate this concern, apart from recording from larger ensembles, would include 1) comparing the results obtained for the session with lowest and highest cell yield (perhaps by highlighting these in the individual data plots) and/or 2) computing correlations between number of cells in a session and some of the parameters analysed and/or 3) show and compare the spatial distribution of the centroids and extent of detected cell assemblies, and that of the cumulated rate maps of all cells, for each session. If cell yield is found to impact the results, the authors should highlight this and discuss how it might impact the interpretation of the results

While we now have added more recording sessions and animals to our dataset to boost the analyses (see below), we would argue that the maze coverage of the recorded units does not confound the quantification of their contextual phase coding. Here we explain why:

For the analyses in the second part of the manuscript, all tests are done at the single-unit level and thus are not affected by how many units are recorded simultaneously (beyond, of course, having a larger sample size for the tests, but about this please see the last paragraph of the reply to this point).

Having more simultaneously recorded units could change the number and size of the detected assemblies. However, the detection of extra units in a session would not affect the assembly patterns already discovered, if not just by adding extra units to the already discovered assemblies. In fact, CAD builds assemblies through an agglomeration algorithm, which first tests the coordination between pairs of recorded units and then passes to larger assembly sizes by progressively testing (and, if significant, adding) other units of the recorded set. *In the algorithm, the addition of any new unit to the assembly does not modify the coordination of the units already present in the assembly.* Thus, if we had recorded more units and if some of them would coordinate with the already detected assemblies, this would not change the part of the assembly pattern associated with the already recorded units (neither the identity of the assembly units, nor the assembly activation pattern, nor the temporal resolution of the assembly). In the agglomeration algorithm, the only effect that adding a new unit can have on the pre-existent

assembly is to eventually reduce the activation instances of the assembly (for example if at some of the assembly activations the added unit misses to fire). This, would reduce the number of spikes included in the analysis but not change their phase preference.

We also want to stress that in any case, the addition of extra units would never interfere with the already detected assemblies, and thus would never lead to “no assemblies”, as the agglomeration algorithm can either include units coordinated with the preexisting assemblies or add no unit.

To clarify this point in the manuscript, we added a part to the Method section that explains the agglomeration algorithm in CAD. At line 1091 we added: *“The algorithm is based on a recursive agglomeration scheme, at each step of which it detects and tests assemblies of progressively larger size. As the first step of the algorithm, CAD scans the activity of all possible pairs of units in the recorded set to select the most common shared (potentially lagged) activation pattern, and test if this reoccurs more frequently than expected by chance. If the test is significant, a new time series is created to reflect the activation of the detected assembly-pattern. In the second step of the algorithm, the new assembly activation time series are tested in turn including the remaining recorded units. If the test is significant the unit becomes part of the assembly and a new assembly activation time series is generated. The algorithm stops when no more units can be added to the assemblies detected in the previous agglomeration step (see (Russo and Durstewitz, 2017) for a more detailed description of the algorithm). It follows that the hypothetical inclusion of supplementary units beyond those actually recorded would affect the number and size of the detected assemblies but it would not alter the coordination patterns already detected in the existing set. In this manuscript, the detection of paired assemblies was performed stopping the agglomeration at the initial pairwise step, while full-size assemblies were detected letting the algorithm agglomerate until completion.”*

In conclusion, in our eyes, the only concern about limited coverage is that it diminishes the number of detected assemblies and tested single-units and thus reduces the statistical power of the analyses. To increase our statistical power, in the revised manuscript we added one extra animal to the original five and included data from two extra sessions for each animal (for a total of 14 extra recording sessions). The new dataset now includes: six animals recorded in four sessions, for a total of 322 total recorded CA1 units (218 of which classified as putative place field units). All tests of the manuscript were re-run including the new data. The results confirmed what was presented in the previous version of the manuscript.

Finally, as suggested by the reviewer (suggestion 2), we computed correlation between the fraction of units per session changing phase in different assemblies (Fig. 3b) and the number of units recorded in that session and we found no significant correlation.

We now added at line 1192: *“Finally, we tested whether the number of units recorded in each session correlated with the fraction of units per session changing phase when firing in different assemblies. We found no significant correlation (Spearman’s correlation $r_s(34) = 0.25, p - value = 0.14$).”*

Minor concerns

R2.5 1. There are a few typos or grammatical mistakes that should be corrected (including in the figure legends).

We apologize for this and hope we have corrected any and all typos in the revised manuscript.

R2.6 2. The unsupervised assembly detection used by the authors uncovers mechanisms that have already been extensively known and studied (rate coding and time coding in hippocampus, place field scale and theta sequences scale). While the authors mention this (e.g. L124), they could attempt to quantify this link somehow, perhaps by comparing some previously published properties of place field and theta sequences to properties of the assemblies? Related to one of my main comments, they could also discuss and quantify to what extent spike assemblies relate more to replay or theta sequences.

We thank the Reviewer for the suggestion. Here is a list of some of the points of comparison between the properties of the detected assemblies and the hippocampal literature now discussed in the manuscript:

Comparison between timescales: In the manuscript we refer to the temporal scales of coordination of place fields, theta sequences, and replay events as broad ranges.

For rate coding we write “lasting from hundreds to thousands of milliseconds” at line 38: “Among pyramidal cells of hippocampal CA1, transient firing rate increases lasting from hundreds to thousands of milliseconds encode the position of an animal within the environment (‘place cells’ (Muller and Kubie, 1987; O’Keefe and Dostrovsky, 1971)), routes through paths with overlapping segments (“splitter cells” (Duvelle et al., 2023; Frank et al., 2000; Wood et al., 2000)), signal goal-locations (Hok et al., 2007), mark time intervals (Eichenbaum, 2014), respond to specific odors (Eichenbaum et al., 1987), sounds (Aronov et al., 2017), objects (Fried et al., 1997) and, in humans, to other people’s identities (Rey et al., 2020).”

While for theta sequences, and replay events we write “single- or multi-unit activity patterns with temporal precision on the order of tens of milliseconds” at line 52: “Theta phase precession (O’Keefe and Recce, 1993), theta sequences (Skaggs et al., 1996), and SWR-associated replay (Lee and Wilson, 2002) produce single- or multi-unit activity patterns with temporal precision on the order of tens of milliseconds.”

These timescales are compatible with those found by our assembly analysis. We referred to these time constants as a range instead of a more precise value as the values reported in the literature can vary from paper to paper as highly affected by: the method used to assess the temporal coordination (e.g. we correct for non-stationarity, which is often not done by other methods), the shape of the maze (elongated corridors or boxes), and the experience of the animal in performing the task (place fields can become longer through learning).

Characterization of rate assembly activation vs. place cells: We compared rate assembly properties with known place field properties by studying their activation in space (Fig. 1f,g and Supplementary Fig. 3), and by computing their spatial information content with respect to their composing place cell units.

Characterization of spike assemblies w.r.t. theta sequences and replay events: As discussed in the manuscript, the temporal scale of spike assemblies is compatible with that of theta sequences and replay events. While it is not the aim of the manuscript to focus on replay events, we now included a whole set of new analyses to establish whether the phase-shift properties observed in the manuscript are a byproduct of effects of sharp-wave ripple associated replay events. As reported at point R2.3, in the revised version of the manuscript we added extra analyses, figures (**Supplementary Fig. 4, 5, 9,**

11), and text to clarify how the detected assemblies and the phase-shift phenomenon relate to replay events (please, see point R2.3 for details).

R2.7 3. The authors use the term ‘episodic’ in the title and elsewhere but episodic memories generally have 3 components: what, where and when. The authors do not evidence a temporal (‘where’) coding aspect. I would recommend replacing ‘episodic’ throughout the manuscript and title by ‘task-related’ or a more accurate word.

We now change the title with a more accurate phrasing: “*Integration of rate and phase codes by hippocampal cell-assemblies supports flexible encoding of spatiotemporal context*”

R2.8 4. The introduction could be a bit less general and more focused. The discussion could also benefit from being more focused on matters directly related to the findings; in the present state, they both dwell quite a lot on theta sequences, which the authors did not directly analyse. I would recommend focusing more on the themes investigated in the paper: cell assemblies, phase locking mechanisms, within-cell variability of phase locking, what is known of the cellular mechanisms of phase locking and phase precession (ideally laying down different models or hypotheses). Some notions quickly mentioned in the results (e.g.L256 “Both experimental and theoretical studies have shown how a change in excitation received by a hippocampal unit can modify its phase of discharge”) could be developed in the introduction. The goal of the study could be generally clarified (especially sentence l64) – what were the hypotheses and predictions?

We have amended both the Introduction and Discussion to enhance clarity.

We have structured our Introduction in four parts: we introduced rate coding in the hippocampus, we introduced temporal coding, we briefly discuss the interrelationships between the two, and finally we introduced our analyses. We believe that all these elements are required to introduce the interaction between temporal and rate codes, and that removing any of them would weaken the narrative. These different facets are however now treated in more detail in the Discussion.

In the Introduction, we do briefly cover works on phase-rate relation in hippocampal units (at line 74: “*Mechanistically, interplay between fast somatic inhibition and slow dendritic depolarization as the animal crosses the respective neuron’s place field has been proposed as a possible mechanism linking firing rate with phase precession (Harris et al., 2002; Kamondi et al., 1998; Magee, 2001; Mehta et al., 2002) that may be tuned by local inhibitory interneurons (Grienberger et al., 2017).*”). The topic is then extensively broadened in the Discussion (from line 443 to line 493). We prefer to leave this in-depth description to the Discussion chapter, as we feel that it would be a bit too long for the introduction.

Finally, we rephrased the sentence previously at line l64 to explicit further the goal of the study, now at line 81: “*Here, we investigate and quantify the extent to which flexible and transient activation of place cell assemblies affects both firing rate and phase/temporal coding modalities of individual place cells, enabling the discrimination of different visits to the same locations under varied cognitive demands.*”

R2.9 5. The purpose and findings of the integrate-and-fire neuronal simulation could also be explained better. It would help if the authors explicitly stated different working hypotheses and the associated predictions that it is testing, and concluded on which one is supported. It is also unclear how this relates to the rest of the paper, given that the experimental findings focus on the difference of theta phase locking between different assemblies while the simulation attempts to explain how different assemblies

can lead to different sequences of activated cells. Perhaps the simulation should focus on explaining phase-locking results instead?

The integrate-and-fire model serves as proof of principle to test whether the activation of different assemblies, simulated by an assembly-specific and unit-specific depolarization provided to the assembly-units, can modulate the preferred phase of firing of the unit and generate assembly-specific theta sequences. Thus, the model directly relates to the findings presented in the previous part of the manuscript and attempts a mechanistic explanation of the phenomena.

The change in the firing phase of the assembly units could be qualitatively seen in the two insets. The change in theta sequences is a consequence of the change in the single unit preferred phase when seen at the population level. To make this clearer, we now added two phase histograms (on the right) showing how units 1 and 3 changed their phase preference when active in assemblies A and B

and added in the figure caption the text: “(Right) Phase histogram of the spikes of units 1 (blue) and 3 (green) when active in assemblies A (magenta) and B (red). Spike’s phases are computed with respect to the oscillatory modulation of g_{exc} . The change in assembly-specific depolarization provided to the simulated units leads to a change in the preferred phase of firing of the units (right) and, at the population level, in the order of unit activation along the theta cycle (inset).”

as well as in the main text at line 384 “This coordination was assembly-dependent and the simulated units changed their phase preference of firing, here computed with respect to the timecourse of the excitatory depolarization, when active in the two assemblies (Fig. 6f right).”

6. General figure comments

R2.10 a. Could the authors indicate the start location (e.g. with a ‘S’) in all relevant maze plots?

The start location of each trial changes according to the trial type. All 4 ends of the H maze are the start of the trial in different trial types. In the figures, we indicated with G1 and G2 the two locations where

guided trials start, and with C1 and C2 the two locations where choice trials start. We now added this information also in Fig. 4 because it was missing.

R2.11 b. Given the generally small sample size in plots ($n \sim 10$) the authors could use boxplots to describe the data instead of bar graphs, which rely on mean and standard deviation, parameters that are likely not accurately summarising samples. This is however not crucial as the authors already show individual data points.

We substituted the bar graphs with boxplots. Moreover, as each dot of the plots represents a per-session average and each session has a different number of contributing units, to the boxplots we also add a line marking the weighted mean (weighted by the number of units contributing to the test per session).

R2.12 c. Could the authors avoid using bright yellow in plots as it is hard to see?

Apologies. We now avoid using yellow to color code lines.

R2.13 d. It would help in some places if legends were added inside the plots instead of having to read the text legend (e.g. Figure 3c).

Thanks for the suggestion. We now updated the figure accordingly:

R2.14 e. Could the authors show the actual position data of the animal on the example spike plots, when applicable (e.g. Figure 5)?

Our figures of example spike plots display the animal's positions in the top-left inset. Each dot corresponds to the position of the animal at the time the spikes (indicated in the rate/phase plot) were fired.

R2.15 f. In some of the plots showing individual data points there seems to be less points than the indicated n . This is probably because points are fully overlapping. Could the authors fix this (e.g Fig 2b)?

We apologize for this and we thank the reviewer for noticing it. In the new boxplots we paid attention not to have overlapping points. As a note, we want to add here that, despite not having overlapping points, some boxplots show fewer dots than the total number of sessions. This is because not all units met the inclusion criteria imposed by some of the analyses, e.g. on the number of place fields, phase locking in different place fields, number of spikes fired in each place field, etc ... (this information is detailed in the method section). In case no unit of the session meets all the required selection criteria,

the session is not included in the analysis and the dot is not added to the plot. This is of course accounted for in the statistical tests.

7. Assembly-related questions

R2.16 a. How are the “assembly activity” and “assembly place fields” computed? From the examples provided (in Fig 3 or fig s3), they don’t seem to correspond to a simple sum of individual rate maps. Could the authors explain this early in the results? e.g. mention of ‘place field’ of assemblies L 121 (same question for sup fig 2). Note for Fig S2, what is shown seem to be rate maps, not place fields (place fields would be a selected subset of the rate map).

We thank the reviewer for noticing this. What is shown in e.g. Fig S2 are indeed rate maps (or activation maps, in case of assemblies) and not place fields. We now modified the wording in the text and figure legends. In line 150 “... (Figs. 1d, e) and **activation maps** (Figs. 1f, g) for both assembly groups...”, at line 591: “**Example of spike- (f) and rate- (g) assembly **activation maps** (activity normalized to 1).**”, at line 1388 “**Supplementary Fig. 3 | **Activation maps** of assemblies and their composing units. Example of **activation maps** of a spike- (a) and a rate- (b) assembly and of its composing units**”

We also now included a description of how the assembly activity is computed. At line 146 we added: “**Assemblies are considered active whenever all units composing the assembly fire spikes matching the assembly activation pattern identified by the algorithm. The assembly is considered to have an activation of n , when all units composing the assembly fire at least n spikes in the bins matching the assembly pattern.**”

R2.17 b. What is the unit of the assembly activity plots (e.g. Fig 1f,g “normalised assembly activity”)?

We now clarified how assembly activity is computed (reported in the previous point R2.16). We added information on the axes of Fig. 3c, and changed the text in the caption of Fig. 1: at line 591 “(f, g) **Example of spike- (f) and rate- (g) assembly **activation maps** (activity normalized to 1).**”

R2.18 c. In Figure 1 L449 “the color scale shows the lag between the activation of each assembly-unit with respect to the unit first active in the assembly.” Could the authors clarify this?

Now added at line 584: “... *the color scale shows the lag between the activation of each assembly-unit with respect to the unit first active in the assembly. **Units marked in dark blue (lag of 0) are the first to activate within the assembly, units marked in dark red the last (2 bins after the activation of the first assembly unit).***”

R2.19 d. What are the minimum and maximum sizes of assemblies, in terms of number of cells, if any?

Indication about the size of the assemblies is now reported at line 128: “...*the temporal precision of hippocampal assemblies active during the task ranged from milliseconds to seconds **and is bimodally distributed** (Hartigans Dip Test, n . bootstrap samples = 10^5 , $dip=0.03$, $p = 0$) into two major groups. We found: (1) **137 sharp spike patterns involving on average about 17% units per session per pattern (with a maximum of a 3-unit assembly in a 20-unit set) and a temporal precision in the range of 0.006 - 0.06 sec centered around 0.028 sec (spike-assemblies) and (2) 204 broader firing rate patterns with on average 28% units per session per pattern (with a maximum of an 11-unit assembly in a 19-unit set) and temporal precision between 0.07 - 5 sec (rate-assemblies) (Fig. 1b).***”

R2.20 Could the authors provide a justification for the chosen duration limits (5 msec and 5 sec)?

These rather inclusive limits were chosen to ensure we do not miss important events, based on our previous experience in assembly analysis on other datasets (Russo and Durstewitz, 2017, Oettl et al. 2020, Londei et al. 2023). In other datasets (recorded in CA1, anterior cingulate cortex, entorhinal cortex, ventral striatum, ventral tegmental area, and zona incerta), we found practically no assemblies with coordination at precisions higher than 5 ms. On the other side of the spectrum, we chose a limit big enough to cover in about one bin the time needed by the rat to cross a place field.

We now added in the Method section 1128: *“Values for tested bin sizes were selected based on previous experience in assembly analysis (Londei et al., 2023; Oettl et al., 2020; Russo and Durstewitz, 2017). The lower limit was chosen because in CA1 we found practically no assemblies at coordination precision higher than 5 ms (Russo and Durstewitz, 2017). The upper limit was chosen to cover in about one bin the time needed by the rat to cross a place field.”*

R2.18 Could the authors clarify what “consistently reoccurring patterns” mean - is it enough for the same group of cells to co-activate within a given time-window to be considered an assembly, or do they have to co-activate in a specific temporal pattern, i.e., order?

We now rephrased the sentence to (line 124): *“Here we thus refer to a ‘functional cell assembly’ as any group of units whose activation coordinates with temporal precision between 5 msec and 5 sec, and with arbitrary time lags between the unit activations, in a consistently reoccurring pattern.”* which should clarify the question together with the added text on assembly activation at lines 146: *“Assemblies are considered active whenever all units composing the assembly fire spikes matching the assembly activation pattern identified by the algorithm. The assembly is considered to have an activation of n , when all units composing the assembly fire at least n spikes in the bins matching the assembly pattern.”*

R2.21 e. Figure 2a, unit 3 seems to be firing continuously like an interneuron. However, the authors mentioned that they only analyse place cells. Can the authors explain this apparent discrepancy? See also my later comment on only analysing putative pyramidal cells.

Thanks for spotting this. This was indeed a plot from a previous version of the manuscript where all units were included in the assembly analysis. We now updated the figure.

R2.22 f. Bimodality of assembly sizes: L 107 “strongly bimodal distribution”: the authors should provide the results of a statistical test of bimodality (e.g. Hartigan's Dip Test?) or otherwise mention instead that the distribution appears / seems bimodal.

We now performed a Hartigan's Dip Test and changed the text accordingly, at line 128: *“...the temporal precision of hippocampal assemblies active during the task ranged from milliseconds to seconds and is bimodally distributed (Hartigan's Dip Test, n . bootstrap samples = 10^5 , dip = 0.03, $p = 0$) into two major groups.”*

R2.23 g. Could the author show the spatial distribution of both assembly types (ie, where do they occur) and comment on any biases in the spatial distribution?

The spatial distribution of the activation of specific assemblies is displayed in Fig. 1f,g and in Supplementary Fig. 4.

We computed also the average activation map of all detected spike and rate assemblies normalized by the activation maps of the recorded single unit. In particular: we first computed the activation maps of each detected assembly and normalized it so that the sum of the activation over the whole maze is one.

We repeated the same procedure for the activation map of single place cells. Then, we averaged the normalized activation maps across all spike assemblies and rate assemblies, separately, and divided them by the average unit activation map. Below are the resulting plots:

(Note, here we manually adjusted the color bars to improve visibility). Even by pushing the color bars to highlight contrasts, no particular trend can be seen in the two distributions (besides possibly a trend of rate-assemblies to cover the central arm), suggesting a balanced interplay between assembly and unit activation maps. While we cannot rule out that further, more extensive analyses could highlight interesting points in this regard, we feel that their inclusion would diverge from this manuscript's core focus. Therefore, we prefer to defer this investigation to subsequent studies.

R2.24 8. The dimensions of the maze should be indicated in methods and the median number of place cells per session as well as lowest and highest numbers should be indicated in results (I90).

In the method section, at line 1042, we now added: "*Maze dimensions were 170 x 130 cm.*", while in the result section, at line 111, it is indicated: "*The following analyses focus exclusively on the remaining 218 units identified as putative place cells (with a median of 7, a minimum of 2, and a maximum of 20 putative place cells per session).*"

R2.25 9. One might wonder to what extent spike-sorting errors (in particular, mistakenly combining two clusters that belong to different cells) could lead to the results about different preferred phases in different place fields. The authors could show that cluster quality measures are not correlated with percentage of cells having mixed phases. See for example Maurer et al., 2006.

Thanks for the suggested analysis. We computed a Kruskal-Wallis test and found that the L-ratio of units with phase changes (either for different place fields or within the same place field but for different trials) was not significantly different from that of the rest of place cells (Kruskal-Wallis Test: $\chi^2(1) = 1.2$, $p = 0.3$). The test was computed on 62 units from 10 recording sessions (unfortunately, L-ratio information was not available for all recorded sessions).

We now added this information in line 280: "*This result could not be explained by covariates such as the animal speed and the ongoing theta power (...) nor the sorting cluster quality of the units (generalized linear mixed-effects model of a unit to phase-shift, either for different place fields or within the same place field but different trials, based on its L-ratio, recording session and animal; with binary dependent variable for significant phase change and logit link function: $F(1,60) = 0.12$, $p = 0.74$).*"

R2.26 10. Can the authors clarify in methods if, and if yes, how they checked if separate sessions recorded from the same animal might contain the same cells?

Tetrode positions were adjusted by +/- 20-60µm between recording days. Although some cells might in principle be recorded on multiple sessions, it is extremely difficult to categorically prove this one way or the other without continuous recording. In common with the vast majority of papers in the field, we did not attempt to establish cell identity across sessions and used all recorded units identified as putative place cells. To account for the non-independence of different sessions of the same animal we now substitute many of the previous statistical tests with general (or generalized) linear mixed-effects models, where animal and session identity is accounted for.

At line 1084 of the Methods we now wrote: *“Because of the dependence introduced in the data by pooling units from multiple sessions of the same animals, when appropriate we perform statistical tests by generalized linear mixed-effects models, where we explicitly account for session and rat identity.”*

R2.27 11. Are errors trials included in the analysis? If the authors could analyse error trials and show, for example, a decrease of phase coding in these it would support the idea that phase coding is relevant for behaviour (one of the most interesting findings in my opinion).

Error trials have been excluded. We initially thought of separately analyzing error trials, but at this stage of training the animals performed a median of 83% correct choice trials, and error trials were too few to build sufficiently-powered analysis.

R2.28 12. Figure 3:

a. If the goal of figure 3c is to show task-dependent assemblies, it would be more convincing to show an example with cells active in the common, central stem, with a different activity for choice vs guided trials for the same place and movement direction. The currently-showed cells just seem to be directional place cells.

The goal of Fig. 3c is to show that the shift in the preferred phase of firing of unit 2 when joining assembly A or B (shown in Fig. 3a) also matches a change in the information encoded by the unit when active in the two assemblies. Whether the activation occurs in the same maze location or not is a topic extensively analyzed in the second part of the manuscript through single-cell analysis. At this point of the manuscript, we are focusing on the relation between unit phase, assembly-membership, and information encoded. Also, we have to consider that choice vs. guided trials run in different directions by task design. Choice trials start from what indicated in the maze as C1 and C2 and end in G1 or G2, while guided trials start from G1 and G2 and end in C1 or C2. Thus, there can not be cells active in the common, central stem, with a different activity for choice vs guided trials but for the same movement direction. The newly selected example shows two assemblies sharing a unit and differentiating between choice left vs. right trials.

R2.29 b. More generally, could the authors clarify what is the general message of panel c and what this means: “The change in firing phase of unit 2 when active within assembly A and B (a) is associated with a change in encoded information of the two assemblies (c)”

We now explained better the general message of Fig. 3c (see previous point, R2.28) changing the sentence into (line 646): *“Unit 2 takes part in both assembly A and B. When firing in the two assemblies, the unit fires preferentially at two different firing phases (a) to encode different task-related information (left vs. right choice trials).”*

R2.30 c. What is the unit of the yaxis (assembly activity) and what is the shaded area around the line?

We thank the Reviewer for noticing this. Assembly activity is defined in the manuscript at line 148: “*The assembly is considered to have an activation of n , when all units composing the assembly fire at least n spikes in the bins matching the assembly pattern.*”. We now modified the yaxis and added the information in the figure legend: “(c) *Mean and SE activity along the maze of the two assemblies displayed in (a) (top) and their composing units during different trial types (bottom).*”

R2.31 13. To provide some explanation of the reason for different phases expressed by the same cell in different assemblies, the authors could investigate whether the cells belonging to a given assembly have more similar phases, i.e. is the phase somehow driven by the assembly? In other words, it would be interesting to compare the within-cell and across-cell as well as within assembly and across-assembly variability in phase preference.

To provide a hypothesis on the mechanism for different phases expressed by the same cell in different assemblies we build an adaptive exponential integrate-and-fire model simulating the effect of the activation of different assemblies. The model is presented in section “Phase-shift leads to context-dependent fine temporal coordination among units” and Fig. 6f. According to our hypothesis the characteristic phase of a unit when firing in an assembly is connected to the assembly-specific degree of depolarization provided to the unit by the activation of the assembly. This is specific to the inputs arriving to the unit when the assembly is active, it is thus assembly-specific but also unit-specific, therefore we do not expect that different units taking part in the same assembly share the same preferred phase.

To clarify that the proposed model indeed aims to provide a possible explanation of why different phases are expressed by the same cell in different assemblies we added new tests and new plots in Fig. 6f (please, see point R2.9).

R2.32 14. Figure 4: Could the plots in b be chosen to correspond to the place cell example given in a? We now updated Fig. 4 so that panels a) and b) (top) correspond to the same unit.

15. Figure 5:

R2.33 a. Could the authors indicate what the grey color represent in the rate-phase plots (I assume, out of field or other fields' spikes)?

Yes, thanks for noticing this missing information. Now added to the legend, line 669: “*Grey dots are spikes fired out of the two tested place fields.*”

R2.34 b. “the degree of differentiation of the unit phase is maximal” where is this shown or tested?

We now added this information in line 300: “*Interestingly, in this example, the degree of differentiation of the unit phase is maximal when the animal has to actively remember its previous path to inform its next turn, and is absent in the guided trials when no active choice has to be made and the path covered from trial onset is identical (with a difference of 2.8 rad between the average spiking phase of the left and right choice trials, and of 0.2 rad between the two guided trials types).*”

R2.35 16. L177 “Place-cell firing phase can encode distinct place fields” It is unclear what 'encoding a place field' means. Perhaps the authors mean "Distinct place fields can express different firing phases"?

To maintain alignment with the title of the next paragraph (“*The firing phase of place-cells can encode distinct task-related information within the same place field*”), we changed the title to “*The theta firing phase of place cells can discriminate between distinct place fields of the same unit*”.

R2.36 17. To properly demonstrate that phase of spikes 'encodes' trial type and not other, covarying, parameters, the authors should incorporate information about possible confounding factors, such as speed and movement direction in the SVM classifier (if this is possible with this type of analysis). It could also include distance to the closest maze end /turn to address one of my main comments.

In line with this suggestion, to assess whether the observed encoding of trial information by spiking phase could merely be explained by confounding factors such as speed and theta power (movement direction is already accounted for in the differentiation between types and discussed in the paragraph associated with Fig. 5d), we tested whether adding phase information improved the accuracy of a classifier trained on speed and theta power. The test confirmed that incorporating phase information provided information for decoding the trial type supplementary to that already present in the two covariate factors.

Reference to this control test is now reported at line 276: *"This observation was corroborated by training a support vector machine (SVM) classifier on the phase of spikes fired in an individual place field to distinguish between trial types. We found that for 32% of units at least two trial types could be distinguished above chance level within at least one of the unit place fields (see Methods for details). This result could not be explained by covariates such as the animal speed and the ongoing theta power (a general linear mixed-effects model of the accuracy of an SVM classifier trained on speed and theta power or on speed, theta power, and theta phase of each spike found a significant increase in accuracy for the latter classifier. Contrast tests between the two classifier types: $F(1,530) = 8.9, p = 0.003$. Test computed on the subset of place fields which could distinguish above chance trial identity in the latter, more powerful, classifier)"*

R2.37 18. The authors could cite the recent findings of Tang et al., 2021 (<https://elifesciences.org/articles/66227>) in the discussion when mentioning prefrontal-hippocampal assemblies as well as context-dependent theta sequences (I390).

We added the citation in line 515: *"One hypothesis is that during the early stages of learning, inter-regional assemblies (e.g. prefrontal-hippocampal assemblies (Benchenane et al., 2010; Domanski et al., 2023; Eichenbaum, 2017; Ito et al., 2015; Schmidt et al., 2019; Tang et al., 2021) or medial-septum-hippocampal assemblies (Aoki et al., 2019)), recruited at each theta cycle (Jezek et al., 2011), modulate the depolarization of hippocampal units thereby dynamically producing goal-dependent cycling activity."* and in line 508: *"This could contribute to the formation of context-dependent theta sequences (Tang et al., 2021), supporting the formation of episodic memories and planning (Kay et al., 2020)."*

R2.38 19. It seems that the Benjamini–Hochberg correction relies on defining a false discovery rate. Have the authors defined this somewhere?

The Benjamini–Hochberg correction controls for false discovery rate fixing it at α (Benjamini and Hochberg, 1995, *Journal of the Royal Statistical Society: Series B*). The α value was indicated in the Method section, we now added the information also in the main text every time we mention Benjamini–Hochberg correction.

R2.39 20. The authors compare properties of firing within assemblies and out of assemblies. They could comment on whether these differences, for rate assemblies, relate to observed differences between in-field and out-of-field firing in past papers (see for example Molter et al., 2012 <https://www.sciencedirect.com/science/article/pii/S0896627312006228>; Grienberger et al., 2017 <https://www.nature.com/articles/nn.4486>).

The manuscript presents a comparison between the phase locking of spikes emitted by a unit within an assembly and the unit's overall firing activity. This manuscript's findings suggest that the overall spiking of individual units comprises modes induced by the activation of specific assemblies. Consequently, the observed higher phase locking in assembly spikes is likely attributable to testing phase locking in a more uniform firing mode, as opposed to a multimodal distribution. These results thus do not necessarily indicate that spikes fired at a low firing rate, that is outside of the unit's place fields (or out of assemblies), are different in nature than those fired within assemblies.

We now clarify better this point in the Discussion at line 419: "*We show that such theta locking is most pronounced for spike-assemblies, but present in rate-assemblies as well. A possible explanation for this enhanced phase locking is that isolating spikes fired within an assembly configuration effectively separates a specific mode from the otherwise multimodal phase distribution of the unit firing. In fact, the enhanced phase modulation of hippocampal units during assembly activations also revealed that units taking part in multiple assemblies changed their preferred spiking phase according to the assembly active at the time.*"

In Grienberger et al., 2017 the authors highlight how a uniform inhibition serves the function of enhancing spatial information of place cells, limiting out of place-field spikes. In the paper the authors also show how a modulation in inhibition affects the unit phase of firing. In this respect we now cite the paper at line 74: "*Mechanistically, interplay between fast somatic inhibition and slow dendritic depolarization as the animal crosses the respective neuron's place field has been proposed as a possible mechanism linking firing rate with phase precession (Harris et al., 2002; Kamondi et al., 1998; Magee, 2001; Mehta et al., 2002), that may be tuned by local inhibitory interneurons (Grienberger et al., 2017).*"

At line 331: "*Both experimental and theoretical studies have shown how a change in excitation received by a hippocampal unit can modify its phase of discharge (Grienberger et al., 2017; Harris et al., 2002; Kamondi et al., 1998; Losonczy et al., 2010; Magee, 2001; Mehta et al., 2002).*"

And at line 452: "*Spatially uniform inhibitory conductance has been shown to enhance the range of phase precession (Grienberger et al., 2017).*"

Finally, Molter et al. perform a second-order spectral analysis of the LFP and focus on oscillatory modulations of hippocampal theta power. While very interesting, we find that discussing this paper would lead the discussion a bit astray from the already complex topic tackled in our manuscript.

R2.40 21. ~L719 please explain how were the rewards provided / delivered, was this automatic or done by the experimenter, could the rat see the experimenter preparing for delivery?

Apologies for this omission, now explained in 'Training' subsection of Methods, line 1051:

"...where a reward (0.1 ml of 20% sucrose solution in water) was delivered remotely through tubing connecting reward wells to syringe pumps located in the adjacent room, where the experimenter sat."

R2.41 22. Place cell selection: were only putative pyramidal cells selected? If not, this should be added as a criteria (otherwise signals from fibers or low-firing interneurons could be wrongly categorised as hippocampal place cells). In addition, and only as a comment, it is becoming more common to have as an additional criteria a spatial information shuffle to select place cells, i.e. the spatial information of a place cell should be higher than the 95th percentile of the distribution of shuffled data from that same cell (spike train shuffle). This ensures that only cells with some degree of reproducibility of their spatial firing are included. I would recommend that the authors do this at least in future place cell studies.

We thank the reviewer for this suggestion which we will certainly apply in our next analyses. To answer the first question: yes, we included only putative pyramidal cells in the analyses of the paper. This information can be found at the beginning of the result section, line 109: “Among these, we isolated putative place cells by selecting units with a mean firing rate between 0.2 Hz and 4 Hz and with spatial information above 0.5 bits/s (Skaggs et al., 1993). The following analyses focus exclusively on the remaining 218 units identified as putative place cells (with a median of 7, a minimum of 2, and a maximum of 20 putative place cells per session).”

As well as in the method section, line 1081: “Our analysis focused on putative place cells. To select putative place cells we restrict to units with firing rate between 0.2 Hz and 4 Hz and with spatial information above 0.5 bit/s on the maze (Skaggs et al., 1993).”

R2.42 23. The place field detection method (L831+) used by the authors seems new and relatively complex. If it has been used before, the authors should cite the relevant references. To help the reader get a good intuition of what the method does, the authors could add a supplementary figure explaining the method and showing the detected place field contours for a selection of example cells, with spike plots and rate maps. Ideally this figure would also show examples from a more classical field detection method [e.g., for a similar maze, Wirtshafter & Wilson, 2020; <https://elifesciences.org/articles/55252#s4-4-3>]. I note that Figure 4 shows one example but without a visualisation of the place field extent and without comparison to other place field detection methods.

As Reviewer #2 correctly pointed out, the method used for place field identification is based on the spatial density of spikes, irrespective of the firing rate of the unit at the spike time. The method was developed by us for the analyses of this manuscript. To explain why we could not use the more common approach used in Wirtshafter & Wilson, and make an explicit comparison of the output of the two methods, we now extended the section related to place field identification in the Methods.

Importantly, to test the robustness of our results against different methods for place field detection, we also repeated the analysis of Fig 5b and 5d using place fields identified with different methodologies (new Supplementary Fig. 8, 9, 10, 11). The new analyses showed that changing the clustering technique did not change the results presented in the paper.

Now added in the method section ad line 1202:

“Isolation of place fields. Place fields were established only for units with spatial information above 0.5 (place cells) and with phase modulated spikes. Spikes of each unit were divided into different clusters (place fields) on the basis of their place of firing. Place fields are often identified as a region of connected bins in a unit's rate map that surpasses a fixed threshold in firing rate. For example, for a maze comparable to the one used in this manuscript, Wirtshafter and Wilson (Wirtshafter and Wilson, 2020) use a threshold of a standard deviation over the mean firing rate of the unit. In units with multiple place fields, such a high threshold leads to a selection among the fields, at the expense of those with lower rates which remain undetected (c.f. Supplementary Fig. 7). While this is typically not an issue, this study aims to investigate the changes in phase and rate a unit exhibits across different place fields and trial types. Therefore, it is here important to capture a larger variety of fields. We thus explored different techniques to detect place fields and tested the robustness of the analyses in Fig. 5b and 5d.

The compared techniques were:

Rate Map Wirtshafter and Wilson (RM WW)

In Wirtshafter and Wilson (Wirtshafter and Wilson, 2020) place fields are detected by: 1) computing a rate map of firing per occupancy binned with a 2 cm grid and smoothed with a 10 cm standard deviation Gaussian kernel. Only epochs in which the rat moved faster than 12 cm/s were included; 2) thresholding the rate map at $\theta_{RMWW}^1 = \mu(fr) + \sigma(fr)$, with $\mu(fr)$ and $\sigma(fr)$ rate mean and standard deviation, respectively; 3) only fields with at least one bin with rate above $\theta_{RMWW}^2 = \mu(fr) + 2 \cdot \sigma(fr)$ were selected; 4) fields of length less than 15 cm were discarded.

Rate Map (RM)

To detect also place fields with lower firing rates we repeated the procedure described in 'RM WW' but using a $\theta_{RM}^1 = 0.7 \cdot \mu(fr)$ and omitting point 3.

We also included methods that aimed to group spikes according to the density of their spatial clustering:

Rate Map + GMM (RM + GMM)

We modeled the place fields of a unit with a Gaussian Mixture model. In the first step, we proceeded as described in 'RM' but with a harsher threshold of $\theta_{RMGMM}^1 = \mu(fr)$. This allowed us to establish the overall number of place fields and obtain a first estimation of cluster memberships. In the second step, this first clustering was then used as initial condition for the estimation of a Gaussian mixture model (function `fitgmdist`, `matlab`). To train the model we only used the spikes identified as place field members in the first step. Once obtained the model we used it to cluster (function `cluster`, `matlab`) all spikes fired by the unit. Spikes with low probability of being part of any of the identified clusters (i.e. with logarithm of the estimated probability density function smaller than $\log pdf < -20$) were discarded as not assigned to any of the modeled Gaussians (place fields).

DBSCAN

Spikes were clustered with a density-based spatial clustering algorithm (DBSCAN (Ester et al., 1996), `matlab` function `dbscan` with parameters $\epsilon = 0.05$ and `MinPts` = 15). Spikes identified as outliers by the algorithm were discarded.

DBSCAN + GMM

We proceeded as in 'RM + GMM' but used the output of 'DBSCAN' as the initial place field estimate.

Common to all tested place field detection methods, only spikes fired when the rat moved faster than 12 cm/s were included.

Supplementary Fig. 7 shows a comparison between the outputs of the different place field detection methods. Key analyses were repeated on unit's activity parsed by place field computed with different detection methods (Supplementary Fig. 8, 9, 10, 11, and text)."

Finally, Fig. 4a shows a comparison between the unit activation map and the clustering of spikes in place fields. By identifying place fields by clusters of spikes, the applied method does not strictly define a spatial contour that we can report in the activation map. We now added Supplementary Fig. 7, to show a comparison between the output of the "DBSCAN + GMM" algorithm and other place field detection methods.

R2.43 In addition, as the methods seems to rely only on spike numbers but not firing rate, can the authors explain how they deal with spurious regions of increased spike count due to the animal spending more time there?

As now explained in the related Methods section (see point R2.42) we excluded all spikes fired when the animal had a speed lower than 12 cm/sec. This mitigates the problem for the 'DBSCAN' and 'DBSCAN + GMM' methods (while 'RM' and 'RM + GM' are directly rate-based). Moreover, the single

unit analyses have now been repeated also for firing-rate based place field detection methods (Supplementary Fig. 8, 9, 10, 11).

R2.44 24. L841/ L865: can the authors remind the reader of what is considered a trial category/type?

We now added, line 1260: “For each place field we trained a support vector machine (SVM) classifier with linear kernel (slack variables minimized with L1 norm and box constraint = 1) to divide trials according to their trial type (*the trial types categories here considered were: correct choice left, correct choice right, forced left, forced right, forced switch right-left, forced switch left-right*).”

R2.45 25. As mentioned in main comments, the authors could generally link better their findings to the existing literature, e.g. in the discussion. Here are a few papers that seem related to the manuscript and that the authors might want to cite and discuss:

We thank the reviewer for this literature suggestion which we incorporated in several points of the manuscript:

- O’Keefe & Burgess, 2005 <https://onlinelibrary.wiley.com/doi/abs/10.1002/hipo.20115>

We now cite the paper at line 69: “Despite their different timescales, the *information content and processes governing hippocampal rate and temporal coding are not independent (O’Keefe and Burgess, 2005)*.”

- Terada et al., 2017 <https://www.sciencedirect.com/science/article/pii/S0896627317304622>

We now cite the paper at line 62: “In addition to spatial information, recent studies have uncovered theta sequences reflecting *sequences of events (Terada et al., 2017)*, current goals (Johnson and Redish, 2007; Wikenheiser and Redish, 2015), and hypothetical future experiences (Kay et al., 2020), suggesting contributions of hippocampal temporal coding to planning and speculation.”

And at line 71: “Firstly, the order in which units activate during theta cycles and replays *typically* reflects the sequences in which place fields are crossed by the animal during exploration (but see also (Stella et al., 2019)), *or sequences in which sensory cues are encountered (Terada et al., 2017)*.”

- Wu & Yamaguchi, 2010 <https://link.springer.com/article/10.1007/s00422-009-0359-9>

We now cite the paper at line 455: “This correlation is, however, lost as the rate peak is passed and the animal leaves the cell’s place field (Huxter et al., 2003; *Wu and Yamaguchi, 2010*).”

- Venditto et al., 2019 <https://onlinelibrary.wiley.com/doi/full/10.1002/hipo.23100>

We now cite the paper at line 412: “*The relatively broad timescale characteristic of rate-assemblies suggests that their coordination is not solely imposed by the shared modulation of their composing units by the local theta rhythm, in agreement with their robustness to degradation of cholinergic signaling (Venditto et al., 2019)*.”

- Grienberger et al., 2017 <https://www.nature.com/articles/nn.4486>

We now cite the paper at line 74: “*Mechanistically, interplay between fast somatic inhibition and slow dendritic depolarization as the animal crosses the respective neuron’s place field has been proposed as a possible mechanism linking firing rate with phase precession (Harris et al., 2002; Kamondi et al., 1998; Magee, 2001; Mehta et al., 2002), that may be tuned by local inhibitory interneurons (Grienberger et al., 2017)*.”

At line 331: “Both experimental and theoretical studies have shown how a change in excitation received by a hippocampal unit can modify its phase of discharge (Grienberger et al., 2017; Harris et al., 2002; Kamondi et al., 1998; Losonczy et al., 2010; Magee, 2001; Mehta et al., 2002).”

And at line 452: “Spatially uniform inhibitory conductance has been shown to enhance the range of phase precession (Grienberger et al., 2017).”

- Yu & Frank, 2020 (preprint) <https://www.biorxiv.org/content/10.1101/2020.11.23.395012v1.full>

We cite the paper (now published in Plos Biology) at line 66: “There is also growing evidence that spikes fired during different relative phases of local theta cycles may encode different aspects of past and future experiences (Wang et al., 2020; Yu and Frank, 2021).”

- Schmidt et al., 2009 <https://www.jneurosci.org/content/29/42/13232.full> (Isolated spiking was also more tightly phase locked to theta compared with adjacent spiking)

We now cite the paper at line 59: “...during phase precession the position of the animal within a cell’s place field correlates with theta phase of that cell’s spikes (O’Keefe and Recce, 1993; Schmidt et al., 2009)...”

- Maurer et al., 2006 <https://onlinelibrary.wiley.com/doi/abs/10.1002/hipo.20202> specifically: “the cell fires with spikes clustered at two different phases over the theta cycles in which the fields overlap”

We now cite the paper at line 435: “In the dorsal CA1’s deep sublayers, place cells can exhibit dual theta-phase firing preferences as the animal crosses the cell’s place field (Fernández-Ruiz et al., 2017; Maurer et al., 2006; Wang et al., 2020).”

- Feng et al., 2015 (<https://www.jneurosci.org/content/35/12/4890>)

We now cite the paper at line 494: “The fine temporal coordination of unit activities imposed by phase precession is commensurate with the induction of plasticity mechanisms for binding episodic information that, otherwise, would be separated by seconds (Feng et al., 2015).”

R2.46 - The authors cite Kay et al., 2020, but they should more explicitly say how these findings are related to the present findings. See from Kay et al.: “we observed equivalent theta phase coding for additional representational firing patterns in the hippocampus: inbound path coding [...] and extrafield firing”

We now further discuss the relationship between Kay et al. 2020 and our paper in the Discussion, at line 481: “The model predictions are also in line with recent work showing that individual place cells quickly switch between the encoding of alternative future locations or heading directions on alternate theta cycles, with lower firing rates for non-preferred directions occurring in later theta phases (Kay et al., 2020).”

R2.47 - The authors do not explicitly mention trajectory-dependent cells (‘splitter cells’) while several paragraphs seem related to these. They should mention at least the original discoveries (Wood et al., 2000 <https://www.sciencedirect.com/science/article/pii/S0896627300000714>; Frank et al., 2000 <https://www.sciencedirect.com/science/article/pii/S0896627300000180>) L205 and possibly discuss in

what way the findings presented here could be related to what is already known about splitter cell activity (e.g. has it been found that different splitter fields have different preferred phases?).

Thanks for pointing this out. Indeed, we unintentionally omitted reference to 'splitter cells'. The suggested papers are now referenced at line 38: "*Among pyramidal cells of hippocampal CA1, transient firing rate increases lasting from hundreds to thousands of milliseconds encode the position of an animal within the environment ('place cells' (Muller and Kubie, 1987; O'Keefe and Dostrovsky, 1971)), routes through paths with overlapping segments ("splitter cells" (Duvelle et al., 2023; Frank et al., 2000; Wood et al., 2000)), signal goal-locations (Hok et al., 2007), mark time intervals (Eichenbaum, 2014), respond to specific odors (Eichenbaum et al., 1987), sounds (Aronov et al., 2017), objects (Fried et al., 1997) and, in humans, to other people's identities (Rey et al., 2020).*"

And in the discussion at line 477: "*Thus, while correlation between rate and phase changes has been observed during rate remapping (Sanders et al., 2019), our findings demonstrate that phase-shift coding extends beyond rate remapping and occurs also between distinct place fields or assemblies, frequently coinciding with changes in instantaneous firing rate, similar to those observed for splitter cells (Duvelle et al., 2023; Frank et al., 2000; Wood et al., 2000).*"

R2.48 - L310 "generally, units that received the highest depolarization activated first within the theta cycle while, importantly, units receiving just a light depolarization terminated the activation sequence." => this seems related to a recent paper looking at replay sequences, Fernandez-Ruiz et al, 2019 (<https://science.sciencemag.org/content/364/6445/1082.editor-summary>), that the authors could discuss.

The implication that sequential activation of place cells during (prolonged) ripple-associated replay may reflect depolarisation-dependent mechanisms similar to those at play during theta sequences is certainly an interesting one. However, at present, we do not feel comfortable extrapolating our results this far.

R2.49 26. L529 "phase histogram of A spikes are shown" – what is A? and what is B mentioned later, do they correspond to cells shown in insets a and b?

Thank you for noticing this oversight. We now rephrased the sentence in "(e) Cross-correlation between the spikes of two units, *unit n and unit m*, during choice (above) and guided (below) right trials. *Unit n changes firing phase when active in choice and guided trials ($p = 0.03$, phase histograms of n's spikes are shown as inset for the two trial types) thereby changing its relative lag of activation with unit m during theta cycles (two-tailed Wilcoxon rank sum test, $p = 0.01$).*"

R2.50 27. L330 "we showed that such enhanced locking is due to the coordination within spike-assemblies" This seems to imply a causal link which has not been demonstrated. Please reformulate or provide an explanation to support this causal statement.

We now reformulated the sentence, line 419, as "*We show that such theta locking is most pronounced for spike-assemblies, but present in rate-assemblies as well.*"

R2.51 28. L374 "we found that the changes in phase preference also co-occurred with changes in the instantaneous firing rate of the unit ": where is this shown?

This was discussed in the text at line 340: "*In line with this hypothesis, we found that differences in average phase preference between two sets of spikes also co-occurred with differences in the average instantaneous firing rate (see Methods for methodological details). This was true when comparing spikes*

from different place-field locations, within the same location but from different trial types, or spikes occurring as part of different spike- or rate-assemblies (chi-square test of independence on pairs of spike-sets with p-value > or < 0.05 when testing phase and instantaneous firing rate differences: different place fields: $\chi^2(1, N = 2165) = 28.9, p = 7.5 \cdot 10^{-8}$, same location different trial types: $\chi^2(1, N = 1004) = 20.5, p = 5.9 \cdot 10^{-6}$, spike-assemblies: $\chi^2(1, N = 318) = 4.9, p = 0.027$, rate-assemblies: $\chi^2(1, N = 5078) = 63.2, p = 2.0 \cdot 10^{-15}$).

Reviewer #3 (Remarks to the Author):

This manuscript by Russo et al studies how the hippocampal place cell activities depend on task variables other than those directly related to place field properties, including task demand and the future and past of spatial trajectories. The authors examined two types of activities. One is participation in a cell assembly and the other is theta phase. They found that same cells can participate in different cell assemblies, which are expressed in two related time scales, and display different preferred theta phases, when the cells have multiple place fields that reflect different levels of task demand (choice vs guided trials) or different past/future trajectories. Although the topic has been examined in many previous studies, the results obtained here are valuable. However, there are a couple of issues that could be addressed to improve the manuscript.

We thank the Reviewer for appreciating our work. We hope that the new data, analyses, and additions to the text helped improving the manuscript and allay the remaining doubts.

R3.1 1. A key question is whether the main result in the manuscript can be explained by the **rate remapping** phenomenon together with phase precession, without assigning phase preference itself into a functional role. A major part of the manuscript examines different phase preferences when cells participate in different cell assemblies or along different running trajectories. The underlining reason for all these seems due to the rate differences of same place fields along different trajectories, **which is the well-known “split cell” phenomenon or trajectory-dependent rate remapping**. Across the manuscript, it seems that low rate is associated with phases around 0/360 degree and high rate with 90-270 degree (examples in Figs. 4- 6). Therefore, phases differences in the same field across different conditions maybe just passively reflect the rate remapping, due to less degree of phase precession (e.g. not many spikes reaching the 180-270 phases, which typically associated with high instantaneous rates) when firing rate is low. The authors may consider explicitly to investigate this possibility.

We do believe that the intracellular mechanism that is at the basis of phase precession plays a big role in what we observe. As simulated in our proposed model, the activation of a cell assembly, feeds into the CA1 units an assembly-characteristic degree of depolarization. This degree of depolarization is assembly-specific and is the same, every time the animal finds itself in the same task condition (thus activating the same assembly). As shown by the model, different depolarizations result in different phases of spiking of the units. Thus, if the unit receives a progressive depolarization as the animal crosses the place field the mechanism could give rise to the classic phase precession phenomenon. If the depolarization of the unit instead changes discretely with the activation of the assembly, the model predicts discrete values of phase and rate. At the rate level this corresponds to the splitter cell phenomenon (now explicitly cited and discussed in the manuscript).

Despite the common mechanism, the here observed phase modulation serves information coding much beyond phase precession. Phase precession refers to the shift to earlier times in the theta cycle of the spiking phase of a place cell as the animal moves through that unit's spatial receptive field. In this context, the phase of firing encodes the position of the animal with respect to a specific place field. Here we show that the firing phase can also encode non spatial information on a same location (similarly to remapping / splitter cells for rate coding) or different locations in the maze.

The synergy between rate and phase coding in carrying contextual and spatial information allows to propagate the encoded information through multiple pathways and trigger different network processes: While rate coding is more robust than phase coding (as integrates over many spikes) and can be read reliably by downstream regions in hundreds of ms, The information carried by phase coding is more dynamic, as relying on few spikes, and is immediately readable in milliseconds. Moreover, if unit rate

(or better the unit depolarization) and phase correlate, as shown by our results, the spiking order of two neurons along the theta sequence is flexible and can vary according to the assembly active in a specific context. This flexibility first largely broadens the pool of sequences that can be produced by a set of units, and second, allows the creation of theta sequences that can distinguish different contexts in the same location. Finally, the temporal coordination between different units of a theta sequence is much higher than that provided by their rate coordination (this is captured explicitly by the cell assembly analysis which assigns, in an unsupervised manner, a smaller temporal resolution at spike-assemblies than rate assemblies). This enhanced coordination facilitates spike-timing dependent plasticity and other forms of LTP likely to shape sequential learning over the course of experience.

To clarify this line of thoughts:

- We now refer to 'splitter cells', which we unintentionally omitted to explicitly reference, at line 290: *"Moreover, while it is known that splitter cells can differentiate between trial types by rate modulation, we found that adding information relative to the spike phase to the instantaneous firing rate further improved the decoding performance of the SVM (generalized linear mixed-effects model of the ...", at line 38:"Among pyramidal cells of hippocampal CA1, transient firing rate increases lasting from hundreds to thousands of milliseconds encode the position of an animal within the environment ('place cells' (Muller and Kubie, 1987; O'Keefe and Dostrovsky, 1971)), routes through paths with overlapping segments ("splitter cells" (Duvelle et al., 2023; Frank et al., 2000; Wood et al., 2000)), signal goal-locations (Hok et al., 2007), mark time intervals (Eichenbaum, 2014), respond to specific odors (Eichenbaum et al., 1987), sounds (Aronov et al., 2017), objects (Fried et al., 1997) and, in humans, to other people's identities (Rey et al., 2020)." And in the discussion at line 477: "Thus, while correlation between rate and phase changes has been observed during rate remapping (Sanders et al., 2019), our findings demonstrate that phase-shift coding extends beyond rate remapping and occurs also between distinct place fields or assemblies, frequently coinciding with changes in instantaneous firing rate, similar to those observed for splitter cells (Duvelle et al., 2023; Frank et al., 2000; Wood et al., 2000)."*
- We have substantially revised the Discussion (which we have not copied here for reasons of space) to more clearly delineate the connections and distinctions between our findings and the extant literature on remapping, splitter cells, and theta phase modulation (please, see "Discussion" section).

R3.2 2. It seems that place fields close to the reward sites were included in the analysis. Since theta is reduced and animal speed is different (lower) around the reward sites, the author may consider to remove place fields close the reward sites.

We thank the reviewer for this suggestion. We now added further analyses and controls to explicitly address these confounds:

First, all single unit analyses are now performed including only spikes fired when the animal is walking at a speed of at least 12 cm/sec.

Second, to assess whether the observed encoding of trial information by spiking phase could be explained by confounding factors, such as speed and theta power, we tested whether adding phase information improved the accuracy of a classifier trained on speed and theta power. The test confirmed

that incorporating the phase of spiking provided information for decoding the trial type supplementary to that already present in the two covariate factors.

Reference to this control test is now reported at line 276: *“This observation was corroborated by training a support vector machine (SVM) classifier on the phase of spikes fired in an individual place field to distinguish between trial types. We found that for 32% of units at least two trial types could be distinguished above chance level within at least one of the unit place fields (see Methods for details). This result could not be explained by covariates such as the animal speed and the ongoing theta power (a general linear mixed-effects model of the accuracy of an SVM classifier trained on speed and theta power or on speed, theta power, and theta phase of each spike found a significant increase in accuracy for the latter classifier. Contrast tests between the two classifier types: $F(1,530) = 8.9$, $p = 0.003$. Test computed on the subset of place fields which could distinguish above chance trial identity in the latter, more powerful, classifier) ...”*

Third, Supplementary Fig. 9 and 11, replicate the analyses of Supplementary Fig. 8,10 but excluding spikes fired at low theta phase and during SWR.

R3.3 3. This paper by Sanders et al (Hippocampus, 29: 111, 2019) is highly relevant and should be cited.

We thank the reviewer for this suggestion, the paper is indeed highly relevant and has now been cited in the Discussion (lines 477):

“Thus, while correlation between rate and phase changes has been observed during rate remapping (Sanders et al., 2019), our findings demonstrate that phase-shift coding extends beyond rate remapping and occurs also between distinct place fields or assemblies, frequently coinciding with changes in instantaneous firing rate, similar to those observed for splitter cells (Duvette et al., 2023; Frank et al., 2000; Wood et al., 2000).”

REVIEWERS' COMMENTS

Reviewer #2 (Remarks to the Author):

All my concerns have been addressed very thoroughly and convincingly by the authors via additional explanations, analyses, figures, data and discussion points. I believe this study will make an important and rigorous contribution to the literature of spatial cognition and decision-making. I only have a few minor comments below that do not require an answer but that the authors might want to consider for what I think should be the next and final version of the manuscript.

Minor comments:

1. IN fig 1g, the two rate maps clearly belong to two different sessions (as evidenced by the different distributions of visited bins) but they are showing units belonging to the same assembly – this looks like an error?

2. Thank you for showing that the results withstand removing the swr-occurring spike assemblies. It would be good to indicate how many assemblies (in total and also per rat and/or session) were removed by this.

3. Related to this, L410 (discussion): “and a sub-second temporal scale (‘spike-assemblies’) compatible with the entrainment of spikes by theta rhythms and during SWR-associated replay events.” This seem to suggest that the assemblies could be associated to theta or SWR LFP state, indifferently. A quantification of whether this is the case or if instead they relate more to one specific LFP state would be very useful. Indeed, if assemblies were mostly theta-related, it might speak to the variability of swr-related activity, which usually represent longer trajectories or even trajectories from other environments, compared to the consistency of theta sequence activity which repeats very similar patterns every time the animal runs on the same part of the track. Perhaps this might be worth mentioning in the discussion.

4. L 214 “spatial information of 2.25 ± 0.06 when compared with 1.52 ± 0.05 ”: could units be added?

5. L 662 “Place cells firing phase can encode distinct place fields”: according to the response to reviewer, point R2.35, this should have been updated to “The theta firing phase of place cells can discriminate between distinct place fields of the same unit” – however, the original title, which I still find unclear, is still present.

Minor typos:

L 1369 Rasteplots -> Rasterplots

I 665: DBBSCAN -> DBSCAN?

L 284 + the following text, which re-occurs in other places, is a bit hard to follow. Its readability could be improved. “Contrast tests between the two classifier types: $F(1,530) = 8.9$, $p = 0.003$. Test computed on the subset of place fields which could distinguish above chance trial identity in the latter, more powerful, classifier) nor the sorting cluster quality of the units (generalized linear mixed-effects model of a unit to phase-shift, either for different place fields or within the same place field but different trials, based on its L-ratio, recording session and animal; with binary dependent variable for significant phase change and logit link function: $F(1,60) = 0.12$, $p = 0.74$).

L 467 “While a recent study has shown that anatomically distinct place cell subpopulations in superficial and deep sublayers of dorsal CA1 pyramidal cell layer bias towards rate and phase coding of spatial information respectively,” -> Should it be ‘are biased’?

L 471 “to supports coding” -> to support

L527: “with place field far from the animal but” -> place fields ?

Reviewer #3 (Remarks to the Author):

The authors' responses to the review comments are extensive. All of my comments are adequately addressed. I thus support the publication of this manuscript.

POINT-BY-POINT RESPONSE TO THE REVIEWERS' COMMENTS

Thank you for your continued support with our manuscript. We report here our responses to the final minor comments of the reviewers, noting that Reviewer 2's helpful comments are framed as "considerations".

Reviewer #2 (Remarks to the Author):

All my concerns have been addressed very thoroughly and convincingly by the authors via additional explanations, analyses, figures, data and discussion points. I believe this study will make an important and rigorous contribution to the literature of spatial cognition and decision-making. I only have a few minor comments below that do not require an answer but that the authors might want to consider for what I think should be the next and final version of the manuscript.

Minor comments:

1. IN fig 1g, the two rate maps clearly belong to two different sessions (as evidenced by the different distributions of visited bins) but they are showing units belonging to the same assembly – this looks like an error?

Figure 1g is correct and presents the rate maps of two distinct assemblies, each detected in separate sessions, as indicated in the figure legend and annotated within the figure itself. The rate maps of the individual units taking part to assemblies are shown in Supplementary Figure 3.

2. Thank you for showing that the results withstand removing the swr-occurring spike assemblies. It would be good to indicate how many assemblies (in total and also per rat and/or session) were removed by this.

We now added this information in the figure legend of Fig. 1b and Supplementary Fig. 4 where we show the distribution of the temporal resolution of all detected assemblies detected including or not SWR-epochs, respectively.

Fig. 1b caption: "... (b) The distribution of the temporal precision of the assemblies detected during the spatial working memory task shows the presence of two predominant timescales: one peaked around 28 ms and one on the second scale. Bars show weighted mean and SE computed across 6 animals and 4 sessions (sessions without assemblies were excluded, n. sessions = 22, n. assembly pairs = 4914)..."

Supplementary Fig. 4 caption: "... Bars show weighted mean and SE computed across 6 animals and 4 sessions (sessions without assemblies were excluded, n. sessions = 23, n. assembly pairs = 4826)."

3. Related to this, L410 (discussion): "and a sub-second temporal scale ('spike-assemblies') compatible with the entrainment of spikes by theta rhythms and during SWR-associated replay events." This seem to suggest that the assemblies could be associated to theta or SWR LFP state, indifferently. A quantification of whether this is the case or if instead they relate more to one specific LFP state would be very useful. Indeed, if assemblies were mostly theta-related, it might speak to the variability of swr-related activity, which usually represent longer trajectories or even trajectories from other environments, compared to the consistency of theta sequence activity which repeats very similar patterns every time the animal runs on the same part of the track. Perhaps this might be worth mentioning in the discussion.

In the Results and Discussion sections of the manuscript, we discuss the possibility that some of the detected assemblies are associated with sharp-wave ripples (SWRs). To determine whether the phase shift effect described in this manuscript depends or not on SWR-associated assemblies, we repeated

all analyses on assemblies detected excluding spikes fired during SWRs. These analyses confirmed the presence of a contextual phase shift, leading us to conclude that the effect is not dependent on SWRs.

While investigating assemblies active during SWRs and the differences from those linked to theta sequences is a compelling topic, it is broad and goes beyond the scope of this manuscript. Thus, we believe that adding further speculation on the characteristics of SWR-associated assemblies goes beyond the scope of this manuscript and thus could confuse the readers.

4. L 214 “spatial information of 2.25 ± 0.06 when compared with 1.52 ± 0.05 ”: could units be added?

Thanks for noticing it, we now added the information.

“...resulting in average spatial information of 2.25 ± 0.06 bit/s when compared with 1.52 ± 0.05 bit/s for single putative place cells...”

5. L 662 “Place cells firing phase can encode distinct place fields”: according to the response to reviewer, point R2.35, this should have been updated to “The theta firing phase of place cells can discriminate between distinct place fields of the same unit” – however, the original title, which I still find unclear, is still present.

We apologize for this oversight, we now corrected the title accordingly.

Minor typos:

L 1369 Rasterplots -> Rasterplots

We now corrected the typo.

L 665: DBSCAN -> DBSCAN?

We now corrected the typo.

L 284 + the following text, which re-occurs in other places, is a bit hard to follow. Its readability could be improved. “Contrast tests between the two classifier types: $F(1,530) = 8.9$, $p = 0.003$. Test computed on the subset of place fields which could distinguish above chance trial identity in the latter, more powerful, classifier) nor the sorting cluster quality of the units (generalized linear mixed-effects model of a unit to phase-shift, either for different place fields or within the same place field but different trials, based on its L-ratio, recording session and animal; with binary dependent variable for significant phase change and logit link function: $F(1,60) = 0.12$, $p = 0.74$).

We now modified the paragraph to improve readability:

“This result could not be explained by covariates such as the animal speed and the ongoing theta power. In fact, we found that adding phase information to a classifier built on the animal speed and on theta power increased its accuracy (Classifier 1: SVM on speed and theta power; Classifier 2: SVM on speed, theta power, and theta phase. General linear mixed-effects model of Classifier 1 and Classifier 2 accuracy, contrast tests between the two classifier types: $F(1,530) = 8.9$, $p = 0.003$. Test computed on the subset of place fields which could distinguish above chance trial identity by Classifier 2, more powerful and thus granting a larger sample size). The observed relation between phase of firing and trial type could also not be explained by the sorting cluster quality of the units (generalized linear mixed-effects model of a unit to phase-shift, either in different place fields or within the same place field but in different trials, based on its L-ratio, recording session and animal; with binary dependent variable for significant phase change and logit link function: $F(1,60) = 0.12$, $p = 0.74$).”

L 467 “While a recent study has shown that anatomically distinct place cell subpopulations in superficial and deep sublayers of dorsal CA1 pyramidal cell layer bias towards rate and phase coding of spatial information respectively,” -> Should it be ‘are biased’?

We now corrected as suggested.

L 471 “to supports coding” -> to support

We now corrected as suggested.

L527: “with place field far from the animal but” -> place fields ?

We now corrected as suggested.

Reviewer #3 (Remarks to the Author):

The authors' responses to the review comments are extensive. All of my comments are adequately addressed. I thus support the publication of this manuscript.

No changes required.